# Response of low to mid-latitude ionosphere to the geomagnetic storm of September 2017

Nadia Imtiaz[1], Waqar Younas[2], and Majid Khan[2]

[1]Theoretical Physics Division, PINSTECH, Nilore, Islamabad, Pakistan
[2]Department of Physics, Quaid-i-Azam University, Islamabad, Pakistan

**Correspondence:** Nadia Imtiaz (nhussain@ualberta.ca)

**Abstract.** We study the impact of the geomagnetic storm of 7-9 September 2017 on the low-to-mid latitude ionosphere. The prominent feature of this solar event is the sequential occurrence of two SYM-H minima with values of $-146$ nT and $-115$ nT on 8 September at $1:08$ UT and $13:56$ UT, respectively. The study is based on the analysis of data from the Global Positioning System (GPS) stations and magnetic observatories located at different longitudinal sectors corresponding to Pacific, Asia, Africa and America during the period 4-14 September 2017. The GPS data are used to derive the global, regional and vertical total electron content (vTEC) in the four selected regions. It is observed that the storm time response of the vTEC over the Asian/Pacific sectors is earlier than over the African/American sectors. Magnetic observatory data are used to illustrate the variation in the magnetic field particularly, in its horizontal component. The global thermospheric neutral density ratio; i.e., $O/N_2$ maps obtained from the Global UltraViolet Spectrographic Imager (GUVI) on board the Thermosphere Ionosphere Mesosphere Energetics and Dynamics (TIMED) satellite are used to characterize the storm-time response of the thermosphere. These maps exhibit a significant storm time depletion of the $O/N_2$ density ratio in the northern middle and lower latitudes over the western Pacific/American as compared to the eastern Pacific, Asian and African sectors. However, the positive storm effects in the $O/N_2$ ratio can be observed in the low-latitudes and equatorial regions. It can be deduced that the storm-time thermospheric and ionospheric responses are correlated. Overall, the positive ionospheric storm effects appear over the dayside sectors which are associated with the ionospheric electric fields and the traveling atmospheric disturbances. It is inferred that a variety of space weather phenomena such as the coronal mass ejection, the high speed solar wind stream and the solar radio flux are the cause of multiple day enhancements of the vTEC in the low-to-mid latitude ionosphere during the period 4-14 September 2017.

## 1 Introduction

It is well known fact that the geomagnetic storm is a temporary variation of the Earth's magnetic field induced by the coronal mass ejection (CME) or the high speed solar wind stream (HSSWS). The most widely used indices and parameters to study the physical processes occurring during the geomagnetic storms are: the disturbance storm time ($Dst$) index, the SYM-H index, the Kp index, the Ap index and the $B_z$ component of the interplanetary magnetic field (IMF) (Rostoker (1972); Gonzalez et al. (1994); Saba et al. (1997)). On the basis of different values of the $Dst$ index and the $B_z$ component of the

IMF, the geomagnetic storms can be categorized as follows: weak or minor storms ($Dst \leq -30$ nT, $B_z \leq -3$ nT for 1-hour), moderate storms ($Dst \leq -50$ nT, $B_z \leq -5$ nT for 2-hour), intense storms ($Dst \leq -100$ nT, $B_z \leq -10$ nT for 3-hour) and severe storms ($Dst \leq -200$ nT) (Gonzalez et al. (1994); Loewe and Prolss (1997)). Some scientists prefer to use the SYM-H geomagnetic index in place of the $Dst$ index due to its 1-minute time resolution compared to the 1-hour time resolution of the $Dst$ index (Wanliss and Showalter (2006)). Also the 3-hour value of the Kp index can be used for the classification of the geomagnetic storms as: weak or minor storms ($5- \leq Kp \leq 5$), moderate storms ($Kp \geq 6$), intense storms ($7- \leq Kp \leq 7$) and severe storms ($Kp \geq 8-$) (Gosling et al. (1991)).

During the geomagnetic storms, the ionosphere features vary along the latitudes and longitudes also due to different current systems flowing in the magnetosphere. The physical processes such as the mass transport, prompt penetration of the magnetospheric electric field (PPEF) and an ionospheric disturbance dynamo electric field (DDEF) are the common features of the magnetic storms. A number of models have been utilized to investigate the role of these physical processes in the observations of the global magnetic perturbations (Blanc and Richmond (1980); Fejer (1983); Vasyliunas (1970)). Some theoretical studies devoted to understand the thermal expansion of the thermosphere due to the transport of energy and momentum from the auroral region to the mid-to-low latitudes during a magnetic storm. These studies highlighted the importance of the season and the local time at the beginning of the storm (Fuller-Rowell et al. (1994, 1996)). Sharma et al. (2011) investigated the low latitude ionosphere total electron content (TEC) response to the geomagnetic storm of 25 August 2005. On the day of the storm, a doubly humped peak in the TEC with an amplitude that is almost twice that of a quiet day value is observed. The first peak is attributed to the PPEF however, the second peak is due to the plasma fountain effect. It is also found that the effect of the PPEF is almost uniform along the longitudinal direction. Thomas et al. (2013) studied the storm time TEC variations in the mid-latitude northern American sector. It is observed that the storm time ionosphere response is season dependent; i.e., the storms occurring in the summer have a large negative effects while the winter events have a strong initial positive phase with the minimum negative storm effects. Moreover, the events occurring in the fall and spring have almost the same effects. Many studies have analyzed the St. Patrick day storm (the largest geomagnetic storm of the Solar cycle 24) by using the Global Positioning System (GPS) derived TEC data analysis techniques to understand the positive and negative ionospheric-storm effects due to the energy transfer between the solar wind and the magnetosphere (Fagundes et al. (2016); Nayak et al. (2016)). In this context, Nava et al. (2016) investigated the low and mid-latitude ionospheric response to the St. Patrick day storm of 2015. The storm effects are characterized by using the global electron content (GEC) and regional electron content (REC) in different longitudinal sectors such as Pacific, Asia, Africa and America. The authors observed a strong enhancement of the vertical total electron content (vTEC) in the American sector. It is also found that the Asian sector shows comparatively large decrease in the vTEC. They also used the spectral analysis of the magnetometer data to separate the effects of the convection electric field and of the disturbance dynamo. Zhang et al. (2018) analyzed this event by using the GPS data of the Crustal Movement Observation Network of China. It is found that during the sudden storm commencement (SSC) phase, a rapid enhancement in the ionospheric electron density distorts the structure of the northern equatorial ionization anomaly (EIA) region. It is also observed that during the main phase a significant decrease in the vTEC occurs at the high latitude region as compared to the low latitude region. Moreover, the height of the peak electron density in the F2 layer also increases during

the geomagnetic storm. Watson et al. (2016) presented a study based on the data of about seventeen geomagnetic storms of the solar cycle 24 with $Dst < -100$ nT to identify the solar sources of these geomagnetic storms. It is found that the low geomagnetic activity is associated to the weak dawn-to-dusk solar wind electric field. The authors have shown that the slow CME plays a main role in the commencement of the geomagnetic storms of the solar cycle 24. Kashcheyev et al. (2018) have made a comprehensive analysis on the basis of the two great geomagnetic storms ($Dst \leq -200$ nT) which occurred on 17 March and 22 June 2015. It is found that the absence or presence of a scintillation in the African sector is associated to the local time at the beginning of the storm. Another finding is that the summer storm results into the formation of the plasma bubbles which propagate up to the mid-latitudes and cause strong scintillation in the Global Navigation Satellite System (GNSS) signals. Based upon this comprehensive analysis, the authors suggested that a number of factors such as the local time at the commencement of the storm and the season play an important role in the modeling of the ionosphere response to the solar activity. Blagoveshchensky and Sergeeva (2019) presented a study based on multi-instrument analysis to reveal the variation in the ionospheric parameters during the geomagnetic storm of 6-10 September 2017. The present work aims at investigating the response of low-to-mid latitude ionosphere to the large geomagnetic storm of 6-9 September 2017. The storm effects are analyzed by using the data from the individual GNSS receivers and ground magnetic observatories located in different longitudinal sectors. The approach used in the present study is similar to that used by Nava et al. (2016) and Kashcheyev et al. (2018). In addition, the storm-time response of the neutral atmosphere in the thermosphere is analyzed by using the global $O/N_2$ density ratio maps derived from the Global UltraViolet Imager (GUVI) on board the Thermosphere Ionosphere Mesosphere Energetics and Dynamics (TIMED) satellite. The remainder of this article is organized in the following manner: Section 2 presents a description of data sets used in our analysis. Section 3 briefly describes the case study that is the solar event under investigation and its characterization on the basis of the global plasma parameters. In section 4, we present results and a general discussion of our findings. Finally, the summary/conclusion of this study is presented.

## 2   Data and Analysis

Here we present the characteristics of the solar event along with the data sets that have been used in this study.

For solar event characterization, the relevant information is provided by the National Oceanic and Atmospheric Administration (NOAA) Space Weather Prediction Center (SWPC). According to NOAA SWPC, a number of space weather events were observed between 4-14 September 2017. The detailed description of these events is also given by Redmon et al. (2018). Here we give an overview of these solar events. Several X-class and M-class solar flares along with the CMEs occurred during this period. On 6 September, the sun emitted X2.2 and X9.3 solar flares at $8:57$ and $11:53$ UT, respectively. On 7 September, the two solar flares M7.3 and X1.3 are emitted at $10:11$ and $14:20$ UT, respectively. On 8 September 2017, the M8.1 solar flare is fired off at $15:35$ UT. On 12 September, the X8.3 solar flare is emitted at $15:35$ UT. The associated earthward CMEs have induced the geomagnetic storms of different intensities in the early September 2017.

The solar wind parameters and IMF have been obtained from the OMNI database (https://omniweb.gsfc.nasa.gov/form/dx1.html).

The information about the $B_z$ component of the IMF and the solar wind speed ($V_{sw}$) is provided by the NASA Advanced Composition Explorer (ACE) satellite.

The world data center for Geomagnetism (Kyoto) provides information about different geomagnetic indices, among them are AE, Ap/Kp and SYM-H indices. The AE index is a proxy of the auroral electrojet enhancement which estimates the energy transfer from the solar wind to the auroral ionospheric regions. Both the Ap/Kp indices quantify the disturbance in the horizontal component of the Earth's magnetic field and the SYM-H index measures the intensity of the storm time ring current (Rostoker (1972); Wanliss and Showalter (2006)).

For ionospheric electron density variation, the data sets of the nine GPS stations are analyzed here. These stations are selected on the basis of data availability and their geographic/geomagnetic locations. The geographic and geomagnetic locations of these stations are given in Table 1. Our analysis is based on the four different longitudinal regions: Pacific ($-180° : -120°E$; $150° : 180°E$), Asia ($60° : 150°E$), Africa ($-30° : 60°E$) and America ($-120° : -30°E$). In order to analyze the diurnal variation of the vTEC in different longitudinal sectors, the relevant data with 2-hour time resolution have been extracted from the IGS Global Ionosphere Map (GIM) data available in the IONEX format; (ftp://cddis.gsfc.nasa.gov/gps/products/ionex/). The tomographic kriging GIMs computed by the Technical University of Catalonia (UPC) have been used to study the global electron content (GEC) variations during the storm period under consideration. The GEC is the total number of electrons present in the near-Earth space environment. The GEC is obtained from the UPC GIM data by the summation of the vTEC values in a cell $I_{l,m}$ multiplied by a cell's area $S_{l,m}$ over all GIM cells and it is given by Afraimovich et al. (2006),

$$GEC = \sum_{l,m} I_{l,m}.S_{l,m}. \tag{1}$$

In Eq.1, the symbols $l$ and $m$ represent the latitude and longitude of a certain GIM cell, respectively. The latitudinal and longitudinal extent of the elementary GIM cell is $2.5°$ and $5°$, respectively. The unit of GEC is $1\ GECU = 10^{32}$ electrons. The regional electron content (REC) is the total number of electrons in the specified region of the ionosphere. The REC is calculated similarly to the GEC, with the summation being restricted to the GIM cells of that particular region. For both GEC/REC, the UPC GIM data at the time resolution of 15-minute have been used.

The storm time magnetic field variations are analyzed by using the data obtained from the magnetic observatories located along the geomagnetic equator in the three longitudinal sectors: Asia (Kourou, KOU), Africa (M'Bour, MBO) and America (Guam, GUA). The quasi-definitive data of these observatories available at http://intermagnet.org have been used for the analysis. Table 2 shows the geographic and geomagnetic locations of these observatories. In order to calculate the magnetic field variations, we have adopted the approach of Nava et al. (2016) and Kashcheyev et al. (2018). A brief description of this approach is given here. During the geomagnetic storm, the horizontal component $H$ of the Earth's magnetic field can be expressed as (Le Huy and Mazaudier (2005)):

$$H = H_o + S_R + D_M + D_{iono}, \tag{2}$$

where $H_o$ represents the magnetic field produced in the Earth's core and crust, $S_R$ is the quiet daily variation of the Earth's magnetic field given as $S_q = < S_R >$, $D_M$ is the disturbance which comes from the magnetospheric currents due to the

Chapman Ferraro current, the ring current and the tail current (Cole (1966)) and $D_{iono}$ represents the magnetic field variations due to the disturbed ionospheric currents.

In order to estimate $S_q$, the average value of $\Delta H_i$ is computed from 1-minute time resolution values of the five quietest days in September 2017 by using the following expression:

$$S_{qi} = \frac{1}{n} \sum_{j=1}^{n} \Delta H_i^j, \tag{3}$$

where j is a day number, n is a total number of quiet days and $\Delta H_i = H_i - H_o$ with i=1 to 1440 minutes. The baseline value $H_o$ is an average of hourly values at midnight (LT) and it is computed as:

$$H_o = \frac{H_{22}^j + H_{23}^j + H_{00}^{j+1} + H_{01}^{j+1}}{4}. \tag{4}$$

According to Matsushita and Campbell (1967), the hourly amplitude of the daily variations of the geomagnetic field $S_q$ is subjected to the non-cyclic variation and can be estimated as:

$$\Delta NC = \frac{H_{00}^{j+1} - H_{00}^j}{24}. \tag{5}$$

The corrected hourly solar quiet variation in $H$ that is $S_q(H)$ can be given as:

$$S_{qi}(H) = S_{qi} + \frac{i\Delta NC}{60}, \tag{6}$$

here i=1 to 1440 minutes.

The $D_M$ can be estimated by using the dayside SYM-H index in the following expression:

$$D_M = \text{SYM-H} \times cos\phi, \tag{7}$$

here $\phi$ is the geomagnetic latitude.

The $D_{iono}$ can be estimated using the following expression as given by Le Huy and Mazaudier (2005):

$$D_{iono} = \Delta H - S_q - \text{SYM-H} \times cos\phi, \tag{8}$$

here $\Delta H$ is the variation of the $H$ component of the magnetic field.

The global thermospheric column density $O/N_2$ ratio can also serve as a sensitive indicator in the upper atmosphere for the disturbances induced by geomagnetic storms (Yuan et al. (2015); Wang (2018)). These maps are obtained from the GUVI/TIMED covering the days 5, 8 and 11 September 2017.

## 3   Case Study

In early September 2017, the three CMEs with earthward trajectories are emitted on 4, 6 and 10 September. A CME originating from the massive $X9.3$ solar flare of 6 September reached the Earth at $23:00$ UT on 7 September. The arrival of this CME

caused a significant disturbance in the magnetosphere which leads to a severe geomagnetic storm having the maximum value of the geomagnetic index $Kp_{max} = 8$. However, the arrival of the other two CMEs on 6 and 12 September leads to minor geomagnetic storms of G1 category ($Kp < 3$). Figure 1 illustrates the global morphology of these solar events. In Figure 1, the storm time variations of the various interplanetary plasma and magnetic field parameters are depicted in the following

order (from top to bottom): the $B_z$ component of the IMF, the solar wind speed ($V_{sw}$), the $E_y$ component of the interplanetary electric Field (IEF), the solar radio flux $F_{10.7}$, the SYM-H index, the AE index and the Kp index. The three vertical lines represent the arrival of the CMEs on Earth which lead to the SSC at $23:43$, $23:00$ and $20:02$ UT on 6, 7 and 12 September, respectively, as reported by http://www.obsebre.es/php/geomagnetisme/vrapides/. However, the present study is focused on the effects of the G4 category storm which occurred on 8 September 2017. On the arrival of the interplanetary shock on 7

September at about $23:00$ UT, the initial phase of the storm begins with a rapid variations in the above mentioned parameters. During the main phase, the $B_z$ component of the IMF is more southward reaching the maximum lowest value of about $-32$ nT at 0 UT and then it rapidly increases to the value of approximately $+16$ nT. The $B_z$ becomes southward again by performing a negative excursion of $-17.6$ nT at $11:55$ UT and remains southward until $13:56$ UT. Afterward, the $B_z$ component decreases gradually and stays around 0 nT from 9-12 September. It can be seen that on 8 September, the $V_{sw}$ also exhibits an abrupt

change by attaining the maximum value of about 820 km/s around $02:00$ UT and after $12:00$ UT it gradually decreases. The $E_y$ component of the IEF calculated as $\boldsymbol{E} = -\boldsymbol{V_{sw}} \times \boldsymbol{B}$ also exhibits noticeable variations during the storm period. It depends on the $B_z$ component of the IMF and the $X$ component of the $V_{sw}$. It means that the positive northward IMF leads to the westward IEF on the dayside and the eastward field on the nightside. It can be seen that the $E_y$ fluctuations occur between $-15$ and $+20$ mV/m during the storm. The next plot illustrates the behavior of the solar radio flux $F_{10.7}$. The solar flux fluctuates

significantly during the period 4-14 September 2017. In order to analyze the geomagnetic activity behavior, the SYM-H index is also presented in Figure 1. During the main phase of the storm, the SYM-H index decreases and reaches the negative value of $\simeq -146$ nT thus producing the first minima of the SYM-H index at $1:08$ UT. During the partial recovery phase from $1:08$ UT to $11:00$ UT, the SYM-H also increases from $-146$ nT to the value of $-38$ nT. Thereafter, the SYM-H index decreases again and it reaches the second minimum value of $\simeq -115$ nT. This is the end of the main phase of the storm which lasted for

$\sim 15$ hours. The main phase can be characterized by the occurrence of the two pronounced minima of the SYM-H with values $-146$ nT and $-115$ nT at $1:08$ UT and $13:56$ UT, respectively on 8 September 2017. The recovery phase started after $13:56$ UT on 8 September, the SYM-H increases slowly and returned to its normal value at $14:00$ UT on 11 September. It can be seen that the recovery phase lasted for about 3 days.

The next two plots represent the AE and Kp indices. The AE index shows several peaks during this period. After the arrival of

the first CME, there is an increase in the auroral activity such that the AE index reaches the peak value of about 1430 nT on 7 September at $09:07$ UT. However, the occurrence of the two strong peaks with $AE > 2000$ nT indicates that the most intense auroral activity occurred after the arrival of the second CME. The Kp index shows the two episodes of the maximum value of approximately $Kp = +8$ between $0-3$ UT and $12-15$ UT on 8 September.

## 4   Results and Discussion

In this section, we present the variations in the variety of parameters such as the GEC, the REC, the vTEC, the $H$ component of the magnetic field and the thermosphere neutral composition as a result of the G4 category geomagnetic storm of 7-9 September 2017. Figure 2 shows the $\Delta$GEC (top), the $\Delta$REC (middle) and the SYM-H index (bottom) during the period 4-14

September 2017. Both the $\Delta$GEC and $\Delta$REC are calculated by subtracting the quiet time variation from the value itself. The quiet time variation is computed by using the three quiet days before the storm having the Kp index below 4. The quiet days considered are 2, 3 and 4 September 2017. It can be seen that the $\Delta$GEC shows two positive peaks at $1:08$ UT and $13:56$ UT corresponding to the first and second minima of the SYM-H index, respectively. In order to find the region which contributed to the peaks in the $\Delta$GEC, the $\Delta$REC is plotted for the four longitudinal sectors: Pacific, Asia, Africa, and America. It can

be seen that during the period 4-14 September 2017, the $\Delta$REC varies significantly over the four longitudinal sectors. The observed behavior of the $\Delta$REC can be attributed to the energy inputs from the solar wind to the magnetosphere Nava et al. (2016). The AE index which is an indicator of the energy transfer from the solar wind to the magnetosphere is shown in Figure 1. It can be noticed that the AE index shows several episodes of the energy inputs (having $AE > 1000$ nT) which occurred on 4, 7, 8 and 13 September. In response to these energy inputs, the amplitude and the occurrence time of the maxima/minima of

the REC also vary.

Our analysis shows that the first peak in the GEC is due to the Asian/Pacific sectors and the second peak is due to the African/American sectors. Some authors have analyzed the variations of the global electron content (GEC) with the $10.7 - cm$ solar radio emission; i.e., $F_{10.7}$ index Nava et al. (2016). In order to see the effect of the $F_{10.7}$ index on the GEC, the forth plot illustrates the variation of the $F_{10.7}$ index during the period 4-14 September 2017. It can be seen that the $F_{10.7}$ index is higher

than 100 sfu between 4 to 8 September as shown in Figure 1. During this period, the higher value of the $\Delta$GEC can be noticed which decreases significantly after 9 September as illustrated in Figure 2. According to Afraimovich et al. (2007, 2008) and Nava et al. (2016), there is a correlation between the GEC and the $F_{10.7}$ index. Therefore, it can be inferred that the variation of the GEC from 4 to 8 September can also be affected by the higher solar flux; i.e., $F_{10.7} > 120$ sfu.

The nine plots in Figure 3 illustrate the variation of the vTEC for the individual station of the three longitudinal sectors; i.e.,

Asia, Africa and America, from 4-14 September 2017. In Figure 3, the plots from one to three represent the stations of the Asian sector; i.e., BJFS, BAKO and YAR2, the plots from four to six represent the African sector; i.e., NOTI, NKLG and WIND, the plots from seven to nine represent the stations of the American sector; i.e., BOGT, AREQ and ANTC. On each plot the vTEC is displayed in red and the quiet time daily variations in blue. The quiet time daily variations are computed by averaging the quiet time data of the five days before the storm. These quiet days are chosen on the basis of a minor level

geomagnetic activity having Kp index below 4. The following pertinent features of the vTEC can be noticed:

– An enhancement in the vTEC is observed for all the stations in the three longitudinal sectors on the day of the storm. The three stations in the Asian sector exhibit an increase in the vTEC at the beginning of 8 September. However, the stations in the African region show the increasing trend of the vTEC in the middle and American stations on late 8 September. The variability in the occurrence of the vTEC peaks depend on the local time of the SYM-H minima at these stations.

- On the day of the storm, the northern and southern mid-latitude stations (BJFS and YAR2) in the Asian sector show an increase in the vTEC. However, in the equatorial station (BAKO) relatively less increase in the vTEC is observed.

- In the African region, the largest increase in the vTEC is observed for the equatorial and southern mid-latitude stations (NKLG and WIND) during the storm. However, a small increase in the vTEC can be seen in the northern mid-latitude station (NOTI) in this sector.

- In the American sector, the largest increase in the vTEC is observed for the equatorial station BOGT during the storm period. It can also be noticed that the vTEC decreases significantly for this station after the day of the storm. Both the southern mid-latitude and equatorial trough stations ANTC and AREQ depict a multi-peak structures of the vTEC on the day of the storm. On the day after the storm, the ionization disappears at the southern mid-latitudes and the vTEC returns to its quiet value.

Figure 4 illustrates the variation of the vTEC as a function of time and latitudes over the four longitudinal sectors that are Pacific (first plot), Asia (second plot), Africa (third plot) and America (forth plot). These vTEC plots are extracted from the IGS GIM data which is available in the IONEX files for the entire globe. For a fixed longitude, a vTEC map covering the latitudinal range of $-90°$ to $90°$ can be plotted. The longitudes considered are given as: $150°E$ for Pacific, $110°E$ for Asia, $-10°E$ for Africa and $-70°E$ for America. The vTEC maps shown in Figure 4 cover the period from 4-14 September 2017 and the latitudinal range from ($-90°$ to $+90°$). The SYM-H index over this period is also shown at the bottom in Figure 4. As mentioned earlier, the space weather conditions during this period are highly disturbed due to multiple events such as the CMEs and HSSWS. As a result of these space weather events, the vTEC maps of the four longitudinal sectors also show dramatic variation with the following features:

- During geomagnetically quiet conditions, the vertical $\boldsymbol{E} \times \boldsymbol{B}$ drift at the dip equator lifts the ionospheric plasma upward. Under the influence of the gravitational and pressure gradient forces, the up lifted plasma can diffuse symmetrically with respect to the magnetic equator along the geomagnetic field lines like a plasma fountain. Therefore, the ionospheric electrodynamics generates the fountain effect which leads to the plasma density enhancement also known as the equatorial ionization anomaly (EIA) at $\pm 10 - 15°$ from the equator (Namba and Maeda (1939); Balan and Bailey (1995); Fejer (1991)). As expected, in response to the geomagnetic storm the latitudinal extent of the EIA is increased up to about $30°$ latitudes.

- On 7 September, the observed vTEC enhancement in each sector can be associated with the impact of the first CME which arrived at 23:43 UT on 6 September.

- During the initial phase of the G4 storm on 8 September, the vTEC enhancement mainly occurred in the crest regions of the EIA with a clear latitudinal separation.

- On the day of the storm, the vTEC strongly enhances in the crests of the EIA and in the magnetic equator region as compared to the days before and after the storm. The enhancements of the vTEC in the EIA region in response to the geomagnetic storms have been reported in many studies (Zhao et al. (2005); Astafyeva et al. (2015); Lei et al. (2018)).

– It can be clearly seen that the local dayside sectors such as Asia ($LT = UT + 7$) and Pacific ($LT = UT + 10$) exhibit the largest increase in the vTEC on early 8 September corresponding to the first SYM-H minima. However, at the time of the second minima the other two sectors that is America ($LT = UT - 5$) and Africa ($LT = UT - 1$) are on the dayside and show the largest increase in the vTEC.

– In the Asian sector, a regular pattern of the vTEC which consists of a well defined crests can be observed except on the day of the storm. However, both the African and Pacific sectors show an irregular patterns; i.e., sometimes one and sometimes two crests of the vTEC appear. During the recovery phase on 9 September, the vTEC returns back to its normal pattern. In the American sector, we mostly observed one crest of the vTEC and a very strong ionization on the day of the storm which returns to its normal level after the storm on 9 September.

– An enhancement in the vTEC in particular, in the crest regions of the EIA also occurred on 5 and 11 September. The observed vTEC enhancement can be attributed to the HSSWS effect (Nava et al. (2016)).

The observed dayside positive storm phases can be explained on the basis of the two phenomena; i.e., the PPEF and the traveling atmospheric disturbances (TADs). The TADs originated from the polar regions due to a large amount of energy deposition from the magnetosphere during the storm period can move to the low latitudes and across the equatorial regions (Fuller-Rowell et al. (1994)). The propagation of these TADs to the low latitudes and across the equator can cause disturbance in the ionosphere by moving the plasma up and down along the magnetic field lines. According to Lei et al. (2018), the upward vertical $\boldsymbol{E} \times \boldsymbol{B}$ drifts are very strong between $01 - 14$ UT on 8 September. These enhanced $\boldsymbol{E} \times \boldsymbol{B}$ drifts can be associated with the PPEF and the disturbance dynamo electric fields (DDEFs) driven by the thermospheric winds (Spiro et al. (1988); Blanc and Richmond (1980)). During the main phase of the storm, the upward drifts can shift the plasma to the higher altitudes where the chemical losses are very small and it can induce a super fountain effect. On the other hand, the equatorward winds can inhibit the downward diffusion of plasma (Balan et al. (2010)). The combined effect of the electric field and thermospheric wind can lead to the enhancement of the vTEC in the EIA. Besides these factors, the solar radio flux $F_{10.7}$ which varies greatly during this period can also affect the vTEC (Lei et al. (2018)).

The three plots in Figure 5 represent the magnetic field variations at the three equatorial magnetic observatories corresponding to the three longitudinal sectors of Asia (GUA), Africa (MBO) and America (KOU). Each plot shows the variation in the $H$ component of the magnetic field (in black), the quiet daily variation ($S_q$) (in blue) and the ionospheric disturbances ($D_{iono}$) (in red). The three dashed lines correspond to the impact of the CMEs on 6, 7 and 12 September. The following features of the $H$ component can be noticed in all the three sectors:

– Firstly, an increase in the $H$ component occurs during the initial phase of the storm. This enhancement is due to the Chapman-Ferraro current resulting from the contraction of the magnetosphere (Chapman and Ferraro (1931)).

– Secondly, a strong decrease in the $H$ component can be observed during the main phase of the storms. It can be attributed to the diamagnetic behavior of the equatorial ring current. The enhanced ring current in the magnetosphere induced the

**Table 1.** Geographic latitude, Geographic longitude, Geomagnetic latitude and Geomagnetic longitude of the GPS stations located in different regions used in the analysis.

| Station | Sector | GLAT | GLONG | MLAT | MLONG |
|---------|--------|------|-------|------|-------|
| BJFS | Asia | $39.60°N$ | $115.89°E$ | $30.23°N$ | $172.23°W$ |
| BAKO | Asia | $6.49°S$ | $106.85°E$ | $16.03°S$ | $179.68°E$ |
| YAR2 | Asia | $29.04°S$ | $115.35°E$ | $38.35°S$ | $170.85°W$ |
| NOTI | Africa | $36.87°N$ | $14.98°E$ | $36.43°N$ | $94.94°E$ |
| NKLG | Africa | $0.35°N$ | $09.67°E$ | $1.59°N$ | $82.67°E$ |
| WIND | Africa | $22.57°S$ | $17.09°E$ | $22.09°S$ | $86.00°E$ |
| AREQ | America | $16.50°S$ | $71.50°W$ | $6.82°S$ | $1.30°E$ |
| BOGT | America | $4.64°N$ | $74.08°W$ | $14.19°N$ | $1.27°W$ |
| ANTC | America | $37.34°S$ | $71.53°W$ | $27.58°S$ | $1.18°E$ |

**Table 2.** Locations of the Magnetometers used in the analysis.

| Station | Sector | GLAT | GLONG | MLAT | MLONG |
|---------|--------|------|-------|------|-------|
| GUA | Asia | $13.59°N$ | $144.87°E$ | $5.87°N$ | $143.28°W$ |
| MBO | Africa | $13.34°N$ | $16.97°W$ | $18.48°N$ | $58.16°E$ |
| KOU | America | $5.21°N$ | $52.93°W$ | $14.17°N$ | $20.48°E$ |

magnetic field opposite to the Earth's northward dipole field which strongly reduces the $H$ component (Gonzalez et al. (1994)).

– Following the strongest decrease in the $H$ component, the recovery phase started which lasted for several hours. During the recovery phase, the ring current decays and the $H$ component of the magnetic field returns back to the normal level.

5 – Two pronounced dips in the $H$ component at $1:08$ UT and $13:56$ UT on 8 September are observed in the three stations. It can be seen that the first dip (around $1:08$ on 8 September) is strongly negative for both GUA($-142$ nT) and KOU ($-142.5$ nT) as compared to MBO ($-102$ nT). However, the second dip (around $13:56$ on 8 September) is strongly negative for MBO ($-164$ nT) as compared to GUA ($-133$ nT) and KOU($-112$ nT). This behavior is due to the local time variation of the ring current during the storm.

10 – Overall, the largest disturbance of the $H$ component of the magnetic field is observed at MBO as compared to GUA and KOU.

The disturbance due to ionosphere electric current $D_{iono}$ is represented by the red curve in Figure 5. It follows anti-$S_q$ signature during the storm period. It can be noted that during the first southward excursion of the IMF, the $D_{iono}$ decreases at GUA which is in the noon sector. However, an increasing trend in the $D_{iono}$ is observed for MBO and KOU which are in the night sector.

15 During the second southward excursion of the magnetic field, $D_{iono}$ decreases significantly for MBO and KOU which are now

in the dayside. The $D_{iono}$ contains signatures of the PPEF and DDEF therefore, the observed trend of the $D_{iono}$ can also be explained by these two electric fields (Kashcheyev et al. (2018)).

Another effect that can be seen during the storm is the variation in the thermospheric neutral composition; i.e., the $O/N_2$ density ratio. It is well understood that the thermospheric composition plays important role in the dayside ionospheric density variation. Under quiet conditions, the photoionization of the atomic oxygen and the chemical reaction of the molecular nitrogen $N_2$ and oxygen ion $O^+$ mainly control the ionospheric density. During the intense geomagnetic storms, a large number of oxygen atoms are ionized that leads to an increase in the ionospheric electron density along with the high $O/N_2$ density ratio. This in turn affects the ionospheric TEC and vTEC. The global view of the thermospheric $O/N_2$ ratio obtained from the TIMED/GUVI for the days before, during and after the storm of September 2017 is shown in Figure 6. As evident, a severe storm time depletion of the $O/N_2$ ratio occurs in the higher latitudes while a significant enhancement in the $O/N_2$ ratio is observed in the lower latitudes and equatorial regions. The $O/N_2$ ratio is mainly controlled by the thermospheric neutral winds that are, in turn, related to the Joule heating in high latitudes. Therefore, the severe reduction of the thermospheric $O/N_2$ density ratio in the polar region is caused by the up-welling wind due to enhanced joule and particle heating in the high-latitudes. According to Yuan et al. (2015), the $O/N_2$ density depletion extends from the high-to-mid latitudes due to the expansion of the storm induced heating zone which causes upward flow of the heated $N_2$ enriched air. The storm time depletion of the $O/N_2$ ratio is found to extend to middle and lower latitudes over the north American and western Pacific sectors. On the other hand, the storm time enhancement of the $O/N_2$ density ratio can be noticed in the mid-latitudes over the African, Asian and eastern Pacific sectors as compared to the quiet time pattern. The storm time longitudinal asymmetry of the ionospheric and thermospheric disturbances is associated with the asymmetric longitudinal distribution of the $O/N_2$ density ratio. Moreover, the asymmetric structure of the $O/N_2$ density ratio strongly depends on the location of northern and southern magnetic poles in different hemispheres (Wang (2018); Fuller-Rowell et al. (1994)). After the recovery of the storm on 11 September, the thermospheric composition returns to its normal profile. This observation is consistent with the behavior of the vTEC during the storm period.

## 5   Conclusions

We presented the impact of the geomagnetic storm of 7-9 September 2017 on the low-to-mid latitude ionosphere over the four longitudinal sectors; i.e., Pacific, Asia, Africa and America. The storm effects are characterized by using the diverse parameters including the global, regional and vertical total electron content derived from the GPS data, the geomagnetic field measured at the ground magnetic observatories and the thermospheric neutral composition obtained from the TIMED/GUVI instrument. It is observed that the positive storm effects occur in the local dayside stations. The temporal response of the four sectors shows that the positive storm effects in the REC and vTEC over the Asian/Pacific sectors are observed earlier than the American/African sectors. During geomagnetically quiet conditions, most of the TEC is confined to the equatorial and low latitude regions. However, the latitudinal extent of the bulk of the TEC increases up to the mid-latitudes during the storm period. The vTEC enhancements observed on the other days are due to the high speed solar wind stream event. The analysis of the magnetometers data shows the largest disturbance of the horizontal component of the magnetic field occurred at MBO

as compared to that of GUA and KOU. The storm time variation of the horizontal component is associated with the Chapman-Ferraro and the ring currents. The magnetic field component associated with the disturbed ionospheric current follows the anti-Sq variations which depends on the prompt penetration electric field and the disturbance dynamo electric field. On the day of the storm, the $O/N_2$ density ratio is larger than that in the quiet time over the low latitudes and equatorial region. However, the high and mid-latitudes exhibit storm time depletion of the thermospheric $O/N_2$ density ratio. The storm time longitudinal asymmetric behavior of the thermosphere can also be observed in the lower and middle latitudes over the four sectors. It is found that the thermospheric $O/N_2$ density ratio in the lower and middle latitudes over the African, Asian and eastern Pacific sectors is larger than that it is observed over the American and western Pacific sectors. Moreover, the storm time enhancement in the thermospheric composition (i.e. $O/N_2$ ratio) over the low latitudes and equatorial region is consistent with the observed vTEC behavior. Overall, the positive storm phase occurred on the dayside sectors during the G4 geomagnetic storm of 7-9 September 2017. It can be concluded that the thermosphere-ionosphere dynamics and electrodynamics play important role in the observed perturbations in the low-to-mid latitude ionosphere during the geomagnetic storms of 4-14 September 2017. The study would be useful to understand the response of the low-to-mid latitude ionosphere during the geomagnetic storms.

*Acknowledgements.*  The authors are grateful to the space weather data resources: the OMNI data base https://omniweb.gsfc.nasa.gov/form/dx1.html for providing the solar wind data, to the World Data Center for Geomagnetism at Kyoto University, Japan for providing the geomagnetic data: http://wdc.kugi.kyoto-u.ac.jp/dstrealtime/ and to the International GNSS Service (IGS) team for providing the GPS data. NI acknowledge the UNOOSA/ICTP for providing the financial support to attend GNSS workshop and learn these data analysis techniques. The authors are very grateful to the anonymous referees for their constructive and insightful comments in improving the manuscript. The authors would like to thanks Anton Kashcheyev and Christine Amory Mazaudier for their valuable suggestions. This research work was partly supported by the HEC Pakistan: $GrantNo.7632/Federal/NRPU/RD/HEC/2017$.

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

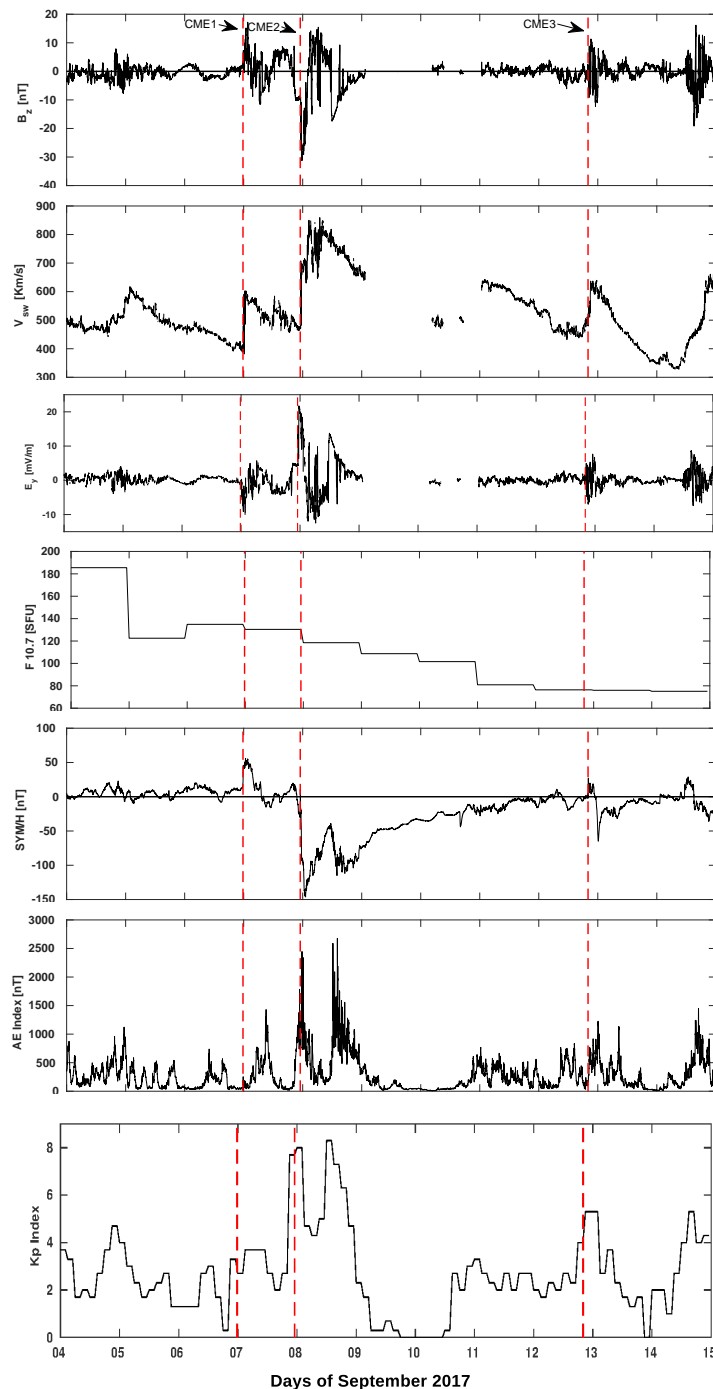

**Figure 1.** Solar wind parameters and geomagnetic indices characterizing the geomagnetic storm during 4-14 September 2017. From top to bottom: $B_z$ component of the magnetic field, solar wind velocity $V_{sw}$ in km/s, $E_y$ component of the interplanetary electric field, $F_{10.7}$, SYM-H index, AE index and the bottom panel illustrates Kp index.

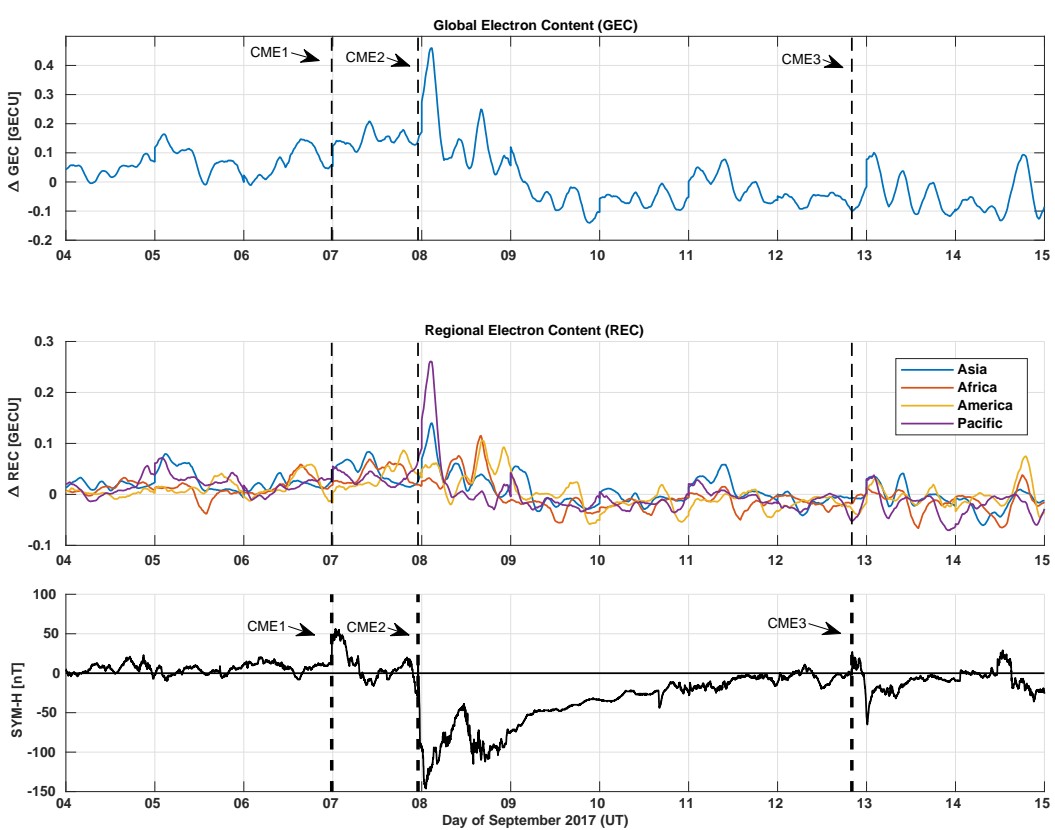

**Figure 2.** Variation of the Global electron content (top), the Regional electron content (middle) and the SYM-H index (bottom) during the geomagnetic storm of 4-14 September 2017.

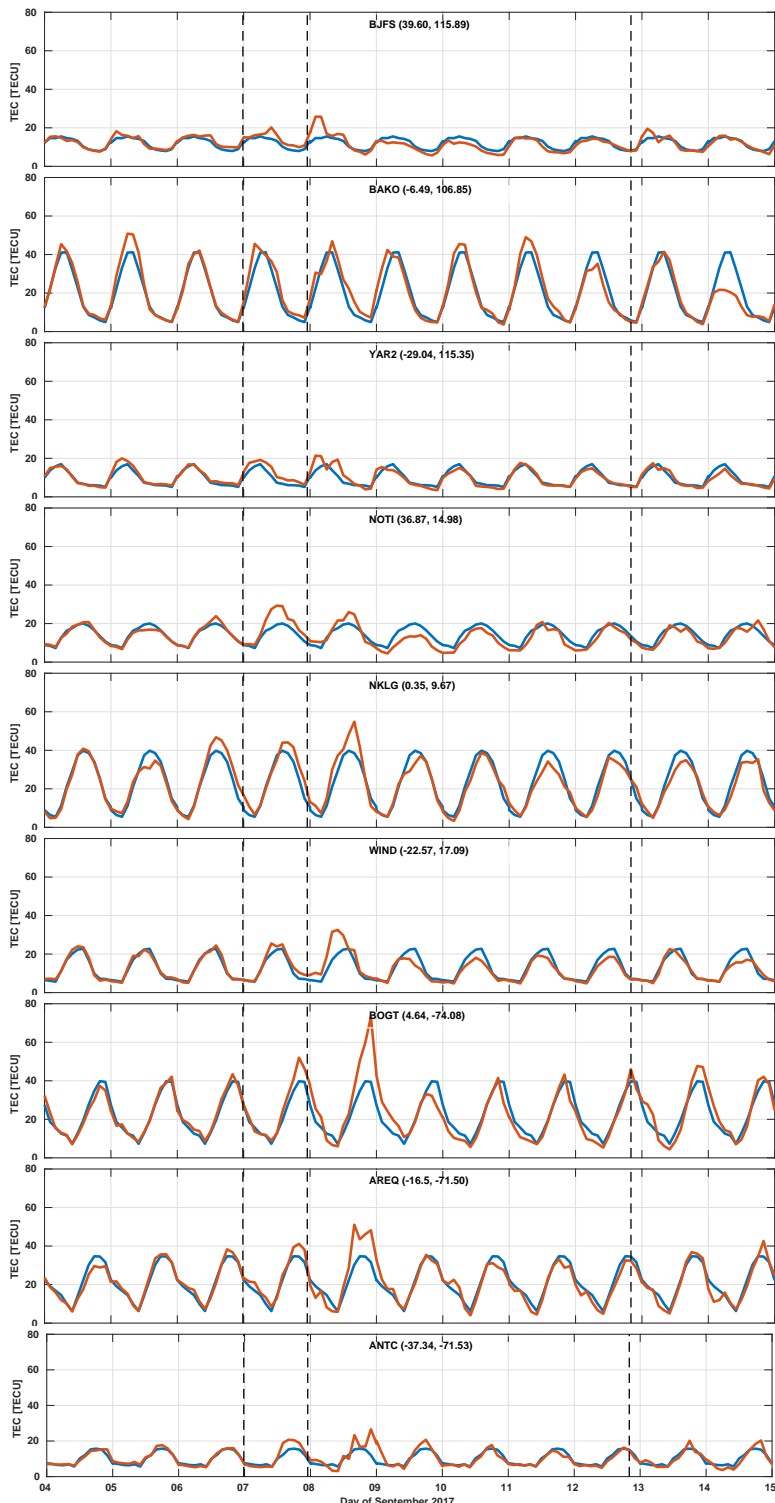

**Figure 3.** The vTEC variations at GPS stations during the geomagnetic storm of 4-14 September 2017. Each plot illustrates the disturbed vTEC (in orange) and its quiet value (in blue).

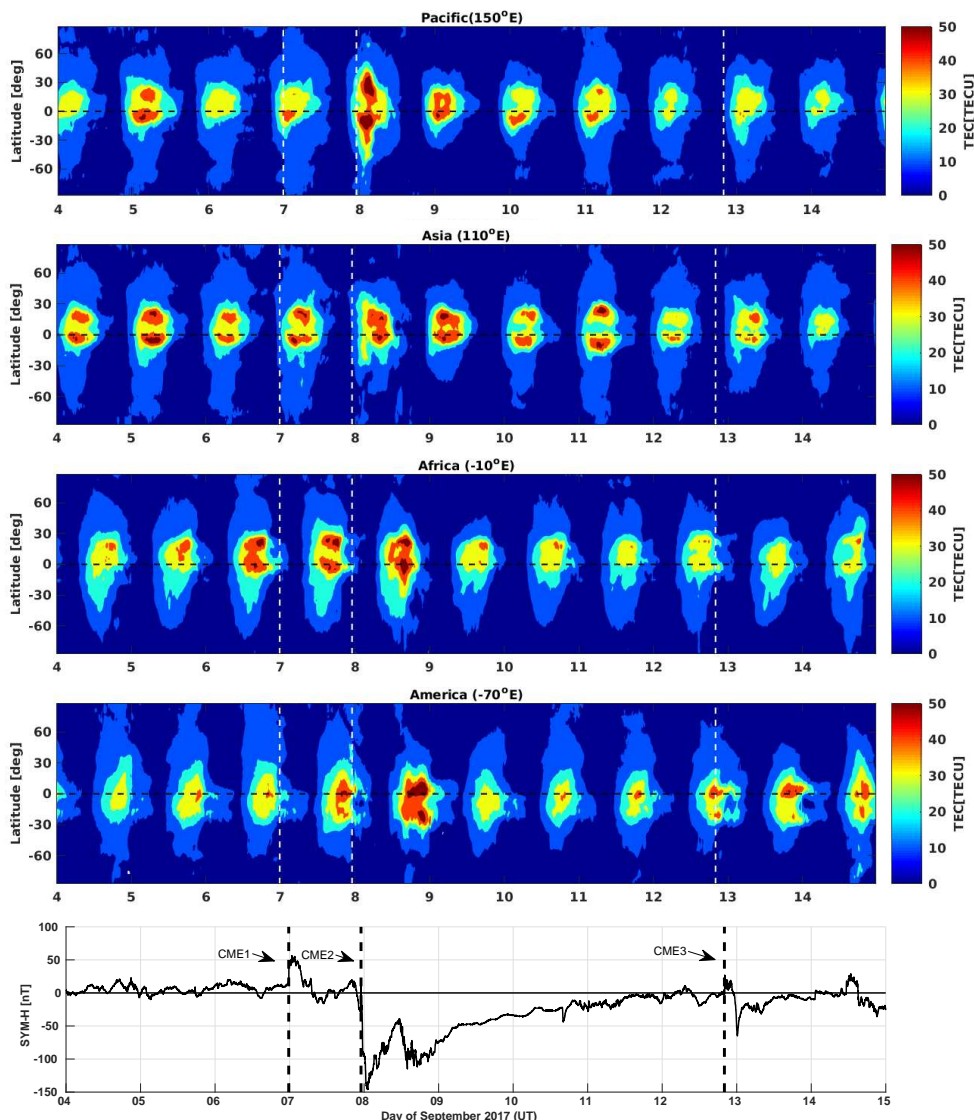

**Figure 4.** The vTEC variations over the Pacific (first plot), Asian (second plot), African (third plot), American (fourth plot) sectors and the SYM-H index (bottom plot) during the geomagnetic storm of 4-14 September 2017.

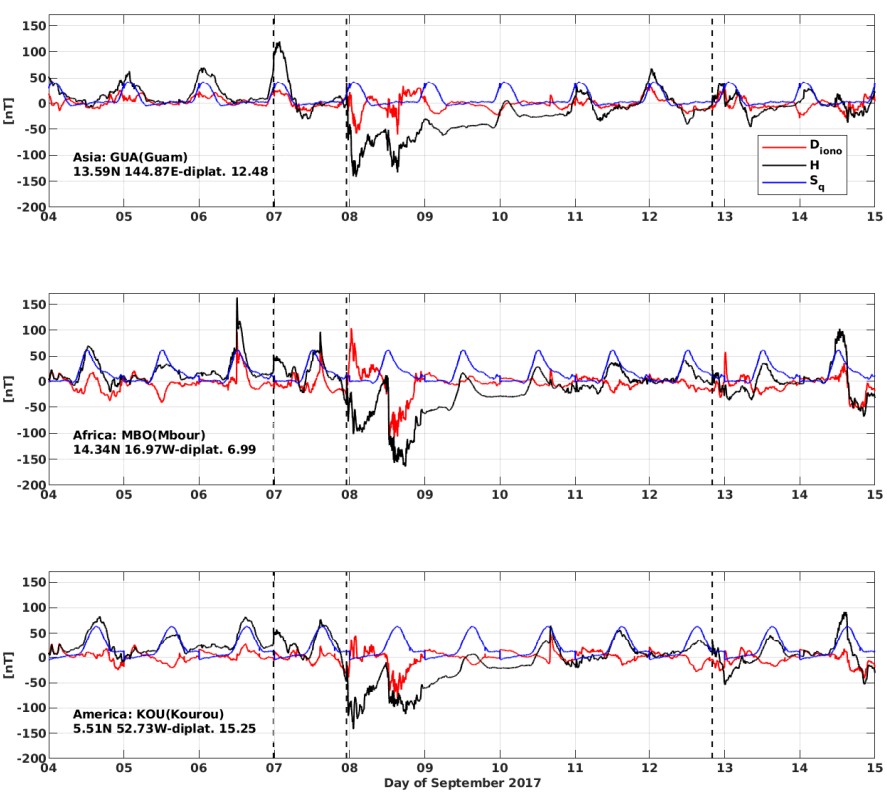

**Figure 5.** The magnetometer $H$ variations at specific stations during 4-14 September 2017 over the three sectors: the Asian (top), the African (middle) and the American (bottom). On each plot the quiet daily variations $S_q$ (in blue), the actual $H$ variations (in black) and the variations due to disturbed ionospheric currents $D_{iono}$ (in red) are plotted.

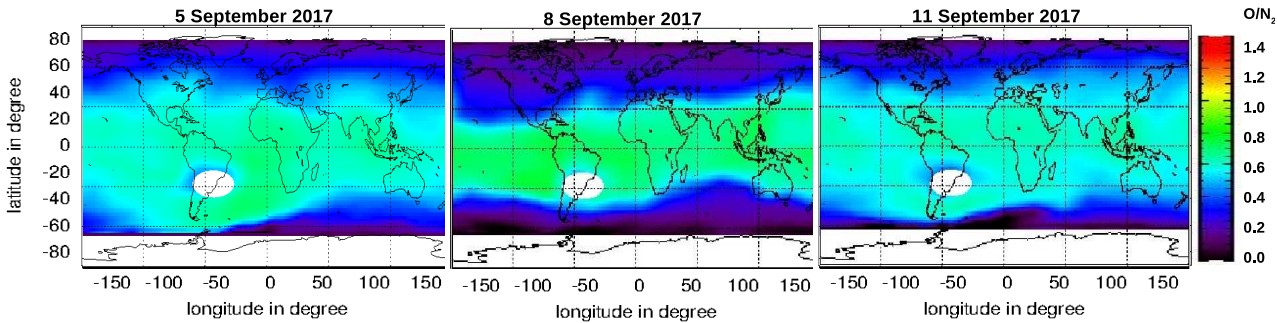

**Figure 6.** The thermospheric $O/N_2$ ratio obtained from the TIMED/GUVI instrument during the G4 category storm of 6-9 September 2017