# Peer review of "Response of low to mid-latitude ionosphere to the geomagnetic storm of September 2017"

_Annales Geophysicae, 2019_

## Referee Comment (RC1) · Anonymous Referee #1 · 19 Mar 2019

Dear Editor,

Please find the review comments in the attached Word document.

Best regards

Please also note the supplement to this comment:
https://www.ann-geophys-discuss.net/angeo-2019-19/angeo-2019-19-RC1-supplement.zip
* * *

---

## Referee Comment (RC2) · Anonymous Referee #2 · 22 Mar 2019

Dear Editor, Please find the review report in the two attached files: one is the report itself, the other consists of an annotated pdf.

Best regards

Please also note the supplement to this comment:
https://www.ann-geophys-discuss.net/angeo-2019-19/angeo-2019-19-RC2-supplement.zip
* * *

---

## Author Comment (AC1) · 3 May 2019

**Dear Editor,**

We are very thankful to the anonymous referee for the valuable suggestions to improve the manuscript. We have tried to answer the referee's queries.  Please note that the referee queries (blue color) and our response (black):

**General comments:**

In the revised version we have tried to explain our results on the basis of  studies published by Fuller-Rowell, Blanc and Richmond, Fejer and Balan.

**Specific comments:**

1. **Specify which exactly GIMs have been used in the study. IGS gives access to GIMs created by different providers (CODE, IGS, ESA, JPL etc.)?**
   **Answer: The vertical Total Electron Content (vTEC) is obtained from the IGS-GIM data which is available in the standard IONEX format on the NASA's website; i.e., Crustal Dynamics Data information                                        system (ftp://cddis.gsfc.nasa.gov/gps/products/ionex/).  These IONEX files contain the vTEC data for the entire globe and for any time, the vTEC data can be obtained from IONEX files at the time resolution of 2-h. The tomographic kriging (UQRG-GIMs) computed by the Technical University of Catalonia (UPC) at 15 minutes time resolution have been used to study the Global and Regional electron content.**

2. **Specify what was the calibration technique used to obtain TEC data for individual GPS stations?**
   **Answer: We didn't use any calibration technique in this study. However, we have extracted the vTEC data of a particular GPS station from the IGS-GIMs by using a MATLAB script. We have validated our methodology by reproducing the results of the previous studies particularly the analysis of the ST. Patrick day storm of MARCH 2015  published by Nava et al. 2016 and Kashcheyev et al. 2018.**

3. **Provide the criteria on how the quite days were selected and what were these quite days when computing the difference for REC and GEC and for vTEC from individual GPS stations.**

**Answer: We considered the three quiet days before the storm having Ap index below 22 nT. These quiet days are 2, 3, and 4 September.**

4. **Fluctuations in Asia REC show a regular pattern with a local maxima at the beginning of the day (e.g. 5, 7, 8, 9 and 11 September). It should be discussed/explained. A proper selection of the quite days might affect the results in this respect.**
   **Answer: It can be seen that during the period 4-14 September 2017, the REC varies significantly over the four longitudinal sectors. The observed behavior of the Δ REC can be attributed to the energy inputs from the solar wind to the magnetosphere Nava et al. (2016). The AE index which is an indicator of the energy transfer from the solar wind to the magnetosphere is shown in Figure 2. It can be noticed that the AE index shows several episodes of the energy inputs (having the maximum value of the AE index greater than 1000 nT) which occur on 4, 7, 8 and 13 September. In response to these energy inputs, the amplitude and the occurrence time of maxima/minima of the REC also vary.**

5. **To understand better the fluctuations of REC and their influence on GEC, please extend Figure 2 plot with the REC computed for the fourth sector.**
   **Answer: The fourth region that is the Pacific sector has been included in the revised version. Please see Figure 2.**

6. **The behavior of vTEC at BAKO station should be discussed/explained. 8 out of 9 days show a significant deviation from a quiet day behavior. Again, the proper selection of quite days might affect this result.**
   **Answer: The vTEC for all the stations have been revised with the proper selection of the quiet days.**

7. **For the sake of comparison purposes, please shrink the bottom panel of Figure 4 so that is has the same width as vTEC plots (without colorbar).**
   **Answer: The size of the bottom panel has been adjusted in Figure 4 of the revised manuscript.**

8. **Provide information on the technique how magnetic field disturbances (D) was computed. At the same time, how daily quite variation (Sq) was computed.**
   **Answer: The technique used to calculate the magnetic field disturbances has been explained in the revised version. Please see Section 2: Data sets (magnetometer data).**
   **In order to calculate the magnetic field variations we adopted the approach of Nava et al. (2016); Kashcheyev et al. (2018). The brief description of this approach is given here. During the geomagnetic storm, the horizontal component 'H' of the Earth's magnetic field can be expressed as:**
   **$H = H_o + D_M + D_{iono} + S_R{}^H$,**
   **where $H_o$ represents the magnetic field component due to the Earth's external core dynamics, $D_M$ is the disturbance which comes from the magnetospheric currents mainly due to Chapman Ferraro current, ring current and tail current Cole (1966). It can be calculated as:**
   **$D_M = $ SYM-H.$\cos\varphi$,**
   **here $\varphi$ is the geomagnetic latitude. The $S_R{}^H$ is the quiet daily regular variation of H and is computed by using the four quietest days having Kp < 2 such as:**
   **$S_R{}^H = \dfrac{1}{n} \sum \left(H_i + D_i^H\right) - H_o$ ,**
   **where n is the number of quiet days. The $D_i{}^H$ depicts the disturbances coming from the ionosphere $D_{iono}$ and the magnetosphere $D_M$. The magnetic disturbance due to ionospheric electric currents can be written as:**
   **$D_{iono} = \Delta H - S_q - $ SYM-H.$\cos\varphi$,**
   **here $S_q =< S_R{}^H>$ is the hourly amplitude of daily variations of the geomagnetic field.**

**Targeted comments:**

**P.1 L.7 "It is observed that the storm time response of the TEC over the pre-noon sector (Asia) is earlier than Africa and America." This sentence has to be modified, as it might be confusing when it comes to the time reference the authors refer to. It can be changes to e.g. "It is observed that the storm time response of the TEC over the pre-noon sector (Asia) is smaller than over Africa and America."**

**Answer: The storm time response of the TEC over the Asia/Pacific is earlier than over Africa/America. The observed positive ionospheric**

storm phase over the local dayside sectors is strongly linked with the occurrence time of the two SYM-H minima.

**P.1 L.14 "In the modern space era, the strength of the geomagnetic storm is characterized by the minimum Dst". It is not clear how space era is related to Dst index, as it is based on ground based measurements. Please change/rephrase the sentence.**

Answer: The sentence has been rephrased in the revised manuscript.

**P.4 L.7 Explain index "I" of vTEC ($I_{il}$) in the formula. Provide information on how vTEC value is computed for a cell, having 4 vTEC values (4 vertices of the rectangle).**

Answer: The GEC is obtained from the UPC-GIM data by the summation of the vTEC values in a cell $I_{i,j}$ multiplied by a cell's area $S_{i,j}$ over all GIM cells and it is given by Afraimovich et al. (2006),

$$GEC = \sum I_{i,j}.S_{i,j}$$

Here i and j represent the latitude and longitude of a certain GIM cell. The latitudinal and longitudinal extent of the elementary GIM cell is about $2.5°$ and $5°$, respectively.

**P.5 L.5-10 Please make sure you use the correct SYM-H data. According to OmniWeb, the minimum SYM-H was -146nT at 01:08 UT (not -148nT at 2 UT) and the second minimum was -115nT at 13:56 UT (not -122nT at 15 UT). Correct also the statement that two AE maxima of 2000nT and 2500nT coincide with the two peaks in SYM-H. It is not correct. As at the times of SYM-H minima, (01:08 UT and 13:56 UT), AE (according to OmniWeb) equal to 784nT and 1259nT correspondingly. At the same time, the two maxima around that time in AE were 2447nT at 23:43 UT (September 7th) and 2677nT at 14:06 UT (September 8th). Therefore, only the second peak might be considered as "coinciding", while the first one is definitely not.**

Answer: Answer: The case study is revised as:

In early September 2017 mainly three CMEs with earthward trajectories were emitted on 4, 6 and 10 September. A CME originating from the massive X9.3 solar flare of 6 September, reached the Earth

at 2300 UT on 7 September. The arrival of this CME caused a significant compression to the day side magnetosphere which provoked a severe geomagnetic storm having maximum value of the geomagnetic index Kp = 8. However, the arrival of the other two CMEs on 6 and 12 September lead to a minor geomagnetic storms of G1 category. Figure 1 illustrates the global morphology of these solar events.

In Figure 1, the storm time variations of the various plasma parameters are depicted in the following order (from top to bottom): the solar wind speed ($V_{sw}$), the Bz component of the IMF, the Interplanetary Electric Field (IEF), the AE index, the SYM-H index and the Solar radio flux F10.7. The three vertical lines represent the CMEs that lead to the Sudden Storm Commencement (SSC) at 2343, 2300 and 2002 UT on 6, 7 and 12 September 2017, as reported by: http://www.obsebre.es/php/geomagnetisme/vrapides/. However, the present study focus on the effects of the G4 category storm which occurs on 8 September 2017. On the arrival of the interplanetary shock on 7 September at about 2300 UT, the initial phase of the storm begins with a rapid variations in plasma parameters. During the main phase, the Bz component of the IMF is more southward reaching to the maximum lowest value of about −32 nT and then it rapidly increases to the value of approximately +16 nT. It again performs a negative excursion and reaches the value of approximately −16 nT. It can be seen that the SYM-H index also follows the behavior of the Bz component. During the main phase of the storm, the SYM-H also decreases and reaches to the negative value of ≃ −146 nT thus producing the first minima of the SYM-H index at 108 UT. From 108 UT until 1100 UT the Bz in northward; i.e., it increases to the positive value. Following the Bz,

the SYM-H index also increases from −146 nT to the value of −38 nT. During this partial recovery phase, the Bz becomes southward again by performing a negative excursion of −17.6 nT at 1155 UT and remains southward until 1356 UT, the SYM-H also reaches to its second minimum value of ≃ −115 nT. This is the end of the main phase of the storm which lasted for ~ 15 h. The main phase can be characterized by the occurrence of the two pronounced minima of the SYM-H with values −146 nT and −115 nT at 108 UT and 1356 UT respectively on 8 September 2017. The recovery phase started after 13:56 UT on 8 September. During the recovery phase, the SYM-H increases slowly and returned to its normal value at 1400 UT on 11 September. The recovery phase lasted for about 3 days. On September 8, the Vsw also exhibits an abrupt change by attaining a maximum value of about 840 km/s around 2 UT and after 12 UT it gradually decreases. The IEF is the Ey component of the electric field which is calculated as $E = -V_{sw} \times B$. It depends on the Bz component of the IMF and the x component of the $V_{sw}$. It means that the positive northward IMF leads to the westward IEF on the day side and eastward field on the night side. It can be seen that the IEF fluctuation occurs between −15 and +20 mV/m during this storm. The next two plots represent the AE and Kp indices. After the arrival of CME1, there is an increase in the auroral activity such that the AE index reaches to the peak value of about 1430 nT on 7 September at 09:07 UT. However, the occurrence of the two strong peaks exceeding 2000 nT in the AE index indicates that the most intense auroral activity occurred after the arrival of CME2. The  Kp index shows two episodes of the maximum value of approximately Kp = +8 for 3 h between 0-3 UT and 12-15 UT on 8 September. The bottom plot illustrates variation in the solar radio flux F10.7. It can be seen that the solar flux fluctuates

**significantly during the period 4-14 September 2017.**

**P.6 L.10 It would be beneficial to present the vTEC subplot from the fourth sector too.**

Answer: The VTEC plot for the Pacific sector has been included in Figure 4 of the revised manuscript.

**P.7 L.3 Please explain why disturbed part of H component of the magnetic field has index 'l', is it supposed to be 'i'?**

Answer: There was a typo and the correct index is "i".

**P.7 L.3 Indicate the time of the beginning of the initial phase of the storm.**

Answer: The initial phase of the storm begins at 2300 UT when CME2 hits the Earth's magnetosphere.

**P.7 L.5 Please specify the time of the beginning and the end of the main phase of the storm.**

Answer: The main phase begins at 2300 UT on September 8, 2017 and it ends up with the occurrence of the second SYM-H minima at 1356 UT on the same day. After that recovery phase started which lasted for about 3 days when SYM-H returns to the quiet conditions.

**P.7 L.15 It should be indicated that at the same time as an enhancement in O/N2 ratio is observed at low to equatorial latitudes, it decreases significantly at mid to high latitudes.**

The change has been incorporated in the revised version.

**P.8 L.2 (also on P.6 L.17) "*Most of the TEC is confined to the equatorial and low latitude regions.*" Please clarify what exactly part of the TEC is discussed in this sentence. It is a regular ionospheric feature to have most of the TEC in low and equatorial latitude regions.**
Answer: During the storm period, an enhancement in the vTEC along with the latitudinal extension of the EIA is observed. These two features have been explained in the revised version.

**P.8 L.7 It is difficult to understand why the authors acknowledge some of the data centers, but not the others in the acknowledgment**

**section. Please do acknowledge all data centers or only those that are not already mentioned in the 'Data Sets' section.**
Answer: We have acknowledged all the data centers.

**Typo/language comments:** The corrections regarding Typo/language have been corrected according to the suggestions mentioned below.

**Change "SYM – H" to "SYM-H" throughout the manuscript.**

Answer: 'SYM – H' has been replaced by 'SYM-H' throughout the manuscript.

**Please specify the titles of the papers in the 'References' section. It might be challenging for the reviewer to find the corresponding paper based on the information given.**

Answer: Titles of the papers has been included in the revised manuscript.

**P.1 L.20 Change "to investigate" to "to investigation of"**

Answer: "to investigate" is replaced by "to investigation of" in the revised manuscript.

**P.2. L.4 Change "is seasonal dependent" to ""is season dependent""**

Answer: "is seasonal dependent" is replaced by ""is season dependent" in the revised manuscript.

**P.2 L.7 Introduce abbreviation for the term "International Ground Station"**

Answer: The abbreviation  for "International Ground Station" has been introduced in the revised manuscript.

**P.2 L.9 Change "inter planetary" to "interplanetary"**

Answer:"inter planetary" has been changed to "interplanetary"  in the revised manuscript.

**P.2. L.10 Change "The author" to "The authors"**

Answer: "The author" has been replaced by "The authors" in the revised manuscript.

**P.2 L.16 Change "clear evidence of a" to "clear evidences of"**

Answer: "clear evidence of a" has been changed to "clear evidences of" in the revised manuscript.

**P.2 L.20 Change "Momani" to "Momani (2012)".**

Answer:"Momani" has been changed to "Momani (2012)" in the revised manuscript.

**P.2 L.27 Remove "Momani (2012)"**

Answer: Momani (2012) has been removed.

**P.2 L.30 Introduce abbreviation for the terms "Global and Regional electron content"**

Answer: The abbreviations GEC and REC for the Global and Regional Electron Content have been introduced.

**P.2 L.33 Change "and magneto-meter data" to "of the magnetometer data"**

Answer: In the revised manuscript "and magneto-meter data" has been to "of the magnetometer data".

**P.2 L.34 Change "disturbance Dynamo" to "disturbance dynamo"**

Answer: In the revised manuscript "disturbance Dynamo" has been changed to "disturbance dynamo".

**P.2 L.34 Change "Crustal movement network of China" to "Crustal Movement Observation Network of China"**

Answer: In the revised manuscript "Crustal movement network of China" has been changed to "Crustal Movement Observation Network of China"

**P.2 L.35 Change "distort" to "distorts" or "distorted"**

Answer: In the revised manuscript "distort" has been replaced by "distorts".

**P.3 L.3 Change "<<" to "<", correct subscript "st" to have "Dst"**

Answer: In the revised manuscript "<<" has been replaced by "<" and subscript "st" has been removed to have "Dst".

**P.3 L.5 Define abbreviation "CME" for "coronal mass ejections"**

Answer: The abbreviation "CME" for "coronal mass ejection" has been introduce.

**P.3 L.14 Change "multi instruments" to "multi-instrument"**

**Answer: In the revised manuscript "multi instruments" has been changed to "multi-instrument".**

**P.3 L.16 Change "pulsation" to "pulsations"**

**Answer: In the revised manuscript "pulsation" has been replaced by "pulsations".**

**P.3 L.18 Start sentence "This paper…" from a new line**

**Answer: In the revised manuscript "This paper…" is from a new line.**

**P.3 L.18 Change "This paper" to "Present work"**

**Answer:"This paper" has been changed to "Present work".**

**P.3 L.23 Change "Section 2 deals with the" to "Section 2 presents a"**

**Answer: In the revised manuscript "Section 2 deals with the" has been changed to "Section 2 presents a".**

**P.3 L.23 Change "data sets and the GPS" to "data sets, GPS"**

**Answer: In the revised manuscript "data sets and the GPS" has been changed to "data sets, GPS".**

**P.3 L.24 Change "describe" to "describes"**

**Answer: In the revised manuscript "describe" has been changed to "describes".**

**P.3 L.29 Remove "the coronal mass ejection" or "CME", as it was already defined earlier.**

**Answer: We have removed "the coronal mass ejection" and used CME in the rest of the revised manuscript.**

**P.3 L.30 Remove "the high speed solar wind stream" or "HSSWS", as it was already defined earlier.**

**Answer: In the revised manuscript "the high speed solar wind stream" has been removed.**

**P.4 L.2 Remove "the interplanetary magnetic field" or "IMF", as it was already defined earlier.**

**Answer: In the revised manuscript "the interplanetary magnetic field" has been removed.**

**P.4 L.5 Change "energy transferred from storm" to "energy transfer from the solar wind"**

Answer: In the revised manuscript "energy transferred from storm" has been changed to "energy transfer from the solar wind".

**P.4 L.7 Change "450Km" to "450 km"**

Answer: In the revised manuscript "450Km" has been changed to "450 km".

**P.4. L.13 Change "summation is restricted" to "summation being restricted"**

Answer: In the revised manuscript "summation is restricted" has been replaced to "summation being restricted".

**P.4 L.17 Please explain term "quasi-definite data" or rephrase.**

Answer: In the revised manuscript "quasi-definite data" has been rephrased as "quasi-definitive data".

**P.4 L.18 Change "inter-magnet.com" to "http://intermagnet.org"**

Answer: In the revised manuscript "inter-magnet.com" has been changed to "http://intermagnet.org".

**P.4 L.18 Change "shows the geographic location" to "shows geographic locations"**

Answer: In the revised manuscript "shows the geographic location" has been changed to "shows geographic locations".

**P.4 L.23 Remove "Coronal Mass Ejection"**

Answer: In the revised manuscript "Coronal Mass Ejection" has been removed.

**P.4 L.24 Change "solar flares" to "solar flare"**

Answer: In the revised manuscript "solar flares" has been changed to "solar flare".

**P.4 L.24 Change "reached on the Earth" to "reached the Earth"**

Answer: In the revised manuscript "reached on the Earth" has been changed to "reached the Earth".

**P.4 L.25 Change "CMEs" to "CME"**

Answer: In the revised manuscript "CMEs" has been changed to "CME".

**P.5 L.3 Change "rapid variation" to "rapid variations"**

**Answer: In the revised manuscript "rapid variation" has been changed to "rapid variations".**

**P.5 L.4 Remove "to the value" (from "reaching to the value") or change to "reaching the value of"**

**Answer: "to" has been removed.**

**P.5 L.5 Remove "to" (from "reach to the value")**

**Answer:"to" has been removed.**

**P.5 L.6 Remove "its" from "to the its"**

**Answer:"its" has been removed.**

**P.5 L.8 Remove "to" from "reaching to the value"**

**Answer:"to" has been removed.**

**P.5 L.5-10 Please harmonize "≤" signs in the text. Minimum value of -148nT is not "≤" to -150nT, also -122nT is not "≤" than 130nT. Should it be an "approximately" sign instead?**

**Answer: "≤" signs has been removed.**

**P.5 L.15-16 Remove "GEC", "REC" and "vTEC" as they were defined above.**

**Answer: In the revised manuscript the abbreviations GEC, REC and vTEC have been used.**

**P.5 L.17 Remove "Regional Electron Content" and "Global Electron Content", leave "REC" and "GEC"**

**Answer:"Regional Electron Content" and "Global Electron Content" have been removed.**

**P.5 L.19 Change "by using the five quiet days before the storm" to "by subtracting the quite time variation from the value itself. The quite time variation is computed using five quiet days before the storm…". Explain what the criteria to select quite days were and what days they were.**

**Answer: The following modification has been done in the revised manuscript:**

**Both the Δ REC and  ΔGEC are calculated by subtracting the quiet time variation from the value itself. The quiet time variation is computed using the three quiet day before the storm having the Ap**

index below 22 nT. The quiet days considered are 2, 3 and 4 September 2017.

**P.5 L.24 Change "the individual station" to "individual stations"**

Answer:"the individual station" has been changed to "individual stations".

**P.5 L.25 Change "the panels first to third represent" to "the first to third panels represent"**

Answer: In the revised manuscript we have used "the plots from one to three".

**P.6 L.4 Change "vTEC increases" to "vTEC increase"**

Answer: In the revised manuscript we have used "increase in the vTEC".

**P.6 L.14 Change "irregular pattern" to "irregular patterns" or to "an irregular pattern"**

Answer: "irregular pattern" has been changed to "irregular patterns" or to "an irregular pattern".

**P.6 L.19 Change "magnetometer variations" to "magnetometer data variations"**

Answer: In the revised manuscript we have used "magnetic field variations".

**P.6 L.20 Change "Asian" to "Asia"**

Answer:"Asian" has been replaced by "Asia".

**P.9 L.5 'Blagoveshchenskya' change to 'Blagoveshchensky'**

Answer:'Blagoveshchenskya' has been changed to 'Blagoveshchensky'

**P.13 "ionosphere disturbance current (blue)" change to "ionosphere disturbance current (red)"**

Answer:"ionosphere disturbance current (blue)" has been changed to "ionosphere disturbance current (red)".

---

## Author Comment (AC2) · 3 May 2019

**Reviewer # 2:**

**Dear Editor,**

**We are thankful to the anonymous referee for his valuable suggestions which would be helpful to improve the manuscript. Here is our reply (in black) to the referee's concerns (in blue):**

**General Comments:**

**In the revised version we have tried to explain our results on the basis of previous studies.**

**Specific comments:**

1. **Table 1 and 2 displays the geographic coordinates of GPS stations and of magnetic observatories however, since you are investigating the effects of a geomagnetic storm on the ionosphere, the position in the magnetic reference frame is much more relevant.**

   **Answer: In the revised version of the manuscript Table 1 and 2 also contain geographic and geomagnetic locations of the GPS stations and magnetometer observatories.**

2. **Figure 1 would be much more useful if all plots were all stacked up, instead of being separate. Moreover, Figure 1 seems to have been downloaded by the OMNI web-page, it is preferable for the authors to draw their own figures.**

Answer: All the plots in Figure 1 are revised according to the referee's suggestions.

3. It is not clear how vTEC has been evaluated. Please specify it.

Answer: The vertical Total Electron Content (vTEC) is extracted from the International GNSS Service (IGS) Global Ionosphere Map(GIM) data which is available in the standard IONEX format on the NASA's website; i.e., Crustal Dynamics Data information system(ftp://cddis.gsfc.nasa.gov/gps/products/ionex/).

These IONEX files contain the vTEC data for the entire globe. For any time, the vTEC data can be obtained from IONEX files at the time resolution of 2-h.

4. The description of the event investigated, given in the Case Study section, is very inaccurate and incorrect. Values of the peaks of SymH and AE are wrong, as well as their occurrence time. The time of the arrival at the Earth's surface of the effect of the CME is wrong, being the correct time 23:00 UT (see http://www.obsebre.es/php/geomagnetisme/vrapides/ssc_2017_d.txt). G-classes of geomagnetic storms are here mentioned but never explained or referenced. The sentence "with the value of geomagnetic index kp = 8 at 23:50UT. " makes no sense, being Kp an index estimated on intervals of 3 hours. Also the

sentence "The solar wind speed increased from 500 to 785km/s." makes no sense, the time interval when this happened being not specified. The timing of AE maxima does not coincide with that of Sym-H minima.

Answer: The case study is revised as:

In early September 2017 mainly three CMEs with earthward trajectories were emitted on 4, 6 and 10 September. A CME originating from the massive X9.3 solar flare of 6 September, reached the Earth at 23 : 00 UT on 7 September. The arrival of this CME caused a significant compression to the day side magnetosphere which provoked a severe geomagnetic storm having maximum value of the geomagnetic index Kp = 8. However, the arrival of the other two CMEs on 6 and 12 September lead to a minor geomagnetic storms of G1 category. Figure 1 illustrates the global morphology of these solar events.

In Figure 1, the storm time variations of the various plasma parameters are depicted in the following order (from top to bottom): the solar wind speed (Vsw), the Bz component of the IMF, the Interplanetary Electric Field (IEF), the AE index, the SYM-H index and the Solar radio flux F10.7. The three vertical lines represent the CMEs that lead to the Sudden Storm Commencement (SSC) at 23:43, 23:00 and 20:02 UT on 6, 7 and 12 September 2017, as reported by:

http://www.obsebre.es/php/geomagnetisme/vrap

[ides/](ides/). However, the present study focus on the effects of the G4 category storm which occurs on 8 September 2017. On the arrival of the interplanetary shock on 7 September at about 23: 00 UT, the initial phase of the storm begins with a rapid variations in plasma parameters. During the main phase, the Bz component of the IMF is more southward reaching to the maximum lowest value of about −32 nT and then it rapidly increases to the value of approximately +16 nT. It again performs a negative excursion and reaches the value of approximately −16 nT. It can be seen that the SYM-H index also follows the behavior of the Bz component. During the main phase of the storm, the SYM-H also decreases and reaches to the negative value of ≃ −146 nT thus producing the first minima of the SYM-H index at 1: 08 UT. From 1: 08 UT until 11: 00 UT the Bz in northward; i.e., it increases to the positive value. Following the Bz, the SYM-H index also increases from −146 nT to the value of −38 nT. During this partial recovery phase, the Bz becomes southward again by performing a negative excursion of −17.6 nT at 11:55 UT and remains southward until 13 : 56 UT, the SYM-H also reaches to its second minimum value of ≃ −115 nT. This is the end of the main phase of the storm which lasted for ~ 15 h. The main phase can be characterized by the occurrence of the two pronounced minima of the SYM-H with values −146 nT and −115 nT at 1: 08 UT and 13: 56 UT respectively

on 8 September 2017. The recovery phase started after 13: 56 UT on 8 September. During the recovery phase, the SYM-H increases slowly and returned to its normal value at 14:00 UT on 11 September. The recovery phase lasted for about 3 days. On September 8, the Vsw also exhibits an abrupt change by attaining a maximum value of about 840 km/s around 2UT and after 12 UT it gradually decreases. The IEF is the Ey component of the electric field which is calculated as E = -Vsw × B. It depends on the Bz component of the IMF and the x component of the Vsw. It means that the positive northward IMF leads to the westward IEF on the day side and eastward field on the night side. It can be seen that the IEF fluctuation occurs between −15 and +20 mV/m during this storm. The next two plots represent the AE and Kp indices. After the arrival of CME1, there is an increase in the auroral activity such that the AE index reaches to the peak value of about 1430 nT on 7 September at 09:07 UT. However, the occurrence of the two strong peaks exceeding 2000 nT in the AE index indicates that the most intense auroral activity occurred after the arrival of CME2. The Kp index shows two episodes of the maximum value of approximately Kp = +8 for 3 h between 0-3 UT and 12-15 UT on 8 September. The bottom plot illustrates variation in the solar radio flux F10.7. It can be seen that the solar flux

fluctuates significantly during the period 4-14 September 2017.

5. Data (as well as figures, see above) from the OMNI website are used, but the acknowledgment OMNI is completely missing.
Answer: We have acknowledged the OMNI data base:
https://omniweb.gsfc.nasa.gov/form/dx1.html.

6. Figures 3 and 4 are missing the labels on the horizontal axes.
Answer: In Figures 3 and 4 the horizontal axes are now labeled.

7. Concerning the description of Figure 3: 1) the increase of TEC on the day of the storm is visible only in BJFS, not in YAR2; 2) in Africa the enhancement during the storm is clearly visible also in Wind (why do you say that is less significant?).
Answer: In the revised manuscript following description has been added:
On the day of the storm, the northern and southern mid-latitude stations (BJFS and YAR2) in the Asian sector show an increase in the vTEC. However, in the equatorial station (BAKO) relatively less increase in the vTEC is observed. In the African region, the largest increase in the vTEC is observed for the equatorial and southern mid-latitude stations (NKLG and WIND) during the storm. However, a small increase in

the vTEC can be seen in the northern mid-latitude station (NOTI) in this sector.

8. Concerning Figure 4. It is not explained how maps covering the latitudinal range from -60° to 60°have been obtained.

Answer: The four plots in Figure 4 represent the vTEC over Asia, Africa, America and Pacific regions which are extracted from the IGS-GIM data (available on ftp://cddis.gsfc.nasa.gov/gps/products/ionex/). These IONEX files contain the vTEC data for the entire globe. Therefore, for a fixed longitude a contour plot covering the latitudinal range of -90° to 90° is made by using MATLAB script. These longitudes are given as: 110° E for Asia, -10°E for Africa, -70°E for America and 150°E for pacific.

9. Concerning Figure 4. It would be very helpful in the interpretation of this figure to have the SymH plot aligned and with the same size of those above.
Answer: The SYM-H plot is re-sized and aligned with the other plots in Figure 4.

10. Concerning the description of Figure 4: 1) in the Asian sector a pattern similar (in shape and values) to that observed on the 8 th of September is observed also on the day after the storm; 2) in the African sector a pattern

**similar to that observed on the 8th of September is observed also on the two days preceding the storm. How do you explain these features?**

Answer: The space weather conditions during 4-14 September are highly disturbed due to multiple CMEs and HSSWS.

- All the four longitudinal sectors show an enhancement in the vTEC on 6 September. This behavior can be associated with the impact of the CME1 which arrived at 23:43 UT on 6 September.
- During the initial phase of the storm on 7 September, it can be seen that the vTEC enhancements mainly occurred in the  crest regions of the EIA with a clear latitudinal separation.
- On the day of the storm that is 8 September,  a strong enhancement in the vTEC can be observed clearly in the crest regions of the EIA and in the equatorial regions over the four longitudinal sectors. Also the latitudinal extent of the EIA also increased up to the mid latitudes.
- In the Asian sector, the regular behavior of the vTEC that is having well defined crests can be observed except on the day of the storm. On September 9 that is during the recovery phase, the vTEC return back to its normal pattern with well defined crests.
- In the African/Pacific sector, the vTEC exhibits an irregular behavior; i.e., sometimes one and sometimes two crests of the EIA appear.
- In the American sector, we mostly observed one crest of the EIA and a very strong ionization on the day of the storm which return to its normal level after the storm on 9 September.

- The enhancement in the vTEC also observed on 5 and 11 September which can be due to the HSSWS effect. Moreover, the solar radio flux $F_{10.7}$ varies greatly during this period which can also affect the vTEC.

11. **Concerning the "interpretation" of Figure 5, this is just a mere description of what is the well-known and expected behavior of the geomagnetic field during a geomagnetic storm.**

Answer: The three plots in Figure 5 represent the magnetic field variations at the three equatorial magnetic observatories corresponding to the three longitudinal sectors of Asia (GUA), Africa (MBO) and America (KOU). Each plot shows the variation in the horizontal (H) component of the magnetic field (in black), the quiet daily variation (Sq) (in blue) and the disturbances (Diono) (in red). The following features of the H component can be noticed in all the three sectors:

– Firstly, an increase in the H component occurred during the initial phase of the storms. This enhancement is due to the Chapman-Ferraro current resulting from the contraction of the magnetosphere Chapman and Ferraro(1931).

– Secondly, a strong decrease in the H component can be observed during the main phase of the storms. It can be attributed to the equatorial ring current. The enhanced ring current in the magnetosphere induced the magnetic field opposite to the Earth's

northward dipole field which strongly reduces the H component.

- Following the strongest decrease in the H component, the recovery phase started which lasted for several hours. During the recovery phase, the ring current decays and the H component of the magnetic field returns back to the normal levels.

- Two pronounced dips in the H component at 1:08 UT and 13:56 UT on September 8 are observed in the three stations. It can be seen that the first minima is strongly negative for MBO as compared to GUA and KOU. However, the second dip is strongly negative for MBO as compared to GUA and KOU. This behavior is due to the local time variation of the ring current during the storm. Overall, the largest disturbance of the H component of the magnetic field with amplitude −180 nT is observed at MBO as compared to −150 nT at KOU and −140 nT at GUA.

The disturbance due to ionosphere electric current Diono which is the sum of the PPEF and the disturbance dynamo electric field (DDEF), is represented by the red curve in Figure 5. It follows anti-Sq signature during the storm period. It can be noted that during the first southward excursion of the magnetic field, the $D_{iono}$, decreases at the GUA which is the noon sector. However, an increasing trend in the $D_{iono}$ is observed for the MBO and KOU which are the night sector. During the second southward

excursion of the magnetic field, $D_{iono}$ decreases significantly for the MBO and KOU which are now on the day side.

**Targeted comments:**

**Page 1, lines 15-17: The classification of geomagnetic storms that is most widely accepted in the magnetospheric/geomagnetic community is that compiled by Gonzalez et al. (1994), so I suggest to refer to it in place of that by Loewe and Prolls (1997). Moreover, the citation of Tsurutani et al. (1992) at this point is not appropriate. I therefore suggest to cite Tsurutani et al. (1992) in place of Gonzalez et al. (1994) and vice versa. Of course, when citing the classification of Gonzalez et al. (1994) please check the thresholds of the Dst intervals and change the names of the different intensities of the geomagnetic storms.**

Answer: In the revised manuscript following modification is done:

On the basis of the Dst index and the Bz component of the IMF, the geomagnetic storms can be categorized as follows: weak or minor storms (Dst ≤ −30 nT, Bz ≤ −3 nT during 1 hour), moderate storms (Dst ≤ −50 nT, Bz ≤ −5 nT during 2 hours), intense storms (Dst ≤ −100 nT, Bz ≤ −10 nT for 3 hours) and severe storms (Dst ≤ −200 nT) (Gonzalez et al. (1994); Tsurutani et al. (1992); Loewe and Prolss (1997)). Some scientists have used the SYM-H geomagnetic index as a replacement of the Dst index due to advantage of its 1 min time resolution compared to the 1 h time resolution of the Dst index (Wanliss and Showalter (2006)). The 3 h value of the Kp index has also been used for the classification of the geomagnetic storms as: weak or minor storms (5− ≤ Kp ≤ 5), moderate storms(Kp ≥ 6), intense storms (7− ≤ Kp ≤ 7) and severe storms (Kp ≥ 8−) (Gosling et al. (1991)).

**Page 1, line 19:** Change "Therefore, the effects of geomagnetic storms are non uniform in different regions of the magnetosphere." Into "Therefore, geomagnetic storms produce effects that are different in the different regions of the magnetosphere".

Answer: The change is incorporated in the revised manuscript.

**Page 1, line 21:** Change "...observed which is almost two times higher than that of the quiet day value." Into "...observed, these have an amplitude that is almost twice that of a quiet day." Here the authors refer to "the quiet day". Are they referring to a specific quiet day or in general to "a quiet day"?

Answer: The change is incorporated in the revised manuscript.

**Page 2, line 1:** PPEF is generally used as the acronym of Prompt Penetration Electric Field and not Prompt Penetration Effects. Please correct the sentence.

Answer: The change is incorporated in the revised manuscript.

**Page 2, line 2:** Change "It is also found that the prompt penetration effect is almost uniform along the longitudinal direction." Into "It is also found that the effect of the prompt penetration electric field is almost uniform along the longitudinal direction."

Answer: The change is incorporated in the revised manuscript.

**Page 1, line 19:** "The ionosphere features vary along the latitudes and longitudes due to different current systems flowing in the magnetosphere." This sentence is too general and not completely correct. Better to say "During geomagnetic storms, the ionosphere features vary along the latitudes and longitudes also due to different current systems flowing in the magnetosphere."

Answer: The change is incorporated in the revised manuscript.

**Page 2, line 29: Please specify something about the "energy transfer", e.g. it occurs between …**

Answer: Following modification is done:

Many authors have analyzed the St. Patrick day storm (the largest geomagnetic storm of the Solar cycle 24) by using the GPS-TEC data analysis techniques to understand the positive and negative ionospheric-storm effects due to energy transfer between the solar wind and the magnetosphere.

**Page 2, line 32: I do not understand the logical sense of using "However" at this point.**

Answer: 'However' has been removed.

**Page 3, line 1: For the first time in the manuscript you mention here a "Northern equator anomaly". Which anomaly are you talking of? Please add something more.**

Answer: Following modification is done in the revised manuscript:

A rapid enhancement in the ionospheric electron density distorts the structure of the northern equatorial ionization anomaly region. It is also observed that during the main phase a significant decrease in the vTEC occurs at the high latitude as compared to the lower latitude region. Moreover, the height of the peak electron density in the F2 layer also increases during the geomagnetic storm.

**Page 4, lines 5-10: Please add a reference for Sym-H index and for AE index.**

Answer: In the revised manuscript the references for the SYM-H and the AE indices have been added.

**Page 4 line 15 Change "definite" into "definitive".**

Answer: "definite" has been replaced by "definitive".

**Page 5, lines 17-19: What differences are you talking of? Please specify. Correct, accordingly, also the caption of Figure 2.**

Answer: The following modification is done in the revised manuscript:

Figure 2 shows the Δ REC (top), the Δ GEC (middle) and the SYM-H index (bottom) during the period 4 -14 September, 2017. The Δ REC is calculated by taking the difference between the REC of each sector and the average daily values of the three quiet days before the storm having the Ap index below 22 nT. Similarly, the Δ GEC is the difference between the GEC and the average daily value of the three quiet days as considered in Δ REC.

**Page 5, line 19: What do you mean by "the five quiet days"? Maybe "the five quietest days"? In any case you have to specify, for these days, the level of geomagnetic activity by using some geomagnetic activity index (e.g., Dst, Kp...).**

Answer: The quiet time variations are computed by using the five quiet days before the storm having the Ap index below 22 nT.

**Page 5, lines 24-29: Change "panel" into "plots" everywhere in these lines. Panels are usually a composition of plots.**

Answer: Panels has been replaced by plots in the revised manuscript.

Answer: Correct order "quiet daily" has been used in the revised manuscript.

**Page 6, line 19: Change "magnetometer variations" into "magnetic field variations".**

Answer: In the revised manuscript "magnetometer variations" is replaced by "magnetic field variations".

**Answer: Correct order "quiet daily" has been used in the revised manuscript.**

**Page 7, line 1: Specify how the "disturbances" have been calculated.**

**Answer: In order to calculate the magnetic field variations we adopted the approach of Nava et al. (2016); Kashcheyev et al. (2018). The brief description of this approach is now added in the revised manuscript as:**

**During the geomagnetic storm, the horizontal component 'H' of the Earth's magnetic field can be expressed as:**

$$H = H_o + D_M + D_{iono} + S_R^H,$$

**where $H_o$ represents the magnetic field component due to the Earth's external core dynamics, $D_M$ is the disturbance which comes from the magnetospheric currents mainly due to Chapman Ferraro current, ring current and tail current Cole (1966). It can be calculated as:**

$$D_M = SYM\text{-}H.\cos\varphi,$$

**here $\varphi$ is the geomagnetic latitude. The $S_R^H$ is the quiet daily regular variation of H and is computed by using the four quietest days having Kp < 2 such as:**

$$S_R^H = \frac{1}{n} \sum \left( H_i + D_i^H \right) - H_o ,$$

**where n is the number of quiet days. The $D_i^H$ depicts the disturbances coming from the ionosphere $D_{iono}$ and the magnetosphere $D_M$. The magnetic disturbance due to ionospheric electric currents can be written as:**

$$D_{iono} = \Delta H - S_q - SYM\text{-}H.\cos\varphi,$$

**here $S_q = <S_R^H>$ is the hourly amplitude of daily variations of the geomagnetic field.**

**Page 11: Caption of Figure 3, indicate what the dashed line is for.**

**Answer: The three dashed lines correspond to the impact of the CMEs on 6, 7 and 12 September 2017.**

**Typo/language comments:**

Answer: In the revised manuscript the Typo and language mistakes have been removed according to the referee's suggestions.

Most Typo/language comments have been made directly on an annotated pdf. Below, additional

comments.

"Data" is commonly used as with a plural meaning, please change verbs accordingly throughout the manuscript.

Answer: In the revised manuscript the verb has been changed according to the referee's suggestions.

Add a space between the value and its unit (for instance, change 10nT into 10 nT) throughout the manuscript.

Answer: In the revised manuscript a space has been introduced between the value and the unit.

Change "Index" into "index" if not at the beginning of a sentence, throughout the manuscript.

Answer: In the revised manuscript 'index" has been changed according to the referee's suggestions.

When referring to mid latitudes you use both "mid" and "middle", choose one of the two terms and use it always.

Answer: In the revised manuscript the "mid" has been used to refer mid latitudes.

Concerning the use of acronyms. Two ways can be followed: 1) not to define them, 2) to define them but then to use them. For instance HSSWS is defined twice and never used.

Answer: In the revised manuscript the acronym HSSWS has been defined once and then used it.

References in the bibliography are formatted with different styles, please refer to the specific reference style of the journal.

Answer: In the revised manuscript the bibliography has been updated according to the Journal style.

---

## Author Comment (AC3) · 3 May 2019

General Comments:

In the revised version we have tried to explain our results on the basis of studies published by Fuller-Rowell,ÂăÂăBlanc and Richmond, Fejer andÂăBalan.

Specific comments: 1. Specify which exactly GIMs have been used in the study. IGS gives access to GIMs created by different providers (CODE, IGS, ESA, JPL etc.)? Answer: The vertical Total Electron Content (vTEC) is obtained from the IGS-GIM data which is available in the standard IONEX format on the NASA's website; i.e., Crustal Dynamics Data information system (ftp://cddis.gsfc.nasa.gov/gps/products/ionex/). These IONEX files contain the vTEC data for the entire globe and for any time, the

vTEC data can be obtained from IONEX files at the time resolution of 2-h. The tomographic kriging (UQRG-GIMs) computed by the Technical University of Catalonia (UPC) at 15 minutes time resolution have been used to study the Global and Regional electron content.

2. Specify what was the calibration technique used to obtain TEC data for individual GPS stations?

Answer: We didn't use any calibration technique in this study. However, we have extracted the vTEC data of a particular GPS station from the IGS-GIMs by using a MATLAB script. We have validated our methodology by reproducing the results of the previous studies particularly the analysis of the ST. Patrick day storm of MARCH 2015 published by Nava et al. 2016 and Kashcheyev et al. 2018.

3. Provide the criteria on how the quite days were selected and what were these quite days when computing the difference for REC and GEC and for vTEC from individual GPS stations.

Answer: We considered the three quiet days before the storm having Ap index below 22 nT. These quiet days are 2, 3, and 4 September.

4. Fluctuations in Asia REC show a regular pattern with a local maxima at the beginning of the day (e.g. 5, 7, 8, 9 and 11 September). It should be discussed/explained. A proper selection of the quite days might affect the results in this respect.

Answer: It can be seen that during the period 4-14 September 2017, the REC varies significantly over the four longitudinal sectors. The observed behavior of the $\Delta$ REC can be attributed to the energy inputs from the solar wind to the magnetosphere Nava et al. (2016). The AE index which is an indicator of the energy transfer from the solar wind to the magnetosphere is shown in Figure 2. It can be noticed that the AE index shows several episodes of the energy inputs (having the maximum value of the AE

index greater than 1000 nT) which occur on 4, 7, 8 and 13 September. In response to these energy inputs, the amplitude and the occurrence time of maxima/minima of the REC also vary.

5. To understand better the fluctuations of REC and their influence on GEC, please extend Figure 2 plot with the REC computed for the fourth sector.
Answer: The fourth region that is the Pacific sector has been included in the revised version. Please see Figure 2.

6. The behavior of vTEC at BAKO station should be discussed/explained. 8 out of 9 days show a significant deviation from a quiet day behavior. Again, the proper selection of quite days might affect this result.
Answer: The vTEC for all the stations have been revised with the proper selection of the quiet days.

7. For the sake of comparison purposes, please shrink the bottom panel of Figure 4 so that is has the same width as vTEC plots (without colorbar).
Answer: The size of the bottom panel has been adjusted in Figure 4 of the revised manuscript.

8. Provide information on the technique how magnetic field disturbances (D) was computed. At the same time, how daily quite variation (Sq) was computed.
In order to calculate the magnetic field variations we adopted the approach of Nava2016, Kashcheyev2018. The brief description of this approach is given here. During the geomagnetic storm, the horizontal component 'H' of the Earth's magnetic field can be expressed as:

$$H = H_o + D_M + D_{iono} + S_R^H,$$

where $H_o$ represents the magnetic field component due to Earth's external core dynamics, $D_M$ is the disturbance which comes from the magnetospheric currents mainly due to Chapman Ferraro current, ring current and tail current **?**. It can be calculated as:

$$D_M = SYM - H.cos\phi,$$

here $\phi$ is the geomagnetic latitude.
The $S_R^H$ is the quiet daily regular variation of H and is computed by using the four quietest days having $Kp < 2$ such as:

$$S_R^H = \frac{1}{n} \sum_{i=1}^{n} (H_i + D_i^H) - H_o,$$

where n is the number of quiet days. The $D_i^H$ depicts the disturbances coming from the ionosphere $D_{iono}$ and the magnetosphere $D_M$. The magnetic disturbance due to ionospheric electric currents can be written as:

$$D_{iono} = \Delta H - S_q - SYM - H.cos\phi,$$

here $S_q = < S_R^H >$ is the hourly amplitude of daily variations of the geomagnetic field.

Targeted comments:
P.1 L.7 "It is observed that the storm time response of the TEC over the pre-noon

sector (Asia) is earlier than Africa and America." This sentence has to be modified, as it might be confusing when it comes to the time reference the authors refer to. It can be changes to e.g. "It is observed that the storm time response of the TEC over the pre-noon sector (Asia) is smaller than over Africa and America."
Answer: The storm time response of the TEC over the Asia/Pacific is earlier than over Africa/America. The observed positive ionospheric storm phase over the local dayside sectors is strongly linked with the occurrence time of the two SYM-H minima.

P.1 L.14 "In the modern space era, the strength of the geomagnetic storm is characterized by the minimum Dst". It is not clear how space era is related to Dst index, as it is based on ground based measurements. Please change/rephrase the sentence.
Answer: The sentence has been rephrased in the revised manuscript.

P.4 L.7 Explain index "I" of vTEC (Ii) in the formula. Provide information on how vTEC value is computed for a cell, having 4 vTEC values (4 vertices of the rectangle).
Answer: The GEC is obtained from the UPC-GIM data by the summation of the vTEC values in a cell $I_{i,j}$ multiplied by a cell's area $S_{i,j}$ over all GIM cells and it is given by Afraimovich et al. (2006),

$$GEC = \sum_{i,j} I_{i,j}.S_{i,j}.$$

Here i and j represent the latitude and longitude of a certain GIM cell. The latitudinal and longitudinal extent of the elementary GIM cell is about 2.5◦ and 5◦, respectively.

P.5 L.5-10 Please make sure you use the correct SYM-H data. According to OmniWeb, the minimum SYM-H was -146nT at 01:08 UT (not -148nT at 2 UT) and the second

minimum was -115nT at 13:56 UT (not -122nT at 15 UT). Correct also the statement that two AE maxima of 2000nT and 2500nT coincide with the two peaks in SYM-H. It is not correct. As at the times of SYM-H minima, (01:08 UT and 13:56 UT), AE (according to OmniWeb) equal to 784nT and 1259nT correspondingly. At the same time, the two maxima around that time in AE were 2447nT at 23:43 UT (September 7th) and 2677nT at 14:06 UT (September 8th). Therefore, only the second peak might be considered as "coinciding", while the first one is definitely not.

Answer: The case study is revised as: In early September 2017 mainly three CMEs with earthward trajectories were emitted on 4, 6 and 10 September. A CME originating from the massive X9.3 solar flare of 6 September, reached the Earth at 2300 UT on 7 September. The arrival of this CME caused a significant compression to the day side magnetosphere which provoked a severe geomagnetic storm having maximum value of the geomagnetic index Kp = 8. However, the arrival of the other two CMEs on 6 and 12 September lead to a minor geomagnetic storms of G1 category. Figure 1 illustrates the global morphology of these solar events. In Figure 1, the storm time variations of the various plasma parameters are depicted in the following order (from top to bottom): the solar wind speed (Vsw), the Bz component of the IMF, the Interplanetary Electric Field (IEF), the AE index, the SYM-H index and the Solar radio flux F10.7. The three vertical lines represent the CMEs that lead to the Sudden Storm Commencement (SSC) at 2343, 2300 and 2002 UT on 6, 7 and 12 September 2017, as reported by: http://www.obsebre.es/php/geomagnetisme/vrapides/. However, the present study focus on the effects of the G4 category storm which occurs on 8 September 2017. On the arrival of the interplanetary shock on 7 September at about 2300 UT, the initial phase of the storm begins with a rapid variations in plasma parameters. During the main phase, the Bz component of the IMF is more southward reaching to the maximum lowest value of about $-32$ nT and then it rapidly increases to the value of approximately +16 nT. It again performs a negative excursion and reaches the value of approximately $-16$ nT. It can be seen that the SYM-H index also follows the behavior of the Bz component. During the main phase of the storm, the SYM-H also decreases

and reaches to the negative value of ≃ −146 nT thus producing the first minima of the SYM-H index at 108 UT. From 108 UT until 1100 UT the Bz in northward; i.e., it increases to the positive value. Following the Bz, the SYM-H index also increases from −146 nT to the value of −38 nT. During this partial recovery phase, the Bz becomes southward again by performing a negative excursion of −17.6 nT at 1155 UT and remains southward until 1356 UT, the SYM-H also reaches to its second minimum value of ≃ −115 nT. This is the end of the main phase of the storm which lasted for âĹij 15 h. The main phase can be characterized by the occurrence of the two pronounced minima of the SYM-H with values −146 nT and −115 nT at 108 UT and 1356 UT respectively on 8 September 2017. The recovery phase started after 13:56 UT on 8 September. During the recovery phase, the SYM-H increases slowly and returned to its normal value at 1400 UT on 11 September. The recovery phase lasted for about 3 days. On September 8, the Vsw also exhibits an abrupt change by attaining a maximum value of about 840 km/s around 2 UT and after 12 UT it gradually decreases. The IEF is the Ey component of the electric field which is calculated as E = -Vsw × B. It depends on the Bz component of the IMF and the x component of the Vsw. It means that the positive northward IMF leads to the westward IEF on the day side and eastward field on the night side. It can be seen that the IEF fluctuation occurs between −15 and +20 mV/m during this storm. The next two plots represent the AE and Kp indices. After the arrival of CME1, there is an increase in the auroral activity such that the AE index reaches to the peak value of about 1430 nT on 7 September at 09:07 UT. However, the occurrence of the two strong peaks exceeding 2000 nT in the AE index indicates that the most intense auroral activity occurred after the arrival of CME2. The Kp index shows two episodes of the maximum value of approximately Kp = +8 for 3 h between 0-3 UT and 12-15 UT on 8 September. The bottom plot illustrates variation in the solar radio flux F10.7. It can be seen that the solar flux fluctuates significantly during the period 4-14 September 2017.

P.6 L.10 It would be beneficial to present the vTEC subplot from the fourth sector too.

Answer: The VTEC plot for the Pacific sector has been included in Figure 4 of the revised manuscript.

P.7 L.3 Please explain why disturbed part of H component of the magnetic field has index 'l', is it supposed to be 'i'?
Answer: There was a typo and the correct index is "i".

P.7 L.3 Indicate the time of the beginning of the initial phase of the storm.
Answer: The initial phase of the storm begins at 2300 UT when CME2 hits the Earth's magnetosphere.

P.7 L.5 Please specify the time of the beginning and the end of the main phase of the storm.
Answer: The main phase begins at 2300 UT on September 8, 2017 and it ends up with the occurrence of the second SYM-H minima at 1356 UT on the same day. After that recovery phase started which lasted for about 3 days when SYM-H returns to the quiet conditions.

P.7 L.15 It should be indicated that at the same time as an enhancement in O/N2 ratio is observed at low to equatorial latitudes, it decreases significantly at mid to high latitudes.
Answer: The change has been incorporated in the revised version.

P.8 L.2 (also on P.6 L.17) "Most of the TEC is confined to the equatorial and low latitude regions." Please clarify what exactly part of the TEC is discussed in this sentence. It is a regular ionospheric feature to have most of the TEC in low and equatorial latitude regions. Answer: During the storm period, an enhancement in the vTEC along with the
latitudinal extension of the EIA is observed. These two features have been explained in the revised version.

P.8 L.7 It is difficult to understand why the authors acknowledge some of the data centers, but not the others in the acknowledgment section. Please do acknowledge all data centers or only those that are not already mentioned in the 'Data Sets' section. Answer: We have acknowledged all the data centers.

Typo/language comments: The corrections regarding Typo/language have been corrected according to the suggestions mentioned below.

Change "SYM – H" to "SYM-H" throughout the manuscript.
Answer: 'SYM – H' has been replaced by 'SYM-H' throughout the manuscript.

Please specify the titles of the papers in the 'References' section. It might be challenging for the reviewer to find the corresponding paper based on the information given.
Answer: Titles of the papers has been included in the revised manuscript.

P.1 L.20 Change "to investigate" to "to investigation of"
Answer: "to investigate" is replaced by "to investigation of" in the revised manuscript.

P.2. L.4 Change "is seasonal dependent" to ""is season dependent""
Answer: "is seasonal dependent" is replaced by ""is season dependent" in the revised manuscript.

P.2 L.7 Introduce abbreviation for the term "International Ground Station"
Answer: The abbreviation for "International Ground Station" has been introduced in

the revised manuscript.

P.2 L.9 Change "inter planetary" to "interplanetary"
Answer:"inter planetary" has been changed to "interplanetary" in the revised manuscript.

P.2. L.10 Change "The author" to "The authors"
Answer: "The author" has been replaced by "The authors" in the revised manuscript.

P.2 L.16 Change "clear evidence of a" to "clear evidences of"
Answer: "clear evidence of a" has been changed to "clear evidences of" in the revised manuscript.

P.2 L.20 Change "Momani" to "Momani (2012)".
Answer:"Momani" has been changed to "Momani (2012)" in the revised manuscript.

P.2 L.27 Remove "Momani (2012)"
Answer: Momani (2012) has been removed.

P.2 L.30 Introduce abbreviation for the terms "Global and Regional electron content"
Answer: The abbreviations GEC and REC have been introduced for the Global and Regional electron content.

P.2 L.33 Change "and magneto-meter data" to "of the magnetometer data"
Answer: In the revised manuscript "and magneto-meter data" has been to "of the magnetometer data".

P.2 L.34 Change "disturbance Dynamo" to "disturbance dynamo"
Answer: In the revised manuscript "disturbance Dynamo" has been changed to "disturbance dynamo".

P.2 L.34 Change "Crustal movement network of China" to "Crustal MovementÂăObservationÂăNetwork of China"
Answer: In the revised manuscript "Crustal movement network of China" has been changed to "Crustal MovementÂăObservationÂăNetwork of China"

P.2 L.35 Change "distort" to "distorts" or "distorted"
Answer: In the revised manuscript "distort" has been replaced by "distorts".

P.3 L.3 Change "«" to "<", correct subscript "st" to have "Dst"
Answer: In the revised manuscript "«" has been replaced by "<" and subscript "st" has been removed to have "Dst".

P.3 L.5 Define abbreviation "CME" for "coronal mass ejections"
Answer: The abbreviation "CME" for "coronal mass ejection" has been introduce.

P.3 L.14 Change "multi instruments" to "multi-instrument"
Answer: In the revised manuscript "multi instruments" has been changed to "multi-instrument".

P.3 L.16 Change "pulsation" to "pulsations"
Answer: In the revised manuscript "pulsation" has been replaced by "pulsations".

P.3 L.18 Start sentence "This paper…" from a new line
Answer: In the revised manuscript "This paper…" is from a new line.

P.3 L.18 Change "This paper" to "Present work"
Answer:"This paper" has been changed to "Present work".

P.3 L.23 Change "Section 2 deals with the" to "Section 2 presents a"
Answer: In the revised manuscript "Section 2 deals with the" has been changed to "Section 2 presents a".

P.3 L.23 Change "data sets and the GPS" to "data sets, GPS"
Answer: In the revised manuscript "data sets and the GPS" has been changed to "data sets, GPS".

P.3 L.24 Change "describe" to "describes"
Answer: In the revised manuscript "describe" has been changed to "describes".

P.3 L.29 Remove "the coronal mass ejection" or "CME", as it was already defined earlier.
Answer: We have removed "the coronal mass ejection" and used CME in the rest of the revised manuscript.

P.3 L.30 Remove "the high speed solar wind stream" or "HSSWS", as it was already defined earlier.
Answer: In the revised manuscript "the high speed solar wind stream" has been

removed.

P.4 L.2 Remove "the interplanetary magnetic field" or "IMF", as it was already defined earlier.
Answer: In the revised manuscript "the interplanetary magnetic field" has been removed.

P.4 L.5 Change "energy transferred from storm" to "energy transfer from the solar wind"
Answer: In the revised manuscript "energy transferred from storm" has been changed to "energy transfer from the solar wind".

P.4 L.7 Change "450Km" to "450 km"
Answer: In the revised manuscript "450Km" has been changed to "450 km".

P.4. L.13 Change "summation is restricted" to "summation being restricted"
Answer: In the revised manuscript "summation is restricted" has been replaced to "summation being restricted".
P.4 L.17 Please explain term "quasi-definite data" or rephrase.
Answer: In the revised manuscript "quasi-definite data" has been rephrased as "quasi-definitive data".

P.4 L.18 Change "inter-magnet.com" to "http://intermagnet.org"
Answer: In the revised manuscript "inter-magnet.com" has been changed to "http://intermagnet.org".

P.4 L.18 Change "shows the geographic location" to "shows geographic locations"

Answer: In the revised manuscript "shows the geographic location" has been changed to "shows geographic locations".

P.4 L.23 Remove "Coronal Mass Ejection"
Answer: In the revised manuscript "Coronal Mass Ejection" has been removed.

P.4 L.24 Change "solar flares" to "solar flare"
Answer: In the revised manuscript "solar flares" has been changed to "solar flare".

P.4 L.24 Change "reached on the Earth" to "reached the Earth"
Answer: In the revised manuscript "reached on the Earth" has been changed to "reached the Earth".

P.4 L.25 Change "CMEs" to "CME"
Answer: In the revised manuscript "CMEs" has been changed to "CME".

P.5 L.3 Change "rapid variation" to "rapid variations"
Answer: In the revised manuscript "rapid variation" has been changed to "rapid variations".

P.5 L.4 Remove "to the value" (from "reaching to the value") or change to "reaching the value of"
Answer: "to" has been removed.

P.5 L.5 Remove "to" (from "reach to the value")
Answer:"to" has been removed.

P.5 L.6 Remove "its" from "to the its"
Answer:"its" has been removed.

P.5 L.8 Remove "to" from "reaching to the value"
Answer:"to" has been removed.

P.5 L.5-10 Please harmonize "$\leq$" signs in the text. Minimum value of -148nT is not "$\leq$" to -150nT, also -122nT is not "$\leq$" than 130nT. Should it be an "approximately" sign instead?
Answer: "$\leq$" signs has been removed.

P.5 L.15-16 Remove "GEC", "REC" and "vTEC" as they were defined above.
Answer: In the revised manuscript the abbreviations GEC, REC and vTEC have been used.

P.5 L.17 Remove "Regional Electron Content" and "Global Electron Content", leave "REC" and "GEC"
Answer:"Regional Electron Content" and "Global Electron Content" have been removed.

P.5 L.19 Change "by using the five quiet days before the storm" to "by subtracting the quite time variation from the value itself. The quite time variation is computed using five quiet days before the storm...". Explain what the criteria to select quite days were and what days they were.
Answer: The following modification has been done in the revised manuscript:

Both the $\Delta$ REC and $\Delta$GEC are calculated by subtracting the quiet time variation from the value itself. The quiet time variation is computed using the three quiet day before the storm having the Ap index below 22 nT. The quiet days considered are 2, 3 and 4 September 2017.

P.5 L.24 Change "the individual station" to "individual stations"
Answer:"the individual station" has been changed to "individual stations".

P.5 L.25 Change "the panels first to third represent" to "the first to third panels represent"
Answer: In the revised manuscript we have used "the plots from one to three".

P.6 L.4 Change "vTEC increases" to "vTEC increase"
Answer: In the revised manuscript we have used "increase in the vTEC".

P.6 L.14 Change "irregular pattern" to "irregular patterns" or to "an irregular pattern"
Answer: "irregular pattern" has been changed to "irregular patterns" or to "an irregular pattern".

P.6 L.19 Change "magnetometer variations" to "magnetometer data variations"
Answer: In the revised manuscript we have used "magnetic field variations".

P.6 L.20 Change "Asian" to "Asia"
Answer:"Asian" has been replaced by "Asia".

P.9 L.5 'Blagoveshchenskya' change to 'Blagoveshchensky'

Answer:'Blagoveshchenskya' has been changed to 'Blagoveshchensky'

P.13 "ionosphere disturbance current (blue)" change to "ionosphere disturbance current (red)"
Answer:"ionosphere disturbance current (blue)" has been changed to "ionosphere disturbance current (red)".

Please also note the supplement to this comment:
https://www.ann-geophys-discuss.net/angeo-2019-19/angeo-2019-19-AC3-supplement.pdf

[Figure]

**Supplement:**

**Response of low to mid-latitude ionosphere to the Geomagnetic storm of September 2017**

Nadia Imtiaz[1], Waqar Younas[2], and Majid Khan[2]

[1]Theoretical Physics Division, PINSTECH, Nilore, Islamabad, Pakistan
[2]Department of Physics, Quaid-i-Azam University, Islamabad, Pakistan

**Correspondence:** Nadia Imtiaz (nhussain@ualberta.ca)

**Abstract.** We study the impact of **the** geomagnetic storm of 6-9 September 2017 on the low-to-mid latitude ionosphere. The prominent feature of this solar event is the sequential occurrence of the two SYM-H minima **with** values of $-146$ **nT and** $-115$ **nT** on 8 September at $1:08$ UT and $13:56$ UT, respectively. The study is based on **the analysis of** data from GPS stations and the **magnetic** observatories located at **the** different longitudinal sectors **corresponding to** Asia, Africa, America and Pacific. The GPS data **are** used to derive the global, regional and vertical total electron content in the four selected regions. **Magnetic observatory data are** used to illustrate the variation in the magnetic field**, particularly in its** horizontal component. It is observed that the storm time response of the vertical total electron content over the Asia/Pacific regions is earlier than **over** Africa and America. Overall, the positive ionospheric storm effects over the local day side sectors are associated with the ionospheric electric fields and the traveling atmospheric disturbances. The global thermospheric composition maps by Global Ultraviolet Imager exhibits a storm time variation in the $O/N_2$ ratio. The positive storm effects in the $O/N_2$ ratio occur in the low-latitudes and equatorial regions. It can be inferred that a variety of space weather phenomena such as the coronal mass ejection, the high speed solar wind stream and the solar radio flux can cause the multiple day enhancements of the vTEC in the low-to-mid latitude ionosphere during 4-14 September 2017.

**1 Introduction**

It is well known fact that the geomagnetic storm is a temporary variation of the Earth's magnetic field induced by the coronal mass ejection (CME) or the high speed solar wind stream (HSSWS). **The most widely used descriptors of the geomagnetic storms are: the disturbance storm time ($Dst$) index (which measures the ring current magnetic field), the SYM-H index (which reflects the variations in the intensity of the ring current), the Kp index, the auroral electrojet (AE) index (which measures the variations in the auroral electrojet), the Ap index and the $B_z$ component of the Interplanetary Magnetic Field (IMF) (Rostoker (1972); Gonzalez et al. (1994); Saba et al. (1997)). On the basis of the $Dst$ index and the $B_z$ component of the IMF, the geomagnetic storms can be categorized as follows: weak or minor storms ($Dst \leq -30$ nT, $B_z \leq -3$ nT during 1 hour), moderate storms ($Dst \leq -50$ nT, $B_z \leq -5$ nT during 2 hours), intense storms ($Dst \leq -100$ nT, $B_z \leq -10$ nT for 3 hours) and severe storms ($Dst \leq -200$ nT) (Gonzalez et al. (1994); Tsurutani et al. (1992); Loewe and Prolss (1997)). Some scientists have used the SYM-H geomagnetic index as a re-**

placement of the $Dst$ index due to advantage of its 1 min time resolution compared to the 1 h 
[revised manuscript text omitted]
, two solar flares; i.e., M7.3 and X1.3 were emitted at** $10:11$ **and** $14:20$ **UT, respectively. On 08 September 2017, M8.1 solar flare was fired off at** $15:35$ **UT. On September 12, X8.3 solar flare was emitted at** $15:35$ **UT. The associated earthward CMEs have induced the geomagnetic storms of different intensities in the early September 2017.**

2. Solar wind parameters: The OMNI database (https://omniweb.gsfc.nasa.gov/form/dx1.html) has been used to get solar wind parameters. The information about the $B_z$ component of the IMF and the solar wind speed ($V_{sw}$) is provided by the ACE satellite.

3. Geomagnetic indices: The world data center for Geomagnetism (Kyoto) provides information about different geomag- netic indices, **among them the AE index, the Ap index, the Kp index and SYM-H. The SYM-H is the proxy of the ring currents and the AE index estimates the energy transfer from the solar wind to the auroral ionospheric regions (Rostoker (1972); Wanliss and Showalter (2006)).**

4. Electron Content Data:**The vTEC is extracted from the International GNSS Service Global Ionosphere Map (IGS- GIM) data that are available in the standard IONEX format on the NASA's website; i.e., Crustal Dynamics Data information system (ftp://cddis.gsfc.nasa.gov/gps/products/ionex/). These IONEX files contain the vTEC data for the entire globe. For any time, the vTEC data can be obtained from IONEX files at the time resolution of 2-h.** The **tomographic kriging GIMs computed by the Technical University of Catalonia (UPC),** have been used to study the GEC variations during the storm period under consideration. The GEC is the total number of electrons present in the

ionosphere at the fixed altitude of about 450 **km**. **The GEC is obtained from the UPC-GIM data by the summation of the vTEC values in a cell** $I_{i,j}$ **multiplied by a cell's area** $S_{i,j}$ **over all GIM cells and it is given by Afraimovich et al. (2006),**

$$GEC = \sum_{i,j} I_{i,j}.S_{i,j}.$$

**Here** $i$ **and** $j$ **represent the longitude and latitude of a certain GIM cells. The latitudinal and longitudinal extent of the elementary GIM cell is about** $2.5°$ **and** $5°$**, respectively. The unit of GEC is** $1 GECU = 10^{32}$ **electrons.** The REC is the total number of electrons in the specified region of the ionosphere. Our analysis is based on the four different regions: Asia ($60° : 150°E$), Africa ($-30° : 60°E$), America ($-120° : -30°E$) and **Pacific (**$-180° : -120°E$**;** $150° : 180°E$). The REC is calculated **similarly to the** GEC, with the summation **being** restricted to the GIM cells of that particular region. **For both GEC/REC, the UPC-GIM data at time resolution of 15 min have been used.**

5. GPS stations: The data of nine GPS stations **are** analyzed here. These stations are selected on the basis of data availability and their geographic/geomagnetic location. The geographic **and geomagnetic locations** of these stations are given in Table 1.

6. Magnetometer Data: The storm time magnetic field variations are analyzed by using the data from the three low latitude observatories in three sectors: Asia (KOU), Africa (MBO) and America (GUA). The **quasi-definitive data** of these observatories which are available at **http://intermagnet.org** have been used for the analysis. Table 2 **shows geographic and geomagnetic locations** of these observatories.**In order to calculate the magnetic field variations we adopted the approach of Nava et al. (2016); Kashcheyev et al. (2018). The brief description of this approach is given here. During the geomagnetic storm, the horizontal component 'H' of the Earth's magnetic field can be expressed as:**

$$H = H_o + D_M + D_{iono} + S_R^H,$$

**where** $H_o$ **represents the magnetic field component due to Earth's external core dynamics,** $D_M$ **is the disturbance which comes from the magnetospheric currents mainly due to Chapman Ferraro current, ring current and tail current Cole (1966). It can be calculated as:**

$$D_M = \textbf{SYM-H}.cos\phi,$$

**here** $\phi$ **is the geomagnetic latitude. The** $S_R^H$ **is the quiet daily regular variation of H and is computed by using the four quietest days having** $Kp < 2$ **such as:**

$$S_R^H = \frac{1}{n}\sum_{i=1}^{n}(H_i + D_i^H) - H_o,$$

**where n is the number of quiet days. The** $D_i^H$ **depicts the disturbances coming from the ionosphere** $D_{iono}$ **and the magnetosphere** $D_M$**. The magnetic disturbance due to ionospheric electric currents can be written as:**

$$D_{iono} = \Delta H - S_q - \textbf{SYM-H}.cos\phi,$$

here $S_q = <S_R^H>$ is the hourly amplitude of daily variations of the geomagnetic field.

7. Thermospheric Composition: For the analysis of the storm time variation in the thermospheric neutral composition, the global maps of $O/N_2$ obtained from the GUVI/TIMED are presented here.

**3  Case Study**

In early September 2017 mainly three CMEs with the earthward trajectories were emitted on 4, 6 and 10 September. A CME originating from the massive $X9.3$ solar flare of 6 September, reached the Earth at $23:00$ UT on 7 September. The arrival of this CME caused a significant compression to the dayside magnetosphere which provoked a severe geomagnetic storm having 3 h maximum value of the geomagnetic index $Kp_{max} = 8$. However, the arrival of the other two CMEs on 6 and 12 September lead to a minor geomagnetic storm of G1 category. Figure 1 illustrates the global morphology of these solar events. In Figure 1, the storm time variations of the various interplanetary plasma and field parameters are depicted in the following order (from top to bottom): the solar wind speed ($V_{sw}$), the $B_z$ component of the IMF, the Interplanetary Electric Field (IEF), the AE index, the SYM-H index and the Solar radio flux $F_{10.7}$. The three vertical lines represent the CMEs which lead to the Sudden Storm Commencement (SSC) at $23:43$, $23:00$ and $20:02$ UT on 6, 7 and 12 September, respectively. As reported by http://www.obsebre.es/php/geomagnetisme/vrapides/. However, the present study focus on the effects of the G4 category storm which occurs on 8 September 2017. On the arrival of the interplanetary shock on 7 September at about $23:00$ UT, the initial phase of the storm begins with a rapid variations in the above mentioned parameters. During the main phase, the Bz component of the IMF is more southward reaching the maximum lowest value of about $-32$ nT and then it rapidly increases to the value of approximately $+16$ nT. It again performs a negative excursion and reaches the value of approximately $-16$ nT. It can be seen that the SYM-H index also follows the behavior of the $B_z$ component. During the main phase of the storm, the SYM-H index also decreases and reaches the negative value of $\simeq -146$ nT thus producing the first minima of the SYM-H index at $1:08$ UT. From $1:08$ UT until $11:00$ UT the $B_z$ in northward; i.e., it increases to the positive value. Following the $B_z$, the SYM-H also increases from $-146$ nT to the value of $-38$ nT. During this partial recovery phase the $B_z$ becomes southward again by performing a negative excursion of $-17.6$ nT at $11:55$ UT and remains southward until $13:56$ UT. At the same time the SYM-H index also reaches the second minimum value of $\simeq -115$ nT. This is the end of the main phase of the storm which lasted for $\sim 15$ h. The main phase can be characterized by the occurrence of the two pronounced minima of the SYM-H with values $-146$ nT and $-115$ nT at $1:08$ UT and $13:56$ UT respectively on 8 September 2017. The recovery phase started after $13:56$ UT on 8 September, the SYM-H increases slowly and returned to its normal value at $14:00$ UT on 11 September. The recovery phase lasted for about 3 days.

On September 8, the $V_{sw}$ also exhibits an abrupt change by attaining a maximum value of about $820$ km/s around $02:00$ UT and after $12:00$ UT it gradually decreases. The IEF is the $E_y$ component of the electric field which is calculated as $E = -V_{sw} \times B$. It depends on the $B_z$ component of the IMF and the x component of the $V_{sw}$. It means that the positive northward IMF leads to the westward IEF on the dayside and eastward field on the nightside. It can be seen

that the IEF fluctuation occurs between $-15$ and $+20$ **mV/m during this storm. The next two plots represent the AE and Kp indices. The AE index shows several peaks during this period. After the arrival of CME1, there is an increase in the auroral activity such that the AE index reaches the peak value of about** $1430$ **nT on 07 September at** $09:07$ **UT. However, the occurrence of the two strong peaks exceeding** $2000$ **nT in the AE index indicates that the most intense auroral activity occurred after the arrival of CME2. The Kp index shows the two episodes of the maximum value of approximately** $Kp = +8$ **for 3 h between** $0-3$ **UT and** $12-15$ **UT on 8 September. The bottom plot illustrates variation in the solar radio flux** $F_{10.7}$**. It can be seen that the solar flux fluctuates significantly during the period 4-14 September 2017.**

**4    Results/Discussion**

In this section, we present variations of the diverse parameters such as the REC, the GEC, the vTEC, the H component of the magnetic field and the $O/N_2$ ratio as a result of the geomagnetic storm of 7-9 September 2017. Figure 2 shows the $\Delta$ REC (top), the $\Delta$ GEC (middle) and the SYM-H index (bottom) during the period September 4-14, 2017. **Both the $\Delta$ REC and $\Delta$ GEC are calculated by subtracting the quiet time variation from the value itself. The quiet time variation is computed using the three quiet days before the storm having the Ap index below** $22$ **nT. The quiet days considered are 2, 3 and 4 September 2017.** It can be seen that the GEC shows two positive peaks at $1:08$ UT and $13:56$ UT corresponding to the first and second minima of the SYM-H index, respectively. In order to find the region which contributed to the peaks in the GEC, the REC is plotted for the four longitudinal sectors: Asia, Africa, America and Pacific. **It can be seen that during the period 4-14 September 2017, the REC varies significantly over the four longitudinal sectors. The observed behavior of the $\Delta$ REC can be attributed to the energy inputs from the solar wind to the magnetosphere Nava et al. (2016). The AE index which is an indicator of the energy transfer from the solar wind to the magnetosphere is shown in Figure 2. It can be noticed that the AE index shows several episodes of the energy inputs (having the maximum value of the AE index greater than 1000 nT) which occur on 4, 7, 8 and 13 September. In response to these energy inputs, the amplitude and the occurrence time of maxima/minima of the REC also vary.**

It is now clear that the first peak in the GEC is due to the Asian/Pacific sectors and the second peak is due to the African/American sectors. **It can be noticed that the $\Delta$ GEC plot also shows some other variation other than the the peaks. According to Afraimovich et al. (2008), there is a correlation between the GEC and the $F_{10.7}$ index. Therefore, the behavior of the GEC can also be affected by the higher solar flux; i.e.,** $F_{10.7} > 100$ **sfu.**

The nine plots of Figure 3 illustrate the variation of the vTEC for the **individual stations** of the three longitudinal sectors from 4-14 September 2017. In Figure 3, the **plots from one to three** represent the stations of the Asian sector; i.e., BJFS, BAKO and YAR2, **the plots from** four to six represent the African sector; i.e., NOTI, NKLG and WIND. The **plots from** seven to nine represent the stations of the American sector; i.e., BOGT, AREQ and ANTC. On each plot the vTEC is **displayed** in red with **quiet daily** variations in blue which are calculated **by subtracting the quiet time variations from the value itself. The**

**quiet time variations are computed by using the five quiet days before the storm having the Ap index below** $22$ **nT.** The following pertinent features of the vTEC can be noticed:

  – An enhancement in the vTEC is observed for all the stations in the three longitudinal sectors on the day of the storm. The three stations in the Asian sector exhibit an increase in the vTEC at the beginning of September 8. However, the stations in the African region show the increasing trend of the vTEC in the middle and American stations on late 8 September. The variability in the occurrence of the vTEC peaks depend on the local time of the SYM-H minima at these stations.

  – On the day of the storm, the northern and southern mid-latitude stations (BJFS and YAR2) in the Asian sector show an increase in the vTEC. However, in the equatorial station (BAKO) relatively less increase in the vTEC is observed.

  – In the African region, the largest increase in the vTEC is observed for the equatorial and southern mid-latitude stations (NKLG and WIND) during the storm. However, a small increase in the vTEC can be seen in the northern mid-latitude station (NOTI) in this sector.

  – In the American sector, the largest increase in the vTEC is observed for the equatorial station BOGT during the storm period. It can also be noticed that the vTEC decreases significantly for this station after the day of storm. Both the southern mid-latitude and equatorial trough stations ANTC and AREQ depict a multi-peak structures of the vTEC on the day of the storm. On the day after the storm the ionization disappears at the southern mid-latitudes and the vTEC returns to its quiet value.

Figure 4 illustrates the variation of vTEC as a function of time and latitudes over the four longitudinal sectors that are Asia (first plot), Africa (second plot), America (third plot) and Pacific (forth plot). **These vTEC maps are extracted from the IGS-GIM data which is available in the IONEX files for the entire globe. For a fixed longitude a contour plot covering the latitudinal range of** $-90°$ **to** $90°$ **is made by using the MATLAB script. The longitudes considered are given as:** $110°E$ **for Asia,** $-10°E$ **for Africa,** $-70°E$ **for America and** $150°E$ **for Pacific.** The four vTEC maps shown in Figure 4 cover the period from 4-14 September 2017 and the latitudinal range ($-90°$ to $+90°$). The SYM-H index over this period is also shown at the bottom in Figure 4. **The space weather conditions during this period are highly disturbed due to multiple events such as the CMEs and HSSWS. As a result the vTEC maps of the four longitudinal sectors show following features:**

  – **During geomagnetically quiet conditions, the** $E \times B$ **drift at the dip equator creates the Equatorial Ionization Anamoly (EIA) at** $\pm 10 - 15°$ **from the equator (Balan and Bailey (1995); Fejer (1991)). In response to the geomagnetic storm the latitudinal extent of the EIA is increased upto about** $30°$ **latitudes.**

  – **All the four sectors show an enhancement in the vTEC on 6 September. This behavior can be associated with the impact of the CME1 which arrived at 23:43 UT on 6 September.**

  – **During the initial phase of the storm on 7 September, the vTEC enhancement mainly occurred in the crest regions of the EIA with a clear latitudinal separation.**

- On the day of the storm, the vTEC strongly enhanced in the crests of the EIA and in the magnetically equatorial region as compared to the days before and after the storm. The enhancements of the vTEC in the EIA region in response to the geomagnetic storms have been reported in many studies (Zhao et al. (2005); Astafyeva et al. (2015); Lei et al. (2018)).

[revised manuscript text omitted]

– Overall, the largest disturbance of the $H$ component of the magnetic field with amplitude $-180$ **nT** is observed at MBO as compared to $-150$ **nT** at KOU and $-140$ **nT** at GUA.

The disturbance due to ionosphere electric current $D_{iono}$ is represented by the red curve in Figure 5. It follows anti-$S_q$ signature during the storm period. It can be noted that during the first southward excursion of the magnetic field, the $D_{iono}$, decreases at the GUA which is in the noon sector. However, an increasing trend in the $D_{iono}$ is observed for the MBO and KOU which are in the night sector. During the second southward excursion of the magnetic field, $D_{iono}$ decreases significantly for the MBO and KOU which are now in the dayside. The $D_{iono}$ contains signatures of the PPEF and DDEF therefore, the observed trend of the $D_{iono}$ can also be explained by these two electric fields. (?) Another effect that can bee seen during the storm is the variation in the the thermospheric neutral composition. The storm strengthens the vertical and meridional winds which leads to the variation in the thermospheric neutral profile. The global view of the thermospheric $O/N_2$ ratio obtained from the TIMED/GUVI for the period September 7-10, 2017 is shown in Figure 6. On the day of the storm, a significant enhancement in the $O/N_2$ ratio is observed at the low and equatorial latitudes. At the same time, the $O/N_2$ decreases significantly at mid to high latitudes as compared to the quiet time pattern. This observation is consistent with the behavior of the vTEC during the storm period. After the recovery of the storm, the thermospheric composition returns to its normal profile.

**5    Conclusions**

We presented the impact of geomagnetic storm of 7-9 September 2017 on the low-to-mid latitude ionosphere over the four longitudinal sectors; i.e., Asia, Africa, America and Pacific. The storm effects are characterized by using diverse parameters including the global, regional and vertical total electron content derived from the GPS data, the horizontal component of the magnetic field obtained from the magnetometers and the neutral composition from the GUVI/TIMED. It is observed that the positive storm effects occur in the local dayside stations. The temporal response of the four sectors shows that the positive storm effects in the REC and vTEC over the Asian/Pacific sectors are observed earlier than the American/African sectors. During geomagnetically quiet conditions, most of the TEC is confined to the equatorial and low latitude regions. However, the latitudinal extent of the TEC increases upto the mid-latitudes during the storm period. The storm time enhancement in the neutrals ratio; i.e., $O/N_2$ over the low latitudes and equator is consistent with the observed TEC behavior. Overall, the positive storm phase occur on the dayside sectors during the G4 geomagnetic storm of September 7-9, 2017. The vTEC enhancements observed on the other days are due to the high speed solar wind stream event. Analysis of the magnetometers data shows the largest disturbance of the horizontal component of the magnetic field at MBO as compared to that of KOU and GUA. The storm time variation of the horizontal component is associated with the Chapman-Ferraro and the ring currents. The magnetic field component associated with the disturbed ionospheric current follows the anti-Sq variations which depends on the prompt penetration electric field and the disturbance dynamo electric field. However, the negative storm effect in the $O/N_2$ ratio can be observed in the higher and mid-latitude regions. It can be concluded that the ionosphere dynamics and electrodynamics

play important role in the observed perturbations in the low-to-mid latitude ionosphere during 4-14 September 2017. The study would be useful for the understanding of storm time response of the low-mid latitude ionosphere.

*Acknowledgements.* The authors are grateful to the space weather data resources: the OMNI data base https://omniweb.gsfc.nasa.gov/form/dx1.html for providing the solar wind data, to the World Data Center for Geomagnetism at Kyoto University, Japan for providing the geomagnetic data: http://wdc.kugi.kyoto-u.ac.jp/dstrealtime/, to the International GNSS Service (IGS) team for providing the GPS data and to the https://ssusi.jhuapl.edu/. NI acknowledge the UNOOSA/ICTP for providing the financial support to attend GNSS workshop and learn these data analysis techniques. The authors are very grateful to the anonymous referees for their constructive and insightful comments in improving the manuscript. The authors would like to thanks Anton Kashcheyev and Christine Amory Mazaudier for their valuable suggestions. This research work was partly supported by the HEC Pakistan: $GrantNo.7632/Federal/NRPU/RD/HEC/2017$.

[Figure]

**Figure 1.** Global parameters: the $B_z$ component of the IMF, the $V_{sw}$, the AE index, the interplanetary electric field (IEF), the **SYM-H** index, the Kp index and $F_{10.7}$ characterizing the geomagnetic storm during September 4-12, 2017

[Figure]

**Figure 2.** Variation of the Regional electron content (top), the Global electron content (bottom) and the SYM-H index during the geomagnetic storm of 4-14 September 2017

[Figure]

**Figure 3.** The vTEC variations at GPS stations during the geomagnetic storm of 4-14 September 2017. Each plot illustrates the disturbed vTEC (in orange) and its quiet value (in blue).

[Figure]

**Figure 4.** The vTEC variations over the Asian (first plot), African (second plot), American (third plot), Pacific (fourth plot) sectors and the SYM-H index (bottom plot) during the geomagnetic storm of 4-14 September 2017

[Figure]

**Figure 5.** The magnetometer $H$ variations at specific stations during 4-14 September 2017 over the three sectors: the Asian (top), the African (middle) and the American (bottom). On each plot the **quiet daily** variations (blue), the actual $H$ variations (black) and the ionosphere disturbance current **(red)** are plotted

[Figure]

**Figure 6.** The thermospheric $O/N_2$ ratio obtained from the GUVI/TIMED during G4 category storm which occurred between 7-10 September 2017

---

## Author Comment (AC4) · 3 May 2019

General Comments:
In the revised version we have tried to explain our results on the basis of previous studies.
Specific comments:

1. Table 1 and 2 displays the geographic coordinates of GPS stations and of magnetic observatories however, since you are investigating the effects of a geomagnetic storm on the ionosphere, the position in the magnetic reference frame is much more relevant. Answer: In the revised version of the manuscript Table 1 and 2 also contain geographic

and geomagnetic locations of the GPS stations and observatories.

2. Figure 1 would be much more useful if all plots were all stacked up, instead of being separate. Moreover, Figure 1 seems to have been downloaded by the OMNI web-page, it is preferable for the authors to draw their own figures.
Answer: All the plots in Figure 1 are revised according to the referee's suggestions.

3. It is not clear how vTEC has been evaluated. Please specify it.
Answer: The vertical Total Electron Content (vTEC) is extracted from the International GNSS Service (IGS) Global Ionosphere Map(GIM) data which is available in the standard IONEX format on the NASA's website; i.e., Crustal Dynamics Data information system(ftp://cddis.gsfc.nasa.gov/gps/products/ionex/). These IONEX files contain the vTEC data for the entire globe. For any time, the vTEC data can be obtained from IONEX files at the time resolution of 2-h.

4. The description of the event investigated, given in the Case Study section, is very inaccurate and incorrect. Values of the peaks of SymH and AE are wrong, as well as their occurrence time. The time of the arrival at the Earth's surface of the effect of the CME is wrong, being the correct time 23:00 UT (see http://www.obsebre.es/php/geomagnetisme/vrapides/ssc_2017_d.txt). G-classes of geomagnetic storms are here mentioned but never explained or referenced. The sentence with the value of geomagnetic index kp = 8 at 23:50UT. makes no sense, being Kp an index estimated on intervals of 3 hours. Also the sentence 'The solar wind speed increased from 500 to 785km/s'. makes no sense, the time interval when this happened being not specified. The timing of AE maxima does not coincide with that of Sym-H minima.

Answer: The case study is revised as: In early September 2017 mainly three CMEs with earthward trajectories were emitted on 4, 6 and 10 September. A CME originating from the massive X9.3 solar flare of 6 September, reached the Earth at 23 : 00 UT on 7 September. The arrival of this CME caused a significant compression to the day side magnetosphere which provoked a severe geomagnetic storm having maximum value of the geomagnetic index Kp = 8. However, the arrival of the other two CMEs on 6 and 12 September lead to a minor geomagnetic storms of G1 category. Figure 1 illustrates the global morphology of these solar events. In Figure 1, the storm time variations of the various plasma parameters are depicted in the following order (from top to bottom): the solar wind speed (Vsw), the Bz component of the IMF, the Interplanetary Electric Field (IEF), the AE index, the SYM-H index and the Solar radio flux F10.7. The three vertical lines represent the CMEs that lead to the Sudden Storm Commencement (SSC) at 23:43, 23:00 and 20:02 UT on 6, 7 and 12 September 2017, as reported by: http://www.obsebre.es/php/geomagnetisme/vrapides/. However, the present study focus on the effects of the G4 category storm which occurs on 8 September 2017. On the arrival of the interplanetary shock on 7 September at about 23: 00 UT, the initial phase of the storm begins with a rapid variations in plasma parameters. During the main phase, the Bz component of the IMF is more southward reaching to the maximum lowest value of about $-32$ nT and then it rapidly increases to the value of approximately $+16$ nT. It again performs a negative excursion and reaches the value of approximately $-16$ nT. It can be seen that the SYM-H index also follows the behavior of the Bz component. During the main phase of the storm, the SYM-H also decreases and reaches to the negative value of about 146 nT thus producing the first minima of the SYM-H index at 1: 08 UT. From 1: 08 UT until 11: 00 UT the Bz in northward; i.e., it increases to the positive value. Following the Bz, the SYM-H index also increases from

$-$

146 nT to the value of

$-$

38 nT. During this partial recovery phase, the Bz becomes southward again by performing a negative excursion of −17.6 nT at 11:55 UT and remains southward until 13 : 56 UT, the SYM-H also reaches to its second minimum value of about 115 nT. This is the end of the main phase of the storm which lasted for around 15 h. The main phase can be characterized by the occurrence of the two pronounced minima of the SYM-H with values 146 nT and 115 nT at 1: 08 UT and 13: 56 UT respectively on 8 September 2017. The recovery phase started after 13: 56 UT on 8 September. During the recovery phase, the SYM-H increases slowly and returned to its normal value at 14:00 UT on 11 September. The recovery phase lasted for about 3 days. On September 8, the Vsw also exhibits an abrupt change by attaining a maximum value of about 840 km/s around 2UT and after 12 UT it gradually decreases. The IEF is the Ey component of the electric field which is calculated as E = -Vsw × B. It depends on the Bz component of the IMF and the x component of the Vsw. It means that the positive northward IMF leads to the westward IEF on the day side and eastward field on the night side. It can be seen that the IEF fluctuation occurs between −15 and +20 mV/m during this storm. The next two plots represent the AE and Kp indices. After the arrival of CME1, there is an increase in the auroral activity such that the AE index reaches to the peak value of about 1430 nT on 7 September at 09:07 UT. However, the occurrence of the two strong peaks exceeding 2000 nT in the AE index indicates that the most intense auroral activity occurred after the arrival of CME2. The Kp index shows two episodes of the maximum value of approximately Kp = +8 for 3 h between 0-3 UT and 12-15 UT on 8 September. The bottom plot illustrates variation in the solar radio flux F10.7. It can be seen that the solar flux fluctuates significantly during the period 4-14 September 2017.

5. Data (as well as figures, see above) from the OMNI website are used, but the acknowledgment OMNI is completely missing.
Answer: We have acknowledged the OMNI data base: OMNI data base https://omniweb.gsfc.nasa.gov/form/dx1.html.

6. Figures 3 and 4 are missing the labels on the horizontal axes.
Answer: In Figures 3 and 4 the horizontal axes are labeled.

7. Concerning the description of Figure 3: 1) the increase of TEC on the day of the storm is visible only in BJFS, not in YAR2; 2) in Africa the enhancement during the storm is clearly visible also in Wind (why do you say that is less significant?).
Answer: In the revised manuscript following description has been added: On the day of the storm, the northern and southern mid-latitude stations (BJFS and YAR2) in the Asian sector show an increase in the vTEC. However, in the equatorial station (BAKO) relatively less increase in the vTEC is observed. In the African region, the largest increase in the vTEC is observed for the equatorial and southern mid-latitude stations (NKLG and WIND) during the storm. However, a small increase in the vTEC can be seen in the northern mid-latitude station (NOTI) in this sector.

8. Concerning Figure 4. It is not explained how maps covering the latitudinal range from -60° to 60°have been obtained.

Answer: The four plots in Figure 4 represent the vTEC over Asia, Africa, America and Pacific regions which are extracted from the IGS-GIM data (available on ftp://cddis.gsfc.nasa.gov/gps/products/ionex/). These IONEX files contain the vTEC data for the entire globe. Therefore, for a fixed longitude a contour plot covering the latitudinal range of -90° to 90° is made by using MATLAB script. These longitudes are given as: 110° E for Asia, -10°E for Africa, -70°E for America and 150°E for pacific.

9. Concerning Figure 4. It would be very helpful in the interpretation of this figure to have the SymH plot aligned and with the same size of those above.
Answer: The SYM-H plot is re-sized and aligned with the other plots in Figure 4.

[Figure]

10. Concerning the description of Figure 4: 1) in the Asian sector a pattern similar (in shape and values) to that observed on the 8 th of September is observed also on the day after the storm; 2) in the African sector a pattern similar to that observed on the 8th of September is observed also on the two days preceding the storm. How do you explain these features?

Answer: The space weather conditions during 4-14 September are highly disturbed due to multiple CMEs and HSSWS.

All the four longitudinal sectors show an enhancement in the vTEC on 6 September. This behavior can be associated with the impact of the CME1 which arrived at 23:43 UT on 6 September. During the initial phase of the storm on 7 September, it can be seen that the vTEC enhancements mainly occurred in the crest regions of the EIA with a clear latitudinal separation. On the day of the storm that is 8 September, a strong enhancement in the vTEC can be observed clearly in the crest regions of the EIA and in the equatorial regions over the four longitudinal sectors. Also the latitudinal extent of the enhanced vTEC also increases up to the mid latitudes. In the Asian sector, the regular behavior of the vTEC that is having well defined crests can be observed except on the day of the storm. On September 9 that is during the recovery phase, the vTEC return back to its normal pattern with well defined crests. In the African/Pacific sector, the vTEC exhibits an irregular behavior; i.e., sometimes one and sometimes two crests of the vTEC appear. In the American sector, we mostly observed one crest of the vTEC and a very strong ionization on the day of the storm which return to its normal level after the storm on 9 September. Besides, the enhancement in the vTEC on September 5 and 11 can be due to the HSSWS effect. Moreover, the solar radio flux F10.7 varies greatly during this period which can also affect the vTEC.

11. Concerning the "interpretation" of Figure 5, this is just a mere description of what is the well-known and expected behavior of the geomagnetic field during a geomagnetic storm.

Answer: The three plots in Figure 5 represent the magnetic field variations at the three

equatorial magnetic observatories corresponding to the three longitudinal sectors of Asia (GUA), Africa (MBO) and America (KOU). Each plot shows the variation in the horizontal (H) component of the magnetic field (in black), the quiet daily variation (Sq) (in blue) and the disturbances (Diono) (in red). The following features of the H component can be noticed in all the three sectors:

– Firstly, an increase in the H component occurred during the initial phase of the storms. This enhancement is due to the Chapman-Ferraro current resulting from the contraction of the magnetosphere Chapman and Ferraro(1931).

– Secondly, a strong decrease in the H component can be observed during the main phase of the storms. It can be attributed to the equatorial ring current. The enhanced ring current in the magnetosphere induced the magnetic field opposite to the Earth's northward dipole field which strongly reduces the H component.

– Following the strongest decrease in the H component, the recovery phase started which lasted for several hours. During the recovery phase, the ring current decays and the H component of the magnetic field returns back to the normal levels.

– Two pronounced dips in the H component at 1:08 UT and 13:56 UT on September 8 are observed in the three stations. It can be seen that the first minima is strongly negative for MBO as compared to GUA and KOU. However, the second dip is strongly negative for MBO as compared to GUA and KOU. This behavior is due to the local time variation of the ring current during the storm. Overall, the largest disturbance of the H component of the magnetic field with amplitude $-180$ nT is observed at MBO as compared to $-150$ nT at KOU and $-140$ nT at GUA. The disturbance due to ionosphere electric current Diono which is the sum of the PPEF and the disturbance dynamo electric field (DDEF), is represented by the red curve in Figure 5. It follows anti-Sq signature during the storm period. It can be noted that during the first southward excursion of the magnetic field, the Diono, decreases at the GUA which is the noon sector. However, an increasing trend in the Diono is observed for the MBO and KOU which are the night sector. During the second southward excursion of the magnetic field, Diono decreases significantly for the MBO and KOU which are now on
the day side.

Targeted comments:
Page 1, lines 15-17: The classification of geomagnetic storms that is most widely accepted in the magnetospheric/geomagnetic community is that compiled by Gonzalez et al. (1994), so I suggest to refer to it in place of that by Loewe and Prolls (1997). Moreover, the citation of Tsurutani et al. (1992) at this point is not appropriate. I therefore suggest to cite Tsurutani et al. (1992) in place of Gonzalez et al. (1994) and vice versa. Of course, when citing the classification of Gonzalez et al. (1994) please check the thresholds of the Dst intervals and change the names of the different intensities of the geomagnetic storms.
Answer: In the revised manuscript following modification is done:
On the basis of the Dst index and the Bz component of the IMF, the geomagnetic storms can be categorized as follows: weak or minor storms (Dst $\leq -30$ nT, Bz $\leq -3$ nT during 1 hour), moderate storms (Dst $\leq -50$ nT, Bz $\leq -5$ nT during 2 hours), intense storms (Dst $\leq -100$ nT, Bz $\leq -10$ nT for 3 hours) and severe storms (Dst $\leq -200$ nT) (Gonzalez et al. (1994); Tsurutani et al. (1992); Loewe and Prolss (1997)). Some scientists have used the SYM-H geomagnetic index as a replacement of the Dst index due to advantage of its 1 min time resolution compared to the 1 h time resolution of the Dst index (Wanliss and Showalter (2006)). The 3 h value of the Kp index has also been used for the classification of the geomagnetic storms as: weak or minor storms ($5- \leq$ Kp $\leq 5$), moderate storms(Kp $\geq 6$), intense storms ($7- \leq$ Kp $\leq 7$) and severe storms (Kp $\geq 8-$) (Gosling et al. (1991)).

Page 1, line 19: Change "Therefore, the effects of geomagnetic storms are non uniform in different regions of the magnetosphere." Into "Therefore, geomagnetic storms produce effects that are different in the different regions of the magnetosphere".
Answer: The change is incorporated in the revised manuscript.

Page 1, line 21: Change "...observed which is almost two times higher than that of the quiet day value." Into "...observed, these have an amplitude that is almost twice that of a quiet day." Here the authors refer to "the quiet day". Are they referring to a specific quiet day or in general to "a quiet day"?
Answer: The change is incorporated in the revised manuscript.

Page 2, line 1: PPEF is generally used as the acronym of Prompt Penetration Electric Field and not Prompt Penetration Effects. Please correct the sentence.
Answer: The change is incorporated in the revised manuscript.

Page 2, line 2: Change "It is also found that the prompt penetration effect is almost uniform along the longitudinal direction." Into "It is also found that the effect of the prompt penetration electric field is almost uniform along the longitudinal direction."
Answer: The change is incorporated in the revised manuscript.

Page 1, line 19: "The ionosphere features vary along the latitudes and longitudes due to different current systems flowing in the magnetosphere." This sentence is too general and not completely correct. Better to say "During geomagnetic storms, the ionosphere features vary along the latitudes and longitudes also due to different current systems flowing in the magnetosphere."
Answer: The change is incorporated in the revised manuscript.

Page 2, line 29: Please specify something about the "energy transfer", e.g. it occurs between . . .
Answer: Following modification is done:
Many authors have analyzed the St. Patrick day storm (the largest geomagnetic storm

of the Solar cycle 24) by using the GPS-TEC data analysis techniques to understand the positive and negative ionospheric-storm effects due to energy transfer between the solar wind and the magnetosphere.

Page 2, line 32: I do not understand the logical sense of using "However" at this point.
Answer: 'However' has been removed.

Page 3, line 1: For the first time in the manuscript you mention here a "Northern equator anomaly". Which anomaly are you talking of? Please add something more.
Answer: Following modification is done in the revised manuscript:
A rapid enhancement in the ionospheric electron density distorts the structure of the northern equatorial ionization anomaly region. It is also observed that during the main phase a significant decrease in the vTEC occurs at the high latitude as compared to the lower latitude region. Moreover, the height of the peak electron density in the F2 layer also increases during the geomagnetic storm.

Page 4, lines 5-10: Please add a reference for Sym-H index and for AE index.
Answer: In the revised manuscript the references for the SYM-H and the AE indices have been added.

Page 4 line 15 Change "definite" into "definitive".
Answer: "definite" has been replaced by "definitive".

Page 5, lines 17-19: What differences are you talking of? Please specify. Correct, accordingly, also the caption of Figure 2.
Answer: The following modification is done in the revised manuscript:
Figure 2 shows the $\Delta$ REC (top), the $\Delta$ GEC (middle) and the SYM-H index (bottom)

during the period September 4-14, 2017. The $\Delta$ REC is calculated by taking the difference between the REC of each sector and the average daily values of the three quiet days before the storm having the Ap index below 22 nT. Similarly, the $\Delta$ GEC is the difference between the GEC and the average daily value of the three quiet days as considered in $\Delta$ REC.

Page 5, line 19: What do you mean by "the five quiet days"? Maybe "the five quietest days"? In any case you have to specify, for these days, the level of geomagnetic activity by using some geomagnetic activity index (e.g., Dst, Kp...). Answer: The quiet time variations are computed by using the five quiet days before the storm having the Ap index below 22 nT.

Page 5, lines 24-29: Change "panel" into "plots" everywhere in these lines. Panels are usually a composition of plots.
Answer: Panels has been replaced by plots in the revised manuscript.

Page 5, line 28: Invert the order of "daily" and "quiet".
Answer: Correct order "quiet daily" has been used in the revised manuscript.

Page 6, line 19: Change "magnetometer variations" into "magnetic field variations".
Answer: In the revised manuscript "magnetometer variations" is replaced by "magnetic field variations".

Page 7, line 1: Invert the order of "daily" and "quiet".
Answer: Correct order "quiet daily" has been used in the revised manuscript.

Page 7, line 1: Specify how the "disturbances" have been calculated.
Answer: In order to calculate the magnetic field variations we adopted the approach of

Nava et al. (2016); Kashcheyev et al. (2018). The brief description of this approach is now added in the revised manuscript as:

The storm time magnetic field variations are analyzed by using the data from the three low latitude observatories in three sectors: Asia (KOU), Africa (MBO) and America (GUA). The quasi-definitive data of these observatories which are available at http://intermagnet.org have been used for the analysis. Table 2 shows geographic and geomagnetic locations of these observatories. In order to calculate the magnetic field variations we adopted the approach of Nava et 2016, Kashcheyev 2018. The brief description of this approach is given here. During the geomagnetic storm, the horizontal component 'H' of the Earth's magnetic field can be expressed as:

$$H = H_o + D_M + D_{iono} + S_R^H,$$

where $H_o$ represents the magnetic field component due to Earth's external core dynamics, $D_M$ is the disturbance which comes from the magnetospheric currents mainly due to Chapman Ferraro current, ring current and tail current **?**. It can be calculated as:

$$D_M = SYM - H.cos\phi,$$

here $\phi$ is the geomagnetic latitude. The $S_R^H$ is the quiet daily regular variation of H and is computed by using the four quietest days having $Kp < 2$ such as:

$$S_R^H = \frac{1}{n}\sum_{i=1}^{n}(H_i + D_i^H) - H_o,$$

where n is the number of quiet days. The $D_i^H$ depicts the disturbances coming from the ionosphere $D_{iono}$ and the magnetosphere $D_M$. The magnetic disturbance due to ionospheric electric currents can be written as:

$$D_{iono} = \Delta H - S_q - SYM - H.cos\phi,$$

here $S_q = < S_R^H >$ is the hourly amplitude of daily variations of the geomagnetic field

Page 11: Caption of Figure 3, indicate what the dashed line is for.
Answer: The three dashed lines correspond to the impact of the CMEs on 6, 7 and 12 September 2017.

Typo/language comments:
Answer: In the revised manuscript the Typo and language mistakes have been removed according to the referee's suggestions.
Most Typo/language comments have been made directly on an annotated pdf. Below, additional comments.
"Data" is commonly used as with a plural meaning, please change verbs accordingly throughout the manuscript.
Answer: In the revised manuscript the verb has been changed according to the referee's suggestions.
Add a space between the value and its unit (for instance, change 10nT into 10 nT) throughout the manuscript.
Answer: In the revised manuscript a space has been introduced between the value and the unit.
Change "Index" into "index" if not at the beginning of a sentence, throughout the manuscript.
Answer: In the revised manuscript 'index" has been changed according to the referee's suggestions.
When referring to mid latitudes you use both "mid" and "middle", choose one of the two terms and use it always.
Answer: In the revised manuscript the "mid" has been used to refer mid latitudes.
Concerning the use of acronyms. Two ways can be followed: 1) not to define them, 2) to define them but then to use them. For instance HSSWS is defined twice and never used.
Answer: In the revised manuscript the acronym HSSWS has been removed since we

haven't used it.

References in the bibliography are formatted with different styles, please refer to the specific reference style of the journal.

Answer: In the revised manuscript the bibliography has been updated according to the Journal style.

Please also note the supplement to this comment:
https://www.ann-geophys-discuss.net/angeo-2019-19/angeo-2019-19-AC4-supplement.pdf

---

## Referee Report (RR1)

Dear Editor,

The manuscript has been improved. The discussion of the physical processes has been significantly extended. However, it still contains a few major and several minor issues/typos that must be addressed before publishing.

Best regards

**Major issues:**

P.5 L.1 Change "The GEC is the total number of electrons present in the ionosphere at the fixed altitude of about 450 km" to "The GEC is the total number of electrons present in the near-Earth space environment"

P.8 L.4 Change "quiet daily variations in blue which are calculated by subtracting the quiet time variations from the value itself. The quiet time variations are computed" To "quiet time daily variations in blue. The quiet time variations are computed…" The quite time daily variations are computed by averaging the quite time data, but not subtracting.

P.8 L.23 The figure 4 does not contain "contour plots", these are vTEC values color coded. I believe, Matlab surf function was used. Please correct the description.

P.10 L.14 Please clarify where exactly on the plot a reader can see the first (1:08 UT, September 8) of H component at MBO to be strongly negative. Also please make sure 'however' must be used, as both the first and the second statement say: "is strongly negative". From the plot provided, it can not be read the described details. A solution might be to indicate the peaks ("dips") with the arrows on the plot.

P.11 L.4 From the Figure 5, it can be clearly seen that H component at GUA drops below -150 nt (around 170-180), that is inconsistent with what is written. Please clarify.

**Typo/Corrections:**

Figure 4: It would be logical to have Pacific sector coming first, as the increase (a peak) in TEC is moving rightward, according to the local time of the sector.

Figure 5: The y label has Sh, while the legend shows Sq. Please harmonize them.

Figure 5 Title: Change "the ionosphere disturbance current" to "the variations due to disturbed ionospheric currents"

Everywhere: remove space between Δ and parameter name, e.g. "Δ REC" – "ΔREC"

P.2 L.18 Change "International Ground Station (IGS)" to "International GNSS Service (IGS)"

P.4 L.1 Change "three different longitudinal sectors" to "four different longitudinal sectors"

P.4 L.24 Change "among them the AE index, the Ap index, the Kp index and SYM-H" to "among them are AE, Ap, Kp and SYM-H indices"

P.4 L.27 Add space into "Data:The" – "Data: The"

P.4 L.27 Change "International GNSS Service Global Ionosphere Map" to "IGS Global Ionosphere Map (GIM)"

P.4 L.27 Change "data that are available in the standard IONEX format" to "data available in IONEX format"

P.4 L.28 Remove "; i.e., Crustal Dynamics Data information system"

P.4 L.29 Remove "These IONEX files contain the vTEC data for the entire globe. For any time, the vTEC data can be obtained from IONEX files at the time resolution of 2-h." As GIM maps with 15-min time interval are used in the study.

P.5 L.2 Change "the vTEC values" to "vTEC values", change "UPC-GIM" to "UPC GIM"

P.5 L.5 Change "of a certain GIM cells." to "of a certain GIM cell."

P.5 L.6 Change "is about" to "is"

P.5 L.6 Change "UPC-GIM" to "UPC GIM"

P.5 L.17 Add space into "observatories.In" - "observatories. In"

P.5 L.23 Change "current Cole (1966)" to "current (Cole, 1966)", change "It can be calculated" to "It can be estimated"

P.5 L.23 Change "." in the formula to the multiplication sign "·"

P.6 L.2 Change "." in the formula to the multiplication sign "·"

P.6 L.2 Define ΔH

P.6 L.7 Remove "mainly"

P.6 L.10 Remove "3 h"

P.6 L.11 Change "lead to a minor geomagnetic storm" to "lead to minor geomagnetic storms"

P.6 L.23 Change "in northward" to "is northward"

P.7 L.2 Change "The IEF is the Ey component" to "The Ey component of the IEF"

P.7 L.6 Change "CME1," to "the first CME"

P.7 L.9 Change "CME2," to "the second CME"

P.7 L.10 Remove "for 3 h"

P.8 L.23 Change "IGS-GIM" to "UPC GIM" (if these were UPC GIMs).

P.9 L.5 Change "magnetically equatorial region" to "magnetic equator region"

P.9 L.19 Change "An enhancement in the vTEC particularly, in the crests regions of the EIA are" to "An enhancement in the vTEC, in particular in the crests regions of the EIA, is"

---

## Referee Report (RR2)

Dear Editor,

Please find below the review of the revised manuscript by Nadia Imtiaz, Waqar Younas and Majid Khan entitled "Response of low to mid latitude ionosphere to the Geomagnetic storm of September 2017", submitted to Annales Geophysicae.

Despite the authors have deeply revised the manuscript it still need significant further improvements, for instance the problems of the Introduction addressed in the previous review have not been solved at all. Below, a list of main comments. Other comments (both main and minor) have been made directly on an annotated pdf.

**Main comments:**

1) Readability of the manuscript has increased but need to be improved. Several typos and inaccuracies are present also in the revised version. I encourage the author not to make only the corrections suggested by the reviewers but also to critically re-read the manuscript, possibly asking some colleague with a good English writing to read it.

2) The Introduction is very heavy to read and does not provide the reader a clear picture of the problem that is addressed by the manuscript, this part needs a thorough review. A lot of previous studies are listed without logical links and a logical sequence; it appears like a mere list of papers. Moreover not all of them are pertinent with the topic of the manuscript. Maybe some part could be removed and some other moved in the "Results/Discussion" section. Some suggestions are present in the annotated manuscript, but do not limit to them.

3) Section 2 should be renamed for instance "Data sets" into "Data and Analysis", "Results/Discussion" into "Results and Discussion".

4) "Data sets" section should be structured in a "narrative" form rather than in a "list" form. Moreover, this section does not give important information as the time sampling of the data used, the presence of gaps and the quality of data in general, the list of solar wind parameters used.

5) "Data sets" section: Formulas used in the part devoted to geomagnetic data are not rigorous. As they are, they are relations among constants and not among time varying quantities, time $t$ or an index is missing. Moreover these formulas need to be check (see Kashcheyev et al.) being one of them wrong (more details in the annotated pdf).

6) Page 6, lines 10-15. The sentence "The arrival of this CME caused a significant compression to the dayside magnetosphere which provoked a severe geomagnetic storm…" This is not correct, the only compression of the magnetosphere due to the arrival of the CME is not sufficient to generate a geomagnetic storm".

7) The plots of Figure 1 need to be reordered. Solar wind parameters/IMF are mixed with geomagnetic indices. I suggest to group, for instance on the top, the plots of Bz, Vsw and Ey, then F10.7 and all the remaining indices.

8) The section case study describe the behaviour of the single quantities plotted in Figure 1 but do not explain the physics, as far as concerns the present knowledge, underlying the observed behaviour.

**Response of low to mid-latitude ionosphere to the Geomagnetic storm of September 2017**

Nadia Imtiaz[1], Waqar Younas[2], and Majid Khan[2]

[1]Theoretical Physics Division, PINSTECH, Nilore, Islamabad, Pakistan
[2]Department of Physics, Quaid-i-Azam University, Islamabad, Pakistan

**Correspondence:** Nadia Imtiaz (nhussain@ualberta.ca)

**Abstract.** We study the impact of **the** geomagnetic storm of 7-9 September 2017 on the low-to-mid latitude ionosphere. The prominent feature of this solar event is the sequential occurrence of  two SYM-H minima **with** values of $-146$ **nT and** $-115$ **nT** on 8 September at $1:08$ UT and $13:56$ UT, respectively. The study is based on **the analysis of** data  from GPS stations and  **magnetic** observatories located at **the** different longitudinal sectors **corresponding to** Asia, Africa, America and Pacific during the period 4-14 September 2017. The GPS data  **are** used to derive the global, regional and vertical total electron content in the four selected regions. **Magnetic observatory data**  **are** used to illustrate the variation in the magnetic field**, particularly in its** horizontal component. It is observed that the storm time response of the vertical total electron content over the Asia/Pacific  is earlier than **over** the Africa and America  Overall, the positive ionospheric storm effects appear over the  dayside sectors which are associated with the ionospheric electric fields and the traveling atmospheric disturbances. The global thermospheric composition maps by Global Ultraviolet Imager exhibits a storm time variation in the $O/N_2$ ratio. The positive storm effects in the $O/N_2$ ratio occur in the low-latitudes and equatorial regions. It can be inferred that a variety of space weather phenomena such as the coronal mass ejection, the high speed solar wind stream and the solar radio flux can cause the multiple  enhancements  the vTEC in the low-to-mid latitude ionosphere during the period 4-14 September 2017.

**1 Introduction**

It is a well known fact that the geomagnetic storm is a temporary variation of the Earth's magnetic field induced by the coronal mass ejection (CME) or the high speed solar wind stream (HSSWS). **The most widely used**  **geomagnetic storms are: the disturbance storm time** ($Dst$) **index**  **the SYM-H index**  **the Kp index,** the auroral electrojet **(AE) index**  **the Ap index and the** $B_z$ **component of the Interplanetary Magnetic Field (IMF) (Rostoker (1972); Gonzalez et al. (1994); Saba et al. (1997)). On the basis of the** $Dst$ index and the $B_z$ component of the  geomagnetic storms can be categorized as follows: weak or minor storms ($Dst \leq -30$ **nT,** $B_z \leq -3$ **nT**  1 hour), moderate storms ($Dst \leq -50$ **nT,** $B_z \leq -5$ **nT**  2 hours), intense storms ($Dst \leq -100$ **nT,** $B_z \leq -10$ **nT for 3 hours) and severe storms (**$Dst \leq -200$ **nT) (Gonzalez et al. (1994);**

Tsurutani et al. (1992); Loewe and Prolss (1997)). Some scientists the SYM-H geomagnetic index of the $Dst$ index due to its 1-min time resolution compared to the 1-h 
[revised manuscript text omitted]
 the AE index, the Ap index, the Kp index and SYM-H. The SYM-H is the proxy of the ring currents and the AE index estimates the energy transfer from the solar wind to the auroral ionospheric regions (Rostoker (1972); Wanliss and Showalter (2006)).**

4. Electron Content Data:**The vTEC is extracted from the International GNSS Service Global Ionosphere Map (IGS-GIM) data that are available in the standard IONEX format on the NASA's website; i.e., Crustal Dynamics Data information system (ftp://cddis.gsfc.nasa.gov/gps/products/ionex/). These IONEX files contain the vTEC data for the entire globe. For any time, the vTEC data can be obtained from IONEX files at the time resolution of 2-h.** The **tomographic kriging GIMs computed by the Technical University of Catalonia (UPC),** have been used to study the

GEC variations during the storm period under consideration. The GEC is the total number of electrons present in the ionosphere at the fixed altitude of about 450 **km**. **The GEC is obtained from the UPC-GIM data by the summation of the vTEC values in a cell $I_{i,j}$ multiplied by a cell's area $S_{i,j}$ over all GIM cells and it is given by Afraimovich et al. (2006),**

$$GEC = \sum_{i,j} I_{i,j}.S_{i,j}.$$

**Here $i$ and $j$ represent the longitude and latitude of a certain GIM cells. The latitudinal and longitudinal extent of the elementary GIM cell is about $2.5°$ and $5°$, respectively. The unit of GEC is $1 GECU = 10^{32}$ electrons.** The REC is the total number of electrons in the specified region of the ionosphere. Our analysis is based on the four different regions: Asia ($60° : 150°E$), Africa ($-30° : 60°E$), America ($-120° : -30°E$) and **Pacific ($-180° : -120°E$; $150° : 180°E$).** The REC is calculated **similarly to the** GEC, with the summation **being** restricted to the GIM cells of that particular region. **For both GEC/REC, the UPC-GIM data at time resolution of 15 min have been used.**

5.  The data sets of nine GPS stations **are** analyzed here. These stations are selected on the basis of data availability and their geographic/geomagnetic location. The geographic **and geomagnetic locations** of these stations are given in Table 1.

6.  The storm time magnetic field variations are analyzed by using the data from the three low latitude observatories in the three sectors: Asia (KOU), Africa (MBO) and America (GUA). The **quasi-definitive data sets** of these observatories which are available at **http://intermagnet.org** have been used for the analysis. Table 2 **shows the geographic and geomagnetic locations** of these observatories. **In order to calculate the magnetic field variations, we have adopted the approach of Nava et al. (2016); Kashcheyev et al. (2018). The brief description of this approach is given here. During the geomagnetic storm, the horizontal component $H$ of the Earth's magnetic field can be expressed as:**

$$H = H_o + D_M + D_{iono} + S_R^H,$$

**where $H_o$ represents the magnetic field  Earth's  core , $D_M$ is the disturbance which comes from the magnetospheric currents mainly due to the Chapman Ferraro current, the ring current and the tail current Cole (1966). It can be calculated as:**

$$D_M = \textbf{SYM-H}.cos\phi,$$

**here $\phi$ is the geomagnetic latitude. The $S_R^H$ is the quiet daily  variation of $H$ and is computed by using the five quietest days of September 2017 having $Kp < 2$ such as:**

$$S_R^H = \frac{1}{n}\sum_{i=1}^{n}(H_i + D_i^H) - H_o,$$

where n is the number of quiet days. The $D_i^H$ depicts the disturbances coming from the ionosphere $D_{iono}$ and the magnetosphere $D_M$. The magnetic disturbance due to the ionospheric electric currents can be written as:

$$D_{iono} = \Delta H - S_q - \text{SYM-H}.cos\phi,$$

here $S_q = <S_R^H>$ is the hourly amplitude of daily variations of the geomagnetic field.

[revised manuscript text omitted]

  – **During geomagnetically quiet conditions, the** $E \times B$ **drift at the dip equator creates the Equatorial Ionization Anamoly (EIA) at** $\pm 10 - 15°$ **from the equator (Balan and Bailey (1995); Fejer (1991)). In response to the geomagnetic storm the latitudinal extent of the EIA is increased upto about** $30°$ **latitudes.**

- **All the four sectors show an enhancement in the vTEC on 6 September.** This behavior can be associated with the impact of the CME1 which arrived at 23:43 UT on 6 September.

- During the initial phase of the storm on 7 September, the vTEC enhancement mainly occurs in the crest regions of the EIA with a clear latitudinal separation.

- On the day of the storm, the vTEC strongly enhanced in the crests of the EIA and in the  as compared to the days before and after the storm. The enhancements of the vTEC in the EIA region in response to the geomagnetic storms have been reported in many studies (Zhao et al. (2005); Astafyeva et al. (2015); Lei et al. (2018)).

[revised manuscript text omitted]

15  **At the same time, the $O/N_2$ decreases significantly at mid to high latitudes as compared to the quiet time pattern.** This observation is consistent with the behavior of the vTEC during the storm period. After the recovery of the storm, the thermospheric composition returns to its normal profile.

**5   Conclusions**

We presented the impact of geomagnetic storm of 7-9 September 2017 on the low-to-mid latitude ionosphere over the four

20  longitudinal sectors; i.e., Asia, Africa, America and Pacific. The storm effects are characterized by using the diverse parameters including the global, regional and vertical total electron content derived from the GPS data, the horizontal component of the magnetic field obtained from the magnetometers and the neutral composition from the GUVI/TIMED. It is observed that the positive storm effects occur in the local dayside stations. The temporal response of the four sectors shows that the positive storm effects in the REC and vTEC over the Asian/Pacific sectors are observed earlier than the American/African sectors.

25  During geomagnetically quiet conditions, most of the TEC is confined to the equatorial and low latitude regions. However, the latitudinal extent of the TEC increases upto the mid-latitudes during the storm period. The vTEC enhancements observed on the other days are due to the high speed solar wind stream event. The analysis of the magnetometers data shows the largest disturbance of the horizontal component of the magnetic field at MBO as compared to that of KOU and GUA. The storm time variation of the horizontal component is associated with the Chapman-Ferraro and the ring currents. The magnetic field

30  component associated with the disturbed ionospheric current follows the anti-Sq variations which depends on the prompt penetration electric field and the disturbance dynamo electric field. The storm time enhancement in the neutrals ratio; i.e., $O/N_2$ over the low latitudes and equator is consistent with the observed TEC behavior. However, the negative storm effect in

the $O/N_2$ ratio can be observed in the higher and mid-latitude regions. Overall, the positive storm phase occur on the dayside sectors during the G4 geomagnetic storm of September 7-9, 2017. It can be concluded that the ionosphere dynamics and electrodynamics play important role in the observed perturbations in the low-to-mid latitude ionosphere during 4-14 September 2017. The study would be useful for the understanding of storm time response of the low-mid latitude ionosphere.

5   *Acknowledgements.*   The authors are grateful to the space weather data resources: the OMNI data base https://omniweb.gsfc.nasa.gov/form/dx1.html for providing the solar wind data, to the World Data Center for Geomagnetism at Kyoto University, Japan for providing the geomagnetic data: http://wdc.kugi.kyoto-u.ac.jp/dstrealtime/ and to the International GNSS Service (IGS) team for providing the GPS data. NI acknowledge the UNOOSA/ICTP for providing the financial support to attend GNSS workshop and learn these data analysis techniques. The authors are very grateful to the anonymous referees for their constructive and insightful comments in improving the manuscript. The authors would like

10  to thanks Anton Kashcheyev and Christine Amory Mazaudier for their valuable suggestions. This research work was partly supported by the HEC Pakistan: $GrantNo.7632/Federal/NRPU/RD/HEC/2017$.

[Figure]

**Figure 1.**  $B_z$ component  $V_{sw}$,  AE index,  interplanetary electric field (IEF),  **SYM-H** index,  Kp index and $F_{10.7}$ characterizing the geomagnetic storm during 4-14 September 2017

[Figure]

**Figure 2.** Variation of the Regional electron content (top), the Global electron content (bottom) and the SYM-H index during the geomagnetic storm of 4-14 September 2017

[Figure]

**Figure 3.** The vTEC variations at GPS stations during the geomagnetic storm of 4-14 September 2017. Each plot illustrates the disturbed vTEC (in orange) and its quiet value (in blue).

[Figure]

**Figure 4.** The vTEC variations over the Asian (first plot), African (second plot), American (third plot), Pacific (fourth plot) sectors and the SYM-H index (bottom plot) during the geomagnetic storm of 4-14 September 2017

[Figure]

**Figure 5.** The magnetometer $H$ variations at specific stations during 4-14 September 2017 over the three sectors: the Asian (top), the African (middle) and the American (bottom). On each plot the **quiet daily** variations (blue), the actual $H$ variations (black) and the ionosphere disturbance current **(red)** are plotted

[Figure]

**Figure 6.** The thermospheric $O/N_2$ ratio obtained from the GUVI/TIMED during G4 category storm which occurred between 7-10 September 2017

9) In Results/Discussion section, to calculate DGEC and DREC the quiet values of GEC and REC are defined through Ap index. Why do the authors now refer to this index and not to Sym-H or Kp, that have been already used. It is necessary to introduce another index? Where do the value of 22 nT, considered as a threshold, come from?

10) Figure 2. Since the authors talk of DGEC before of DREC, the order of the two plots should be inverted. Moreover, Figure 2 shows SymH not AE as written in the maintext. Which is right? Text or figure? Please correct the one that is wrong.

11) On the discussion of Figure 2. Features observed in DGEC and DREC are related to features of the AE index. The AE index do give an idea of the energy transfer from the solar wind to the magnetosphere but it is highly representative of phenomena occurring at the high latitudes while this paper deals with mid-to-low latitudes. How phenomena occurring at mid-to-low latitudes can be explained in terms of high-latitude ionosphere dynamics and hence of AE index?

12) Page 8, line 30. What the authors mention here is not accurate. Indeed, the intensification of the EIA is present also before the geomagnetic storm in the Pacific (5 September) and African sectors (6 September), while in the Asian sector the pattern of vTEC is does changes dramatically from 4 September to 11 September.

13) Figure 5. Since here the authors have plotted all but Hm components of the observed magnetic field (i.e. Diono and Sq), I suggest to add also Hm.

14) Page 11, lines 11-17. The part on the observation of O/N2 is not convincing and not essential for the manuscript. First of all, the authors dedicate to this part about 1/3 of the abstract and a very minimal part of the manuscript. In these lines the authors talk of a significant decrease, but how do they objectively measure this significance? It is not possible to discern it only by looking at the maps that, on top of that, do not show very evident changes in the colours.

---

## Referee Report (RR3)

Dear Editor,

This paper has been improved in accordance with the reviewer comments. However, there are still few issues that must be addressed before publishing.

Best regards

P.6 L.6-8
Looking at Figure 1, the following statements are incorrect and must be modified correspondingly:
"The SYM-H index also follows the pattern of the Bz component."
"From 1 : 08 UT until 11 : 00 UT the Bz is northward; i.e., it increases to the positive value."
"Following the Bz, the SYM-H also increases from −146 nT to the value of −38 nT."

SYM-H is not following the Bz behavior, as it changes the direction and does not stay always negative from 1:08 to 11 UT.

P.10 l.11-12
"Overall, the largest disturbance of the H component of the magnetic field with amplitude −180 nT is observed at GUA as compared to −150 nT at MBO and −140 nT at KOU."
By looking at Fig 5, one can easily see that:

H drops down below -150 nT at MBO.

[Figure]

H drops down below -140 nT at KOU.

[Figure]

Please, correct the sentence accordingly, or make sure plots are consistent with the text. Also, make the three subplots having the same dynamic range.

P.6. L.21
"Figure 2 shows the _REC (top), the _GEC (middle) and the SYM-H index (bottom) during the period 4-14 September 2017"
must be
"Figure 2 shows the _GEC (top), the _REC (middle) and the SYM-H index (bottom) during the period 4-14 September 2017"

P.10 L.14
"It can be noted that during the first southward excursion of the magnetic field…"
should be
"It can be noted that during the first southward excursion of the IMF"
or clarify what "southward excursion" of the magnetic field you are referring to.

P.3 L. 17-18 Change "a number of space weather events observed" to "a number of space weather events were observed"

P.10 L.25
"the latitudinal extent of the TEC increases"
replace with
"the latitudinal extent of the bulk of the TEC increases"

---

## Author Response (AR2)

Dear Editor,
We are thankful to the referee 1 for reviewing and helping us to improve this paper. The manuscript has been improved according to the referee's suggestions. In the revised manuscript, the modifications are in the bold letters.
Best regards,

Nadia Imtiaz.

Response to the Major issues:

**P.5 L.1 Change "The GEC is the total number of electrons present in the ionosphere at the fixed altitude of about 450 km" to "The GEC is the total number of electrons present in the near-Earth space environment"**

Response: In revised manuscript we changed "The GEC is the total number of electrons present in the ionosphere at the fixed altitude of about 450 km" to "The GEC is the total number of electrons present in the near-Earth space environment".

**P.8 L.4 Change "quiet daily variations in blue which are calculated by subtracting the quiet time variations from the value itself. The quiet time variations are computed" To "quiet time daily variations in blue. The quiet time variations are computed..." The quite time daily variations are computed by averaging the quite time data, but not subtracting.**

Response: The revised manuscript has been modified as: "The quiet time daily variations are computed by averaging the quiet time data of the five days before the storm having the Kp index below $4$ "

**P.8 L.23 The figure 4 does not contain "contour plots", these are vTEC values color coded. I believe, Matlab surf function was used. Please correct the description.**

Response: We didn't use Matlab surf function. The vTEC plots are "contour plots". However, we followed the referee suggestion and removed it.

**P.10 L.14 Please clarify where exactly on the plot a reader can see the first (1:08 UT, September 8) of H component at MBO to be strongly negative. Also please make sure 'however' must be used, as both the first and the second statement say: "is strongly negative". From the plot provided, it can not be read the described details. A solution might be to indicate the peaks ("dips") with the arrows on the plot.**

Response: In the revised manuscript the correction has been made. The first dip  (at 1:08 UT on September 8) of the H component is strongly

negative for GUA as compared to MBO and KOU. However, the second dip (at 13:56 UT on September 8) of the H component is strongly negative for MBO as compared to GUA and KOU.

**P.11 L.4 From the Figure 5, it can be clearly seen that H component at GUA drops below -150 nt (around 170-180), that is inconsistent with what is written. Please clarify.**

Response: In the revised manuscript the following correction has been made: Overall, the largest disturbance of the H component of the magnetic field with amplitude −180 nT is observed at **GUA** as compared to −150 nT at **MBO** and −140 nT at **KOU.**

**Typos/Corrections:**

In the revised manuscript the following Typos/Corrections have been made:

**Figure 4: It would be logical to have Pacific sector coming first, as the increase (a peak) in TEC is moving rightward, according to the local time of the sector.**

Response: The plots in Figure 4 has been arranged in the logical order according to the local time of each sector.

**Figure 5: The y label has Sh, while the legend shows Sq. Please harmonize them.**

Response: The discrepancy in the label and legend in Figure 5 has been removed.

**Figure 5 Title: Change "the ionosphere disturbance current" to "the variations due to disturbed ionospheric currents"**

Response: In the Caption of Figure 5, "the ionosphere disturbance current" has been changed to "the variations due to disturbed ionospheric currents".

**Everywhere: remove space between Δ and parameter name, e.g. "Δ REC" – "ΔREC"**

Response: In the revised manuscript, a space between Δ and parameter name GEC/REC has been removed.

**P.2 L.18 Change "International Ground Station (IGS)" to "International GNSS Service (IGS)"**

Response: In the revised manuscript, "International Ground Station (IGS)" has been replaced with "International GNSS Service (IGS)".

**P.4 L.1 Change "three different longitudinal sectors" to "four different longitudinal sectors"**

Response: In the revised manuscript, "three different longitudinal sectors" has been changed to "four different longitudinal sectors".

**P.4 L.24 Change "among them the AE index, the Ap index, the Kp index and SYM-H" to "among them are AE, Ap, Kp and SYM-H indices"**

Response: In the revised manuscript, "the AE index, the Ap index, the Kp index and SYM-H" has been changed to "among them are AE, Ap, Kp and SYM-H indices".

**P.4 L.27 Add space into "Data: The" – "Data: The"**

Response: In the revised manuscript, a space has been added "Data: The".

**P.4 L.27 Change "International GNSS Service Global Ionosphere Map" to "IGS Global Ionosphere Map (GIM)".**

Response: In the revised manuscript, "International GNSS Service Global Ionosphere Map" has been changed to "IGS Global Ionosphere Map (GIM)".

**P.4 L.27 Change "data that are available in the standard IONEX format" to "data available in IONEX format"**

Response: In the revised manuscript, "data that are available in the standard IONEX format" has been changed to "data available in IONEX format".

**P.4 L.28 Remove "; i.e., Crustal Dynamics Data information system"**

Response: In the revised manuscript, "; i.e., Crustal Dynamics Data information system" has been removed.

**P.4 L.29 Remove "These IONEX files contain the vTEC data for the entire globe. For any time, the vTEC data can be obtained from IONEX files at the time resolution of 2-h." As GIM maps with 15-min time interval are used in the study.**

Response: In the revised manuscript, "These IONEX files contain the vTEC data for the entire globe. For any time, the vTEC data can be obtained from IONEX files at the time resolution of 2-h." has been removed.

**P.5 L.2 Change "the vTEC values" to "vTEC values", change "UPC-GIM" to "UPC GIM"**

Response: In the revised manuscript, "the vTEC values" has been changed to "vTEC values", changed "UPC-GIM" to "UPC GIM".

**P.5 L.5 Change "of a certain GIM cells." to "of a certain GIM cell."**

Response: In the revised manuscript, "of a certain GIM cells." has been changed to "of a certain GIM cell."

**P.5 L.6 Change "is about" to "is"**

Response: In the revised manuscript, "is about" has been changed to "is".

**P.5 L.6 Change "UPC-GIM" to "UPC GIM"**

Response: In the revised manuscript, "UPC-GIM" has been changed to "UPC GIM".

**P.5 L.17 Add space into "observatories.In" - "observatories. In"**

Response: In the revised manuscript, a space has been added into "observatories.In" as "observatories. In".

**P.5 L.23 Change "current Cole (1966)" to "current (Cole, 1966)", change "It can be calculated" to "It can be estimated"**

Response: In the revised manuscript, "current Cole (1966)" has been changed to "current (Cole, 1966)", and "It can be calculated" has been changed to "It can be estimated".

**P.5 L.23 Change "." in the formula to the multiplication sign "·"**

Response: In the revised manuscript, the correct mathematical symbol has been used for multiplication.

**P.6 L.2 Change "." in the formula to the multiplication sign "·"**

Response: In the revised manuscript, the correct mathematical symbol has been used for multiplication.

**P.6 L.2 Define ΔH**

Response: In the revised manuscript, we have defined the variation in the H component of the magnetic field which is given as $\Delta H = H - H_o$.

**P.6 L.7 Remove "mainly"**

Response: In the revised manuscript, "mainly" has been removed.

**P.6 L.10 Remove "3 h"**

Response: In the revised manuscript, "3 h" has been removed.

**P.6 L.11 Change "lead to a minor geomagnetic storm" to "lead to minor geomagnetic storms"**

Response: In the revised manuscript, "lead to a minor geomagnetic storm" has been changed to "lead to minor geomagnetic storms".

**P.6 L.23 Change "in northward" to "is northward"**

Response: In the revised manuscript, "in northward" has been changed to "is northward".

**P.7 L.2 Change "The IEF is the Ey component" to "The Ey component of the IEF"**

Response: In the revised manuscript "The IEF is the Ey component" has been changed to "The Ey component of the IEF".

**P.7 L.6 Change "CME1," to "the first CME"**

Response: In the revised manuscript "CME1," has been changed to "the first CME".

**P.7 L.9 Change "CME2," to "the second CME"**

Response: In the revised manuscript "CME2," has been changed to "the second CME".

**P.7 L.10 Remove "for 3 h"**

Response: In the revised manuscript, "for 3 h" has been removed.

**P.8 L.23 Change "IGS-GIM" to "UPC GIM" (if these were UPC GIMs).**

Response: These are IGS GIM.

**P.9 L.5 Change "magnetically equatorial region" to "magnetic equator region"**

Response: In the revised manuscript "magnetically equatorial region" has been changed to "magnetic equator region".

**P.9 L.19 Change "An enhancement in the vTEC particularly, in the crests regions of the EIA are" to "An enhancement in the vTEC, in particular in the crests regions of the EIA, is"**

Response: In the revised manuscript "An enhancement in the vTEC particularly, in the crests regions of the EIA are" has been changed to "An enhancement in the vTEC, in particular in the crests regions of the EIA, is".

Dear Editor,
We are thankful to the referee 2 for reviewing and helping us to improve this paper. The manuscript has been improved according to the referee's suggestions. In the revised manuscript, the modifications are in the bold letters.
Best regards,

Nadia Imtiaz.

**Main comments:**
1) **Readability of the manuscript has increased but need to be improved. Several typos and inaccuracies are present also in the revised version. I encourage the author not to make only the corrections suggested by the reviewers but also to critically re-read the manuscript, possibly asking some colleague.**

   Response: With due apology, it is stated that English is not our native language. We have tried our level best to improve the manuscript and consulted some of our colleagues to help us in improving the manuscript.

2) **The Introduction is very heavy to read and does not provide the reader a clear picture of the problem that is addressed by the manuscript, this part needs a thorough review. A lot of previous studies are listed without logical links and a logical sequence; it appears like a mere list of papers. Moreover not all of them are pertinent with the topic of the manuscript. Maybe some part could be removed and some other moved in the "Results/Discussion" section. Some suggestions are present in the annotated manuscript, but do not limit to them.**

   Response: In the revised manuscript, the introduction has been shortened (less than 3 pages) and studies are linked logically.

3) **Section 2 should be renamed for instance "Data sets" into "Data and Analysis", "Results/Discussion" into "Results and Discussion".**

   Response: In the revised manuscript Section 2 and 4 have been renamed as "Data and Analysis" and "Results and Discussion".

4) **"Data sets" section should be structured in a "narrative" form rather than in a "list" form. Moreover, this section does not give important information as the time sampling of the data used, the presence of gaps and the quality of data in general, the list of solar wind parameters used.**

   Response: In the revised manuscript the Data and Analysis"

section has been structured in a narrative form instead of listing.

5) **"Data sets" section: Formulas used in the part devoted to geomagnetic data are not rigorous. As they are, they are relations among constants and not among time varying quantities, time t or an index is missing. Moreover these formulas need to be check (see Kashcheyev et al.) being one of them wrong (more details in the annotated pdf).**

   Response: In the revised manuscript, the discrepancies in the formulas have been removed by rechecking the published study of Kashcheyev et al.

6) **Page 6, lines 10-15. The sentence "The arrival of this CME caused a significant compression to the day side magnetosphere which provoked a severe geomagnetic storm..." This is not correct, the only compression of the magnetosphere due to the arrival of the CME is not sufficient to generate a geomagnetic storm".**

   Response: In the revised manuscript, the sentence "The arrival of this CME  caused a significant compression to the day side magnetosphere which provoked  a severe geomagnetic storm..."has been corrected as:  "The arrival of this CME caused a significant disturbance in the magnetosphere which leads  to a severe geomagnetic storm having maximum value of the geomagnetic index    Kpmax = 8."

7) **The plots of Figure 1 need to be reordered. Solar wind parameters/IMF are mixed with geomagnetic indices. I suggest to group, for instance on the top, the plots of Bz, Vsw and Ey, then F10.7 and all the remaining indices.**

   Response: In the revised manuscript, the plots of Figure 1 have been reordered according to the referee's suggestions.

8) **The section case study describe the behavior of the single quantities plotted in Figure 1 but do not explain the physics, as far as concerns the present knowledge, underlying the observed behavior.**

   Response: The case study contains a brief description of the space weather  events and the resulting variations in plasma and magnetic field parameters   occurred during September 4-14, 2017. We have explained the underlying physics in    the Results and Discussion section.

9) **In Results/Discussion section, to calculate DGEC and DREC the**

quiet values of GEC and REC are defined through Ap index. Why do the authors now refer to this index and not to Sym-H or Kp, that have been already used. It is necessary to introduce another index? Where do the value of 22 nT, considered as a threshold, come from?

Response: In the revised manuscript, we have removed the Ap index and considered quiet days on the value of Kp<3.

10) Figure 2. Since the authors talk of DGEC before of DREC, the order of the two plots should be inverted. Moreover, Figure 2 shows SymH not AE as written in the main text. Which is right? Text or figure? Please correct the one that is wrong.

Response: In the revised manuscript, the order of ΔGEC and ΔREC plots have been reverted. The bottom plot is SYM-H index, the text of Figure 2 has been    corrected.

11) On the discussion of Figure 2. Features observed in DGEC and DREC are related to features of the AE index. The AE index do give an idea of the energy transfer from the solar wind to the magnetosphere but it is highly representative of phenomena occurring at the high latitudes while this paper deals with mid-to-low latitudes. How phenomena occurring at mid-to-low latitudes can be explained in terms of high-latitude ionosphere dynamics and hence of AE index?

Response: The AE index indicates the storm time energy inputs from the solar wind to the magnetosphere. The auroral region is the region of the strong coupling between the interplanetary medium, the magnetosphere, the thermosphere and the ionosphere. The storm time enhanced auroral electric currents can drive the equatorward thermospheric winds via joule heating and the momentum transfer. The thermospheric winds extending from the auroral to the mid and low latitudes produce strong daytime ionization and hence, increase the electron content.

12) Page 8, line 30. What the authors mention here is not accurate. Indeed, the intensification of the EIA is present also before the geomagnetic storm in the Pacific (5 September) and African sectors (6 September), while in the Asian sector the pattern of vTEC is does changes dramatically from 4 September to 11 September.

Response: In the Asian sector, the vTEC exhibits a regular behavior. Everyday we observe well-defined crests of the EIA except on September 8. On the day of the storm, a strong

intensification of the vTEC with a complex pattern can be observed in the Asian sector. On September 10, the pattern returns to the normal form as it was before the storm. An irregular pattern of the vTEC can be seen in Pacific, African and American sectors during the period September 4-14, 2017.
On September 5 and 11, the amplitude of the vTEC at crests of the EIA is higher than that on the days prior to the storm (September 4). The observed intensification of the vTEC can be attributed to the HSSWS effect.

13) **Figure 5. Since here the authors have plotted all but Hm components of the observed magnetic field (i.e. Diono and Sq), I suggest to add also Hm.**

   Response: In Figure 5, we have plotted three components; i.e., H, Diono and Sq. Here our focus is to study the storm time response of the H component.

14) **Page 11, lines 11-17. The part on the observation of O/N2 is not convincing and not essential for the manuscript. First of all, the authors dedicate to this part about 1/3 of the abstract and a very minimal part of the manuscript. In these lines the authors talk of a significant decrease, but how do they objectively measure this significance? It is not possible to discern it only by looking at the maps that, on top of that, do not show very evident changes in the colour.**

   Response: By considering the referee's point of view about O/N2 observation, we have removed O/N2 observation part from the revised manuscript.

**Note: We have incorporated the minor and major comments made by the referee directly on an annotated pdf.**

---

## Author Response (AR3)

Dear Editor,
We are thankful to the anonymous referees for reviewing and helping us to improve this paper. The manuscript has been improved according to their suggestions. In the revised manuscript, the modifications are in the bold letters.
Best regards,

Nadia Imtiaz.

**Reply to Referee #1:-**

**1. Line 15 on the first page, replace "useful" with "used".**
Response: In the revised manuscript on Line 15, "useful" has been replaced with "used".

**2. The last two equations on Page 4. Clearly, the index i and j are different to those in the first equation on the same page. So I suggest the author to use difference indexes to avoid confusion.**
Response: In the revised manuscript on Page 4, the indices i and j in the first equation have been replaced with l and m.

**3. There is no equation number throughout the paper. Please add.**
Response: In the revised manuscript, the equation numbers have been added.

**4. On page 6, the author mentions the solar F10.7 plot. However, there is not much follow up discussion to relate the main subjects with the solar flux variation. So, either remove the plot or provide more related discussion, if it can strengthen your argument.**
Response: In the revised manuscript on P. 7, the following modification has been done:
Some authors have analyzed the behavior of the global electron content (GEC) with the variations of the 10.7-cm solar radio emission; i.e., F10.7 index Nava et al. (2016).
From 4 to 8 September, the F10.7 index is higher than 100 sfu as shown in Figure 1. During this period, we observed the higher value of ΔGEC which decreases significantly after 9 September as shown in Figure 2. According to Afraimovich et al. (2007, 2008) and Nava et al. (2016), there is a correlation between the GEC and the F 10.7 index. Therefore, it can be inferred that the variation of the GEC from 4 to 8 September can also be affected by the higher solar flux; i.e., F 10.7 > 100 sfu.

**Reply to the Referee #2:-**

**P.6 L.6-8 Looking at Figure 1, the following statements are incorrect and must be modified correspondingly:**

**"The SYM-H index also follows the pattern of the Bz component.**

**"From 1 : 08 UT until 11 : 00 UT the Bz is northward; i.e., it increases to the positive value."**

**"Following the Bz, the SYM-H also increases from −146 nT to the value of −38 nT."**

**SYM-H is not following the Bz behavior, as it changes the direction and does not stay always negative from 1:08 to 11 UT.**

Response: Thank You for the careful reading and pointing out the errors in numerical values which have been corrected in the revised manuscript on P.6 L. 6-8 as following:

In order to analyze the geomagnetic activity behavior, the SYM-H index is also presented in Figure 1. During the main phase of the storm, the SYM-H index decreases and reaches the negative value of ≃ −146 nT thus producing the first minima of the SYM-H index at 1 : 08 UT. During the partial recovery phase from 1:08 UT to 11:00 UT, the SYM-H also increases from −146 nT to the value of −38 nT. Thereafter, the SYM-H index decreases again and it reaches the second minimum value of ≃ −115 nT. This is the end of the main phase of the storm which lasted for ~ 15 hours.

**P.10 l.11-12 "Overall, the largest disturbance of the H component of the magnetic field with amplitude −180nT is observed at GUA as compared to −150 nT at MBO and −140 nT at KOU." By looking at Fig 5, one can easily see that H drops down below -150 nT at MBO.**

Response: In the revised manuscript, on P.10 L. 11-12 following modification is done: It can be seen that the first dip (around 1:08 on 8 September) is strongly negative for both GUA (-142 nT) and KOU (-142.5 nT) as compared to MBO (-102 nT). However, the second dip (around 13:56 on 8 September) is strongly negative for MBO (-164 nT) as compared to GUA (-133 nT) and KOU(-112 nT).

**P.6. L.21 "Figure 2 shows the _REC (top), the _GEC (middle) and the SYM-H index (bottom) during the period 4-14 September 2017" must be "Figure 2 shows the _GEC (top), the _REC (middle) and the SYM-H index (bottom) during the period 4-14 September 2017".**

Response: In the revised manuscript, on P.6 L.21 has been modified as "Figure 2 shows the ΔGEC (top), the ΔREC (middle) and the SYM-H index (bottom) during the period 4-14 September 2017."

**P.10 L.14 "It can be noted that during the first southward excursion of the magnetic field..."**
**should be**
**"It can be noted that during the first southward excursion of the IMF" or clarify what "southward excursion" of the magnetic field you are referring to.**
Response: In the revised version, on P.10 Line 18 has been modified as: "It can be noted that during the first southward excursion of the IMF."

**P.3 L. 17-18 Change "a number of space weather events observed" to "a number of space weather events were observed"**
Response: In the revised version, on P.3 L. 17-18 has been modified as "a number of space weather events were observed between 4-14 September 2017."

**P.10 L.25 "the latitudinal extent of the TEC increases" replace with "the latitudinal extent of the bulk of the TEC increases"**
Response: In the revised manuscript, on P.10, L.31 has been modified as "the latitudinal extent of the bulk of the TEC increases".

---

## Author Response (AR4)

Dear Editor,
We are thankful to the referees and the editor for reviewing and helping us to improve this paper. The report of the Anonymous Referee #01 shows acceptance of the paper as it is. However, the manuscript has been improved according to the suggestions of the Anonymous Referee # 3. In the revised manuscript, the modifications are in the bold letters.

Thank You!
Best regards,

Nadia Imtiaz.

**Reply to Anonymous Referee #3:-**

**The author presents a detailed case study for a strong geomagnetic storm (September 2017) and its impact on ionosphere. However, the discussion is a little thin, mainly focusing on the storm-time electro-dynamic variations, and does not address effects due to the dramatic changes in neutral dynamics on the variations of these observed parameters. As pointed out by David et al., 2016, the variations of the neutral wind in the thermosphere can provide notable uncertainties in ionosphere models, and the work by Li et al, 2018 and 2019, trying to explain the observed temperature enhancement in lower thermosphere above mid-latitude (Yuan et al., 2015), have demonstrated considerable neutral wind changes in the thermosphere during the strong geomagnetic storm above middle and low latitude. Therefore, it is important to recognize the role of changing neutral winds on the variations of ionosphere, especially for middle and low latitudes. I hope the author could address this subject in the discussion.**

Response: In order to demonstrate the changes in the thermosphere during the strong geomagnetic storm, we have added the response of the thermosphere in the revised manuscript. See the following sections: abstract, data analysis, Results and discussion, conclusion and Figure 6.

**Minor Comments:**

**1. On page 1, in the first sentence of Introduction. Please replace "a known fact" with "well-known".**
Response: In the revised manuscript on Page 1, in the first line: "a known fact" has been replaced with "well-known".

**2. On page 2, line 5, please replace "presented" with "utilized".**
Response: In the revised manuscript, "presented" has been replaced with "utilized".

**3. On the same page, line 6, please delete "There are".**
Response: In the revised manuscript, "There are" has been deleted.

**4. On the same page, line 16, please replace "authors" with "studies".**
Response: In the revised manuscript, "authors" has been replaced with "studies".

**5. On page 3, line 2, please replace "On the basis of " with "Based upon".**
Response: In the revised manuscript,"On the basis of" has been replaced with "Based upon".

**6. On the same page, line 15, please delete "under consideration"**
Response: In the revised manuscript, "under consideration" has been deleted.

---

## Author Response (AR5)

Dear Editor,
We are thankful to the anonymous referee and the editor for reviewing and helping us to improve this paper. The manuscript has been improved according to the suggestions of the Referee. In the revised manuscript, the modifications are in the bold letters.

Thank You!
Best regards,

Nadia Imtiaz.

**Reply to Anonymous Referee:-**

**P.1 L.20: Change "well-known fact" to "well known"**

**Response:** In the revised manuscript, on P.1 L.20: "well-known" has been replaced with "well known".

**P.11 L.5: Change "oxygen atom ionized" to "oxygen atoms are ionized".**

**Response:** In the revised manuscript, on P.11 L.5: "oxygen atom ionized" has been replaced with "oxygen atoms are ionized".

**P.11 L.6: Change "affect" to "affects"**

**Response:** In the revised manuscript, on P.11 L.6: "affect" has been replaced with "affects".

**P.11 L.4-6: The statement is either incorrect or formulated purely. It has to be modified "a large number of oxygen atoms are ionized that leads to … along with the high O/N2 ratio". O/N2 ratio is mainly controlled by the thermospheric neutral winds that are, in turn, related to the Joule heating in high latitudes.**

**Response:** In the revised manuscript, on P.11 L.4-6 the following modification has been done: a large number of oxygen atoms are ionized that leads to an increase in the ionospheric electron density along with the high $O/N_2$ density ratio.

**P.11 L.8 Replace "with a" with "while a"**

**Response:** In the revised manuscript, on P.11 L.9: "with a" has been replaced with "while a".

**P.11 L.18: Change "different longitudes" to "different hemispheres"**

**Response:** In the revised manuscript, on P.11 L.18: "different longitudes" has been replaced with "different hemispheres".

**P.12 L.4-5: Please clarify the statement "the O/N2 density ratio is larger in the eastern side than that is observed in the western side". What are the parts of the globe you are referring too: low/mid or high latitudes? It would be better to use sectors in the description, as it is done throughout the manuscript.**

**Response:** In the revised manuscript, on P.12 L.4-5 following modification has been done: The storm time longitudinal asymmetric behavior of the thermosphere can also be observed in the lower and middle latitudes over the four sectors. It is found that the thermospheric $O/N_2$ density ratio in the lower and middle latitudes over the African, Asian and eastern Pacific sectors is larger than that it is observed over the American and western Pacific sectors.

---

## Author Response (AR6)

Dear Editor,

We are thankful to you for reviewing and helping us to improve this paper. The corrections have been made in the manuscript.

Thank You!
Best regards,
Nadia Imtiaz.

1.Title "Geomagnetic" ---> "geomagnetic" (not use a capital "G" here!).
**Response: The correction has been made in the title.**

2.Abstract:Please define abbreviations "GPS" and "vTEC".
"Global Ultraviolet Spectrographic Imager" --→
"Global UltraViolet Spectrographic Imager (GUVI)".
"Thermosphere Ionosphere Mesosphere Energetics and Dynamics" --->
"Thermosphere Ionosphere Mesosphere Energetics and Dynamics (TIMED)".

**Response: The abbreviations "GPS" and "vTEC" have been defined in the abstract of final version of the manuscript as:**

- **Global Positioning System (GPS)**
- **vertical total electron content (vTEC)**

3.Introduction: Please define abbreviations "GPS", "TEC", "GNSS", "TIMED/GUVI".

**Response: In the final manuscript the abbreviations are defined as:**

- **Global Positioning System (GPS)**
- **total electron content (TEC)**
- **Global Navigation Satellite System (GNSS)**
- **Global UltraViolet Imager (GUVI)**
- **Thermosphere Ionosphere Mesosphere Energetics and Dynamics (TIMED) satellite.**

Line 19: "is the season dependent" ---> "is season dependent".
**Response: On Line 20 of the final manuscript "is the season dependent" has been replaced by "is season dependent".**